

# The annual cycle and sources of relevant aerosol precursor vapors in the central Arctic

Matthew Boyer[1], Diego Aliaga[1], Lauriane L. J. Quéléver[1], Silvia Bucci[2], Hélène Angot[3,4], Lubna Dada[3,5], Benjamin Heutte[3], Lisa Beck[6], Marina Duetsch[2], Andreas Stohl[2], Ivo Beck[3], Tiia Laurila[1], Nina Sarnela[1], Roseline C. Thakur[1], Branka Miljevic[7], Markku Kulmala[1], Tuukka Petäjä[1], Mikko Sipilä[1], Julia Schmale[3], Tuija Jokinen[1,8]

[1]Institute for Atmospheric and Earth System Research (INAR)/Physics, Faculty of Science, University of Helsinki, P.O. Box 64, Helsinki, 00014, Finland
[2]Department of Meteorology and Geophysics, University of Vienna, Vienna, Austria
[3]Extreme Environments Research Laboratory, Ecole Polytechnique Fédérale de Lausanne (EPFL), Sion, 1951, Switzerland
[4]Univ. Grenoble Alpes, CNRS, INRAE, IRD, Grenoble INP, IGE, Grenoble, France
[5]Laboratory of Atmospheric Chemistry, Paul Scherrer Institute, 5232, Villigen, Switzerland
[6]Institute for Atmospheric and Environmental Sciences, Goethe University Frankfurt, 60438 Frankfurt am Main, Germany
[7]School of Earth and Atmospheric Sciences, Queensland University of Technology, Brisbane, Australia
[8]Climate & Atmosphere Research Centre (CARE-C), The Cyprus Institute, P.O. Box 27456, Nicosia, 1645, Cyprus

*Correspondence to*: Matthew Boyer (matthew.boyer@helsinki.fi); Tuija Jokinen (t.jokinen@cyi.ac.cy)

**Abstract.** In this study, we present and analyze the first continuous timeseries of relevant aerosol precursor vapors from the central Arctic during the Multidisciplinary drifting Observatory for the Study of Arctic Climate (MOSAiC) expedition. These precursor vapors include sulfuric acid (SA), methanesulfonic acid (MSA), and iodic acid (IA). We use FLEXPART simulations, inverse modeling, sulfur dioxide ($SO_2$) mixing ratios, and chlorophyll-a (chl-a) observations to interpret the seasonal variability of the vapor concentrations and identify dominant sources. Our results show that both natural and anthropogenic sources are relevant for the concentrations of SA in the Arctic, but anthropogenic sources associated with Arctic haze are the most prevalent. MSA concentrations are an order of magnitude higher during polar day than during polar night due to seasonal changes in biological activity. Peak MSA concentrations were observed in May, which corresponds with the timing of the annual peak in chl-a concentrations north of 75°N. IA concentrations exhibit two distinct peaks during the year: a dominant peak in spring and a secondary peak in autumn, suggesting that seasonal IA concentrations depend on both solar radiation and sea ice conditions. In general, the seasonal cycles of SA, MSA, and IA in the central Arctic Ocean are related to sea ice conditions, and we expect that changes in the Arctic environment will affect the concentrations of these vapors in the future. The magnitude of these changes and the subsequent influence on aerosol processes remains uncertain, highlighting the need for continued observations of these precursor vapors in the Arctic.



## 1 Introduction

The Arctic environment is rapidly changing, as the region is warming up to four times faster than the rest of the planet (Rantanen et al., 2022). The central Arctic Ocean, which is largely covered by sea ice, is particularly sensitive to temperature changes due to climate feedbacks involving the decline of sea ice. The Arctic atmosphere is expected to change in response to lower sea ice extent through changing atmospheric transport patterns, increasing anthropogenic activities such as ship traffic, and increasing ocean-atmosphere exchange of gases and particles (Meier et al., 2014).

Such changes can have a direct influence on climate-relevant aerosol precursor vapors in the Arctic. Condensable vapors, either naturally emitted by the Arctic environment or transported from distant anthropogenic sources, can lead to secondary aerosol production, including new particle formation (NPF), which is estimated to be a key source of particles in the summertime Arctic atmosphere (e.g., Willis et al., 2018). When reaching a certain size, the newly formed particles can act as nuclei in cloud formation processes that influence the surface energy budget by scattering and absorbing incoming and outgoing radiation. NPF is estimated to yield up to 90% of the cloud condensation nuclei (CCN) number concentrations in the Arctic during summer, however, these estimates remain poorly constrained (Pierce and Adams, 2006; Dunne et al., 2016; Gordon et al., 2017; Kecorius et al., 2019). Moreover, aerosol-cloud interactions have an important, yet uncertain, role in the surface energy budget in the central Arctic Ocean where aerosol concentrations can be so low that cloud formation is CCN-limited (Mauritsen et al., 2011). Recent observations show that very small particles can act as CCN under such conditions in the Arctic (Koike et al., 2019; Baccarini et al., 2020; Karlsson et al., 2022), hence the sensitivity of the CCN population to aerosol precursor vapors is exceptionally high. The specific vapors that contribute to these climate-relevant processes in the Arctic have remained elusive, particularly over the central Arctic Ocean where data is scarce. Therefore, there is a need for more detailed characterization of the seasonal variability and sources of aerosol precursor vapors in the Arctic (Willis et al., 2018; Kecorius et al., 2019; Schmale and Baccarini, 2021).

Dimethyl sulfide (DMS) is a gaseous byproduct of phytoplankton and microalgae activities (Yoch, 2002) and the largest natural source of reduced sulfur in the atmosphere (Andreae, 1990). Specific phytoplankton species exude dimethylsulphoniopropionate (DMSP) in the water column, which can then be converted to DMS by microbial enzyme activity (Carpenter et al., 2012). A portion of the dissolved DMS subsequently outgases from the surface ocean to the atmosphere (Stefels et al., 2007). Once emitted, DMS oxidizes in the presence of sunlight to form secondary products that act as aerosol precursors, such as methanesulfonic acid (MSA), sulfur dioxide ($SO_2$), and sulfuric acid (SA) (Carpenter et al., 2012). In addition, there are other intermediate sulfur species that result from DMS oxidation, including dimethyl sulfoxide (DMSO) and methane sulfinic acid (MSIA), that readily undergo aqueous-phase processing. Reactions involving these intermediate species can contribute directly to sulfate aerosol mass in the marine boundary layer and affect the yield of MSA, $SO_2$, and SA from DMS oxidation (Hoffmann et al., 2016; Chen et al., 2018). $SO_2$ ultimately reaches its highest oxidation state as SA in the atmosphere, which typically requires sunlight, but has also been observed to occur under conditions with very low solar radiation (Dada et al., 2020; Sipilä et al., 2021). SA and MSA are important vapors for



secondary aerosol formation and growth processes (Kirkby et al., 2011; Zollner et al., 2012; Kulmala et al., 2013; Dunne et al., 2016; Gordon et al., 2017; Jokinen et al., 2018; Yan et al., 2018; Sipilä et al., 2021), where SA is one of the most well-known precursors for NPF and MSA has a demonstrated role in the growth of newly formed particles (Beck et al., 2021).

The concentration of gaseous DMS and its oxidation products (SA and MSA) in both the gas and particle phases have been observed to increase during spring and summer in the Arctic, which corresponds to the appearance of sunlight and phytoplankton activity (Ghahremaninezhad et al., 2017; Abbatt et al., 2019; Nielsen et al., 2019; Beck et al., 2021). Other studies suggest that DMS oxidation products affect the natural aerosol population and CCN concentrations in the marine atmosphere (Mayer et al., 2020) and over the Arctic Ocean (Ghahremaninezhad et al., 2019), which can impact the Arctic climate system (Charlson et al., 1987; Carslaw et al., 2010). Moreover, there is evidence that decadal DMS emission trends are positive across the Arctic due to decreasing sea ice coverage (Galí et al., 2019; Kurosaki et al., 2022), where the strongest increase in emissions is observed at higher latitudes and near the marginal ice zone (Leck and Persson, 1996b; Galí et al., 2021). At the same time, observations of DMS oxidation products in the aerosol phase in the Arctic do not show a persistent positive trend (Sharma et al., 2019; Schmale et al., 2022), which highlights the complexity of understanding aerosol processes involving these precursor vapors. As a result, DMS oxidation remains poorly represented in large scale climate models, despite its climate relevance, and thus, direct observations of its oxidation products can support modeling activities and lead to breakthroughs in the parameterization of chemistry climate models (Quinn and Bates, 2011; Hoffmann et al., 2016; Chen et al., 2018; Veres et al., 2020; Hoffmann et al., 2021).

Anthropogenic sources of sulfur are also important in the Arctic atmosphere, particularly during the so called "Arctic haze" phenomenon in winter (Quinn et al., 2007). During this season, $SO_2$ and particulate sulfate emissions from anthropogenic combustion processes, ore smelters, and shipping activities are transported into the Arctic (Barrie, 1986; Leaitch et al., 1989; Heidam et al., 1999; Tunved et al., 2013; Sharma et al., 2019; Sipilä et al., 2021; Corbett et al., 1999; Smith et al., 2001). Likewise, volcanic activity represents an occasionally large natural source of atmospheric $SO_2$ emissions (Fioletov et al., 2020). Therefore, both anthropogenic pollution and natural sources may influence the sulfur budget, and hence atmospheric SA concentrations, in the Arctic.

In addition to DMS oxidation products and SA, iodic acid (IA) has recently been identified as an important aerosol precursor gas for secondary particle formation (Allan et al., 2015; Sipilä et al., 2016). Detailed studies have identified IA to play a role in the initial steps of NPF in the Arctic atmosphere that may contribute to the CCN budget in the region (Baccarini et al., 2020; Beck et al., 2021). IA forms via atmospheric reactions with iodine, which is typically sourced from biological activities of micro algae and phytoplankton (O'Dowd et al., 2002; Ashu-Ayem et al., 2012; Allan et al., 2015). However, the mechanism of IA formation in the atmosphere is not fully resolved (Finkenzeller et al., 2023). While iodine oxo-acids were thought to involve reactions between HOx and reactive iodine (Khanniche et al., 2017; Plane et al., 2006; Drougas and Kosmas, 2005), He et al. (2021) showed that iodic acid can form via the oxidation of iodine atoms (obtained by photolysis of molecular iodine) under atmospherically relevant conditions with either ozone and water, or from the hydrolysis of intermediate IxOy compounds. New insights into the chemical mechanism show that IA formation can also occur at low





iodine concentrations due to catalytic recycling of iodine gas in the IA formation pathway (Finkenzeller et al., 2023).
Interestingly, these reactions can proceed without light if there is a source of reactive iodine. Furthermore, recent studies

have identified that IA formation follows a diurnal trend, where IA concentrations peak during periods of low solar radiation
(Sipilä et al., 2016; Jokinen et al., 2018; Baccarini et al., 2021; Quéléver et al., 2022). These results may have implications
for secondary aerosol formation during polar night or during seasonal transitions in the central Arctic.

Despite the knowledge from previous studies, there are very few in-situ observations of these gas phase species in the Arctic,
limiting our understanding of NPF, particle growth, and aerosol size distributions to resolve climate-relevant aerosol

processes in the central Arctic (Willis et al., 2018; Croft et al., 2019; Beck et al., 2021). Therefore, it is necessary to obtain
more measurements of atmospheric precursor vapors at high latitudes.

Herein, we present and analyze the concentrations of key aerosol precursor vapors, including MSA, SA, $SO_2$, and IA, in the
central Arctic during the Multidisciplinary drifting Observatory for the Study of Arctic Climate (MOSAiC) expedition. This
dataset represents not only the longest continuous timeseries of SA, MSA, IA, and $SO_2$ in the central Arctic and the first

measurements over the sea ice pack during the polar night, but also one of the longest continuous measurements of these
precursor vapors available in the literature. Our analysis focuses on the annual cycle in the concentrations of these aerosol
precursor vapors and their primary source regions using FLEXPART simulations, emission inventories, and inverse
modeling techniques. The results of this study are valuable for our understanding of these key aerosol precursor vapors and
their sources in the Arctic atmosphere, which is currently undergoing rapid changes.

## 2 Methods

### 2.1 The MOSAiC Expedition

The MOSAiC expedition was a ship-based campaign in the Arctic Ocean on R/V *Polarstern* (Knust, 2017) between
September 2019 and October 2020. The MOSAiC expedition and its objectives have been previously described in detail
elsewhere (Krumpen et al., 2020; Shupe et al., 2022; Nicolaus et al., 2022; Rabe et al., 2022). The ship drifted beside an ice

floe for a full year in the central Arctic Ocean and remained north of 80°N for almost the entire expedition. Refer to Fig. S1
in the supplemental information (SI) for a map of the drift track. MOSAiC was the largest expedition to the central Arctic in
history and the first with a full year of observation of various components of the Arctic environment, including during polar
night. The work presented here focuses on gas phase parameters measured from the *Swiss* container located onboard
*Polarstern*, as described below. Refer to Shupe et al. (2022) for further description of all the atmospheric measurements and

a diagram of the measurement containers onboard *Polarstern* during MOSAiC.

### 2.2 Precursor vapors from a chemical ionization mass spectrometer

The concentrations of MSA, SA, and IA in this study were measured using a nitrate based chemical ionization mass
spectrometer ($NO_3$-CIMS) (CI-APi-TOF, TOFWERK "HTOF", Jokinen et al., 2012) using 5-second time resolution. The



NO$_3$-CIMS was located inside the *Swiss* Container on the starboard side of the bow of *Polarstern* (refer to Beck et al., 2022).

The NO$_3$-CIMS sampled from a dedicated NPF-inlet that was specifically designed to minimize diffusional losses (reaching a 60 + 10 lpm combined inlet flow shared between a neutral cluster air ion spectrometer and the NO$_3$-CIMS, respectively). The NPF-inlet was ~1.3 m long, with a diameter of 10.2 cm, connected to the ¾ inch inlet tube of the CI-inlet by a KNF 50 flange (see Baccarini et al. (2020) for more details). The NO$_3$-CIMS had an individual inlet flow of 10 lpm, which was the result of the difference between a total vacuum flow of 40 lpm and a sheath flow of 30 lpm accomplished by a set of blowers

in a flow generation system (Airel Ltd. AFG -1). The sheath flow contained an additional flow saturated with nitric acid (HNO$_3$) vapor at a flow rate of 5 mlpm that was ionized by x-ray irradiation to form nitrate ions. Background measurements were performed periodically during the campaign by placing a HEPA filter in front of the instrument inlet to quantify the limit of detection (LOD), and a calibration for SA was performed post-campaign according to the procedure described in Kürten et al. (2012). A calibration factor of $6 \times 10^9$ molec·cm$^{-3}$ was obtained, and this calibration factor was then used for

concentration retrieval.

The time series of SA, MSA, and IA concentrations were obtained from TofTools (Junninen et al., 2010) by integrating peaks from the high-resolution mass spectra data at 5-minute averaged time resolution, normalizing the result with the sum of charger ions (NO$_3^-$, HNO$_3$NO$_3^-$, (HNO$_3$)$_2$NO$_3^-$), and multiplying by the calibration factor. SA was determined by peaks at mass to charge ratios (m/z) of 96.9601 Th (HSO$_4^-$) and 159.9557 Th (H$_2$SO$_4$NO$_3^-$), MSA was determined by m/z peaks at

94.9808 Th (CH$_3$SO$_3^-$) and 157.9765 Th (CH$_3$SO$_3$HNO$_3^-$), and IA was determined by m/z peaks at 174.8898 Th (IO$_3^-$) and 237.8854 Th (HIO$_3$NO$_3^-$). The LOD was determined for each species according to the method discussed in He et al. (2023) as follows:

$$LOD = \mu + 3 \times \sigma, \tag{1}$$

where $\mu$ is the average concentration and $\sigma$ is the standard deviation during a filter measurement. The resulting LODs are

$8.8e^4$, $1.5e^5$, and $5.5e^4$ molec·cm$^{-3}$ for SA, MSA, and IA, respectively. The uncertainty range of the measured concentrations from the NO$_3$-CIMS is estimated to be −50%/+100% (Jokinen et al., 2012), while the reported concentrations should mostly be considered as low limit values. This is due to the assumption that MSA and IA are charging at the kinetic limit, and therefore, concentrations reported herein may be underestimated (Ehn et al., 2014; Sipilä et al., 2016).

## 2.3 Local pollution in the NO$_3$-CIMS measurements

Local pollution is an important consideration in ship-based campaigns, as it affects the baseline of ambient samples (Beck et al., 2022). During MOSAiC, local pollution sources included the vessel and logistic activities surrounding the central observatory (helicopter flights, snowmobiles etc.). There are three possible ways in which local pollution could bias the NO$_3$-CIMS measurements: (1) the high particulate loading in pollution plumes could increase the condensation sink and enhance the loss of condensable vapors onto particle surfaces; (2) the pollution could act as a direct source of the gas phase

species; and (3) the pollution could act as a source of primary species that are subsequently oxidized to form species of interest. The secondary formation of SA from SO$_2$ is particularly of concern, as ship stack pollution is a large source of SO$_2$.





Note that the influence of local pollution was removed from the $SO_2$ timeseries data, as described in the following section (2.4).

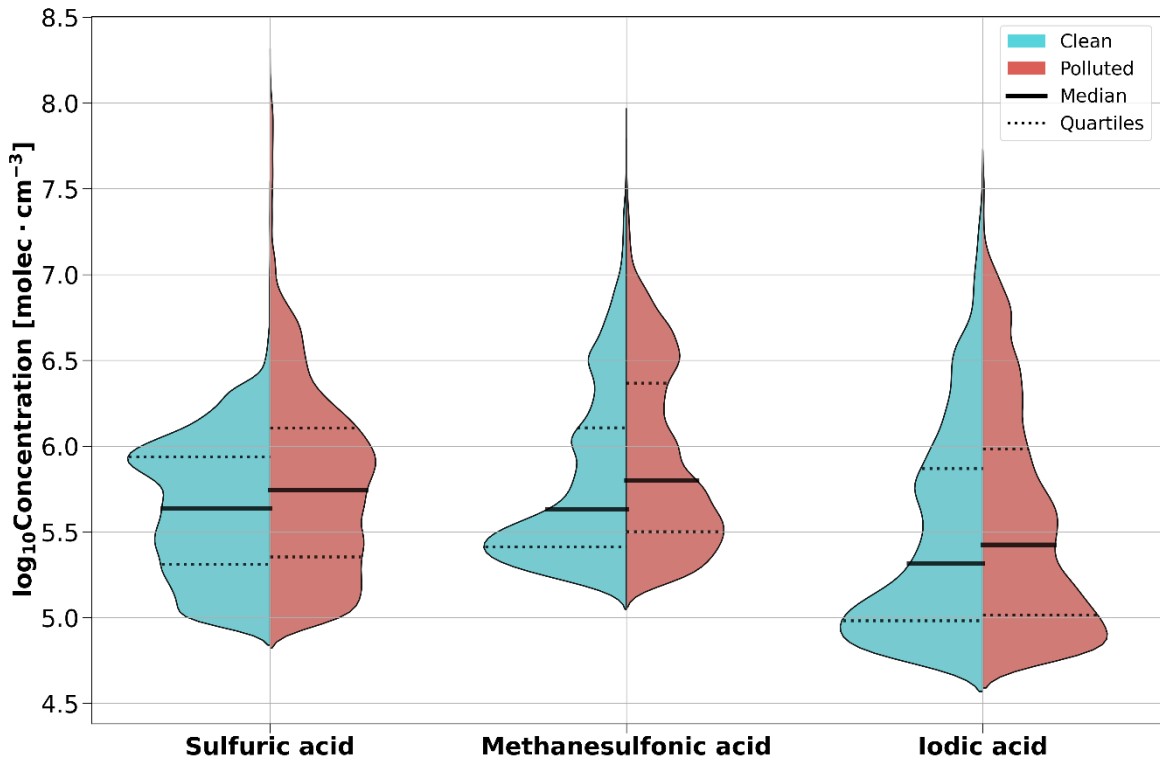

**Figure 1: A comparison of the gas phase timeseries for SA, MSA, and IA during clean and polluted periods**. The violins show the frequency distribution of the gas phase concentration for the clean (red) and polluted (blue) periods to evaluate the influence of local pollution on each species during the campaign. The medians and interquartile ranges of each data subset are represented by the solid and dashed black lines, respectively.

To evaluate the effect of local pollution on the $NO_3$-CIMS measurements, we followed the approach of Baccarini et al. (2021) by comparing the gas phase concentrations of SA, MSA, and IA during clean and polluted periods (Fig. 1). Polluted periods were determined by applying the pollution detection algorithm developed by Beck et al. (2022) to particle concentrations from a condensation particle counter (TSI, model 3025) located in the *Swiss* container. This method identified that 67% of the 5-minute $NO_3$-CIMS data were influenced by primary pollution (out of 87506 total data points). For further details on timing and extent of pollution in the timeseries of SA, MSA, and IA, refer to Figs. S2, S3, and S4, respectively. Overall, the comparison presented in Figs. 1, S2, S3, and S4 suggests that the influence of pollution was negligible for all three gas phase species during the MOSAiC expedition. While the median polluted concentrations were slightly higher for



all species, we conclude that this is coincidental due to natural variability of the gas phase concentrations over time given the

length and extent of pollution episodes in the timeseries. We also expect that secondary pollution and natural variability may be seasonal in the high Arctic due to environmental changes between polar day and polar night, especially for secondary formation of SA from $SO_2$, which typically requires solar radiation. Moreover, the conversion of $SO_2$ to SA occurs on the order of hours under atmospheric conditions (Lee et al., 2011), hence the effect of the ship stack emissions may only be minimal in our measurements. The comparison for SA in Fig. 1 suggests that pollution may have been influential for a small

fraction of SA concentrations higher than ~ $10^7$ molec·cm$^{-3}$ during the year, however, the effect is minimal and again may be coincidental. These periods of high SA concentrations are discussed further in section 3.1. Therefore, we concluded that SA, MSA, and IA were not affected by primary pollution in our measurements, and we did not remove any data during polluted periods.

**2.4 Sulfur dioxide**

Ambient $SO_2$ mixing ratios were measured in the *Swiss* container using a Thermo Fisher Scientific instrument (model 43i). The lower limit of detection for this $SO_2$ instrument is 1 ppb. The measured $SO_2$ mixing ratios were adjusted following cross-evaluation against a certified $SO_2$ standard at the Swiss Laboratories for Materials Science and Technology (EMPA), and the influence of local pollution was removed. For further discussion on the laboratory evaluation and pollution detection and removal in the $SO_2$ dataset, refer to Angot et al. (2022).

**2.5 Satellite ocean color data (chlorophyll-a)**

Eight-day averages of chlorophyll-a (chl-a) concentrations were downloaded from the Ocean-Colour Climate Change Initiative (OC-CCI, http://www.esa-oceancolour-cci.org) with a spatial resolution of 4 km. The OC-CCI is a long-term, consistent, and error-characterized dataset generated from merged normalized remote-sensing reflectance data derived from five satellite sensors: MERIS, Aqua-MODIS, SeaWiFS, VIIRS, and Sentinel3A-OLCI data (Sathyendranath et al., 2019).

Cloudiness or algorithm failure can result in missing data in the Arctic, however, overall, there are typically enough cloud-free data available to evaluate chl-a in the Arctic region during summer (e.g., Becagli et al., 2016).

**2.6 Source region identification using FLEXPART and inverse modeling**

FLEXPART simulations and an inverse model were used to evaluate influential source regions of the aerosol precursor vapors. FLEXPART is a Lagrangian particle dispersion model that can be used to simulate air mass origins. Backward

simulations with FLEXPART v10.4 (Pisso et al., 2019) were used to determine the "Footprint Emission Sensitivity" (FES), or the emission sensitivity < 100 m, according to the simulated "source-receptor relationship" (SRR) (Seibert and Frank, 2004). The FES can then be coupled with an emission inventory, such as ECLIPSE (Evaluating the Climate and Air Quality Impacts of Short-Lived Pollutants), to simulate concentrations at the ship. Refer to Fig. 2 in Boyer et al. (2023) for the FES observed during the MOSAiC expedition.





Additional source region analyses were performed using an inverse modelling technique (Seibert, 1998). The inverse model uses the FLEXPART SRR and the measurement time series data from the ship to identify potential source regions and estimate their emissions using the following equation:

$$y = \mathbf{A}x + n, \tag{2}$$

where $y$ is the measurement time series data, $\mathbf{x}$ is the source term, $\mathbf{A}$ is the FLEXPART SRR transport matrix, and $n$ is the

error. To simplify the model, we assumed that the source term is constant over time. In addition, we used a clustering technique to reduce the dimensionality of the FLEXPART transport matrix to 200 groups, as described in Aliaga et al. (2021) and Faletto and Bien (2022). We also applied an Elastic Net regularization method (Zou and Hastie, 2005; Martinez-Camara et al., 2014 and references therein) using the following equation:

$$\hat{\mathbf{x}} := \underset{\mathbf{x}}{\mathrm{argmin}} \, \frac{\|\mathbf{y} - \mathbf{A}\,\mathbf{x}\|_2^2}{2\,N} + \left( \alpha \, \rho \, \|\mathbf{x}\|_1 + \frac{\alpha(1-\rho)}{2} \|\mathbf{x}\|_2^2 \right), \tag{3}$$

where $\hat{\mathbf{x}}$ is the updated source term with penalization applied, $y$ is the measurement time series data, $\mathbf{x}$ is the source term from Eq. 2, $N$ is the number of time steps in the time series, $\alpha$ is the regularization parameter that controls the overall strength of the penalty, and $\rho$ is the mixing parameter that balances the contributions of Lasso and Ridge penalties (described below). A larger value of $\alpha$ results in more regularization and a simpler model with smaller coefficients. The $\rho$ parameter ranges from 0 to 1. When $\rho$ is set to 1, the Elastic Net model becomes equivalent to a Lasso regression. When $\rho$ is set to 0, it

becomes equivalent to a Ridge regression.

The objective function of the Elastic Net model (Eq. 3) consists of two terms. The first term is the Mean Squared Error (MSE) loss, which quantifies the difference between the observed and predicted responses. The second term is the regularization penalty, which is a combination of the Lasso penalty (L1 regularization) and the Ridge penalty (L2 regularization). The Lasso penalty promotes sparsity by discouraging non-zero coefficients, while the Ridge penalty

encourages small coefficients by penalizing large ones. The Elastic Net penalty is a weighted combination of both penalties, controlled by the mixing parameter $\alpha$.

A common challenge in applying an Elastic Net model is selecting the appropriate values for $\alpha$ and $\rho$. One approach to address this issue is to use cross-validation. Cross-validation is a statistical technique used to assess the performance of machine learning models. It involves dividing the dataset into multiple subsets, training the model on some of these subsets,

and evaluating it on the remaining subsets. This process is repeated multiple times, and the average performance is computed to provide a more reliable estimate of the model's performance.

In our case, we use cross-validation to determine optimal values of $\alpha$ from an initial list of $\rho$ values (0.1, 0.5, 0.9, 0.99, 0.9999, 1). We then construct a matrix of potential solutions by multiplying the optimal values of $\alpha$ obtained before by the following list (L): (1/8, 1/4, 1/2, 1, 2, 4). This results in a matrix of 36 foot-print maps, with rows determined by the $\rho$ values

and columns by L x $\alpha$. Based on prior knowledge of possible source regions, we select one of the source region maps that identifies probable source regions while minimizing noisy regions. Refer to Figs. S5, S6, and S7 in the SI for the source region foot-print maps for $SO_2$, SA, and MSA obtained by the Elastic Net model in this study. The result is a map of



potential source contributions, much like an emission inventory. Using this map of estimated emissions and the FLEXPART FES, the model can simulate measured concentrations at the ship, in a similar way as using the FES coupled with the

emission inventory.

Then, the inverse model identifies source region clusters, represented as polygons on a map, that are sorted by their relative contribution to the simulated concentrations. To filter noise, we only considered clusters where the annual mean source contribution was > 5% relative to all clusters for further analyses. Adjacent clusters were grouped to simplify the interpretation of the results. Finally, we obtained a timeseries of simulated influence from the source region polygons, which

we used to provide qualitative insights on the seasonal contributions from different source regions.

It is important to note that the inverse model assumes a constant emission rate for each identified source region and does not include chemical processing during transport, which would have to be simulated by FLEXPART rather than the inverse model. While this approach simplifies the implementation of the inverse model, it may not describe the true nature of gas phase emissions from the various source regions. Due to the constant emission limitation, the MSA inverse model

simulations were carried out from March – September, or the time during which we expect DMS emissions that lead to the observed maximum MSA concentrations. Since the emission of iodine species necessary for the formation of IA are highly variable, the assumption of constant emissions cannot be made. Therefore, we do not present inverse model results for IA.

## 2.7 Global radiation and temperature

Global radiation, or short-wave downwelling radiation, was measured using a Pyranometer (Kipp & Zonen, CM11), and
ambient air temperature was measured using a Vaisala HMP155. Both sensors were installed on R/V *Polarstern* and operated continuously during the MOSAiC campaign. The global radiation and the temperature sensors were located 34 and 29 m above the sea surface, respectively.

## 3 Results and discussion

In this section, the annual cycle of various precursor vapors is presented and discussed. The monthly median concentrations
of vapors that are known to participate in aerosol formation in remote marine environments, including SA, MSA, and IA, are presented in Fig. 2a. The timeseries of other parameters that provide context to the gas phase species, including temperature, global radiation, the ship's latitude, and satellite observations of chl-a concentrations in seawater are presented in Fig. 2b & 2c. We also present and discuss the measured $SO_2$ mixing ratios in Fig. 3. These datasets offer unique insights on the seasonal variability of these vapors in the high Arctic. Given the high latitude of these measurements, the seasonal cycle that

we present is essentially a "diurnal" cycle between polar day and polar night. Herein, we give an in-depth discussion on the annual cycle of each species individually.







**Figure 2: The annual cycle of aerosol precursor vapor concentrations and related parameters during MOSAiC.** (A)
The monthly median vapor concentrations and the corresponding interquartile ranges of iodic acid (IA), methanesulfonic
acid (MSA), and sulfuric acid (SA) are presented as the dotted lines and shaded regions, respectively. The concentration data
at a 5-minute time resolution are also shown (thin lines). Note that concentrations below the LODs for each vapor species
were removed. (B) The temperature and global radiation measurements are shown at 1-minute resolution with the monthly
averages and interquartile ranges overlayed for context. (C) The chl-a concentrations are shown as 8-day averages integrated
for every 5° of latitude between 60 – 90°N from the OC-CCI satellite dataset. The lack of color between 85 – 90°N indicates
that there is no data available between these latitudes. The dotted white line denotes the monthly median latitude of
*Polarstern* during the campaign. The orange circles show the sea surface chl-a influence index derived from the convolution
of the FLEXPART FES with the satellite chl-a concentrations using air mass ages of 10 days.

### 3.1 Sulfur dioxide and sulfuric acid: anthropogenic and biogenic sulfur species

The median SA concentration observed during the year was $5.1\times10^5$ molec·cm$^{-3}$ (SD = $6.1\times10^6$ molec·cm$^{-3}$), but most
notably, our measurements show a large increase in SA concentrations from January through April, where the median SA
concentration (avg = $9.7\times10^5$ molec·cm$^{-3}$, SD = $9.4\times10^6$ molec·cm$^{-3}$) is approximately twice the annual median (Fig. 2a). We
also observed short-lived spikes of SA concentrations up to $1\times10^8$ molec·cm$^{-3}$ in January and February (Fig. 3). In general,
the monthly median concentrations agree with campaign observations reported from land-based sites in the Arctic (Beck et
al., 2021) and sub-Arctic (Jokinen et al., 2022), and the short-lived spikes of SA in January and February are similar in
magnitude to concentrations observed during local pollution events related to traffic emissions in Helsinki, Finland (Okuljar
et al., 2021; Thakur et al., 2022).

The high concentrations in winter and spring (January – April) indicate a significant source of SA, even during the cold and
light-limited central Arctic environment when biological processes are largely absent (Fig. 2c). The periods of high SA
concentrations in winter and spring coincide with the occurrence of Arctic haze in the aerosol size distribution during
MOSAiC that was driven by a strong positive phase of the Arctic Oscillation (AO) (Lawrence et al., 2020; Boyer et al.,
2023), which suggests that Arctic haze, or anthropogenic pollution, is a key source of the high SA concentrations in winter
and the dominant source of SA during the annual cycle. Indeed, previous analyses have identified that sulfur/sulfate is a key
component of Arctic haze (e.g., Barrie, 1986; Quinn et al., 2007, 2009; Gong et al., 2010; Schmale et al., 2022).



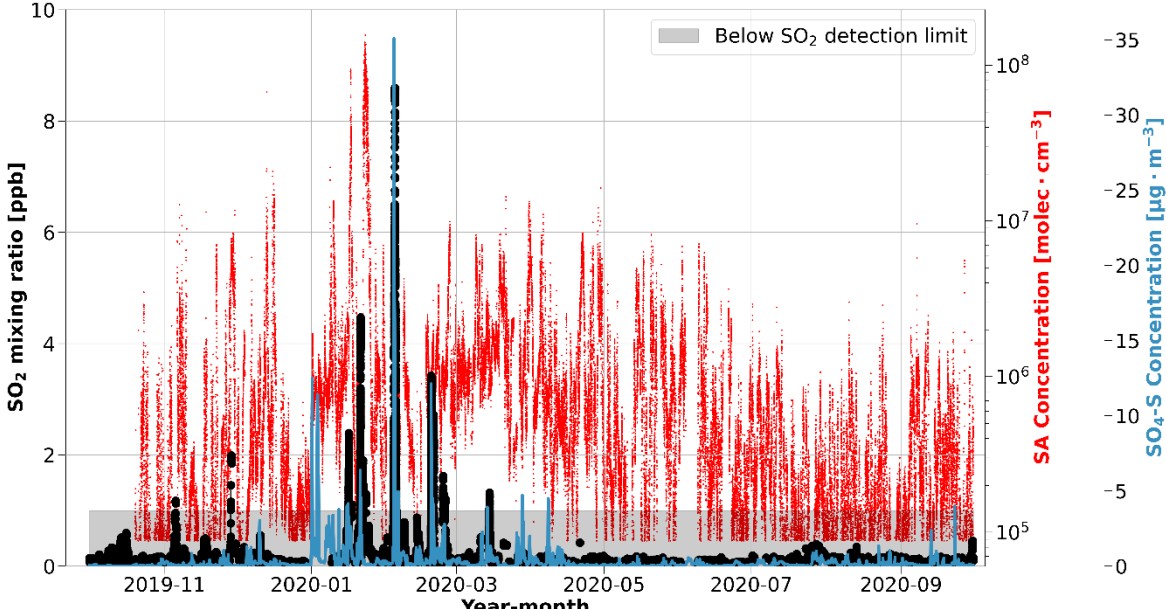

**Figure 3: The seasonal cycle in SO₂ mixing ratios compared to SA during MOSAiC.** The SO$_2$ mixing ratio (black line) is presented as a rolling 10-min median. The shaded gray area identifies the data below the detection limit of the SO$_2$ instrument (1 ppb). The time series of SA concentration (red line) and the simulated mass concentration of sulfate in the aerosol phase (blue line) at the position of *Polarstern* from FLEXPART coupled to ECLIPSE v6b emission inventory (SO$_4$-S) are included to highlight the consistent behavior between the sulfur species.

SO$_2$ is the primary gaseous sulfur species that, once emitted, can form SA in the atmosphere. As a result, we can use the timeseries of SO$_2$ mixing ratios to provide further insight on the sources of SA in the Arctic (e.g., Dada et al., 2020). Comparing the SA data with SO$_2$ mixing ratios leads to a similar conclusion during the winter months—Arctic haze. Throughout most of the year, the SO$_2$ mixing ratio is below the detection limit of the instrument, which is 1 ppb. Only 0.6% of the ship exhaust pollution filtered SO$_2$ dataset is above the limit of detection during the entire campaign. The few occasions that we observe SO$_2$ mixing ratios above the detection limit occurred between November – March, especially during January and February (Fig. 3). The high SO$_2$ mixing ratios occur during the same time of year that SA concentrations reach their highest values, which is again consistent with the timing and occurrence of Arctic haze. Further discussion of these SO$_2$ plumes, which are associated with episodic warm and moist air mass intrusion events, is given in Angot et al. (in prep). It is worth noting that the photochemical conversion of SO$_2$ to SA and the details of the sulfur chemistry is out of the scope of this work and is not explored further.

To further evaluate the sources of SO$_2$ and SA in our measurements, we examined emissions of anthropogenic sulfate in the aerosol phase (SO$_4$-S), which were simulated using the ECLIPSE v6b emission inventory coupled with the FLEXPART



simulations. Since the conversion of $SO_2$ to sulfate aerosol usually occurs on the order of hours in the atmosphere (Lee et al.,
2011), FLEXPART treated anthropogenic $SO_2$ emissions as $SO_4$-S, which yields the $SO_4$-S weighted influence from anthropogenic sources. As such, $SO_4$-S is useful to interpret the sources of $SO_2$, and infer the sources of SA, especially for distinguishing between anthropogenic and natural sources. The simulated $SO_4$-S concentrations are included in Fig. 3 for comparison with the SA and $SO_2$ timeseries. Overall, the $SO_4$-S simulations agree well with the measured gas phase sulfur species, especially in January and February when we observed temporal spikes of each species (Fig. 3). The geographic
regions associated with the $SO_4$-S concentrations, determined by applying a geographic mask to the simulations, are presented in Fig. 4. Refer to Fig. S8 for a more detailed description of anthropogenic $SO_4$-S emissions from each source region in the emission inventory. The results suggest that Northern Asia dominates the anthropogenic emissions of sulfur species during January and February. This supports our conclusions; anthropogenic sulfur emissions have a strong influence in this region, as shown in the satellite-derived emission inventory of $SO_2$ described by Liu et al. (2018). The sulfur
emissions in this region predominately originate from smelters in Norilsk, Russia that are identified as strong point-source emitters (Khokhar et al., 2008), which is consistent with the $SO_4$-S loading in the emission inventory from Northern Asia in Fig. S4.

The inverse modeling analysis for $SO_2$ (Fig. 5) identified three regions as potential contributors to $SO_2$, which we attribute to North Asia/Siberia, Europe, and the Aleutian Peninsula in Alaska. Of these three sources, North Asia/Siberia is dominant.
More specifically, the dominant source region, highlighted by polygon a in Fig. 5, agrees well with the prevalent source of atmospheric sulfur in the emission inventory near Norilsk, Russia (Fig. S8). Therefore, these results again demonstrate that the smelter region in Norilsk, Russia exerts a significant influence on the concentrations of SA and $SO_2$ in the central Arctic during the Arctic haze period (Hirdman et al., 2010; Bauduin et al., 2014). These results agree with Sipilä et al. (2021) who observed that $SO_2$ pollution from smelters in Kola Peninsula, Russia, contributes to a wintertime source of SA even during
the low light conditions of winter. The timing of our wintertime observations of $SO_2$ and SA is also consistent with aerosol observations from the MOSAiC expedition that show an early peak in Arctic haze pollution from Eurasia during January – March 2020 that is linked to an extreme positive phase of the Arctic Oscillation index (Lawrence et al., 2020; Boyer et al., 2023). It is known that these conditions lead to high pollution in the Arctic (Eckhardt et al., 2003). Refer to Angot et al. (in prep) for further analysis of these episodic pollution events associated with high $SO_2$ mixing ratios during polar night on the
MOSAiC expedition.







**Figure 4: Simulated source regions of anthropogenic SO$_4$-S concentrations.** A source region mask (A) was used to identify the contribution of each source region to the SO$_4$-S concentrations in the aerosol phase (B) using the FLEXPART FES and the ECLIPSE v6b emission inventory. The SO$_4$-S concentrations are presented as monthly averages according to source region. Note that while the general sea ice coverage is presented in the source region mask as the white shaded area, there are no SO$_4$-S emissions associated with this region to show in (B). Emissions associated with the Ocean region are due





to anthropogenic emissions from ships. A description of the average SO$_4$-S contributions from each of the regions during the entire year is provided in the SI to show the specific spatial distribution of the sources in the emission inventory (Fig. S8).

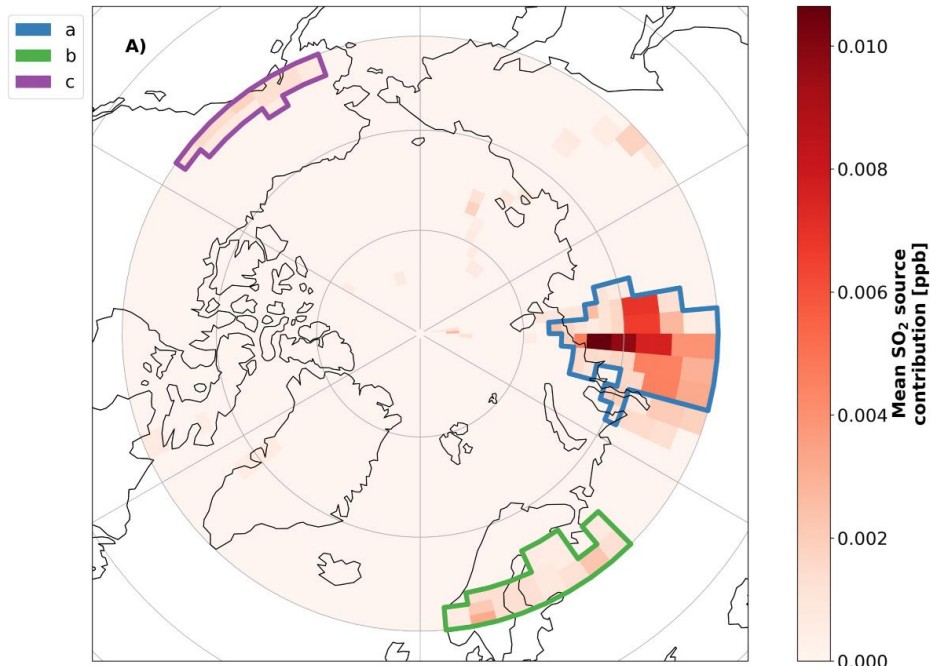

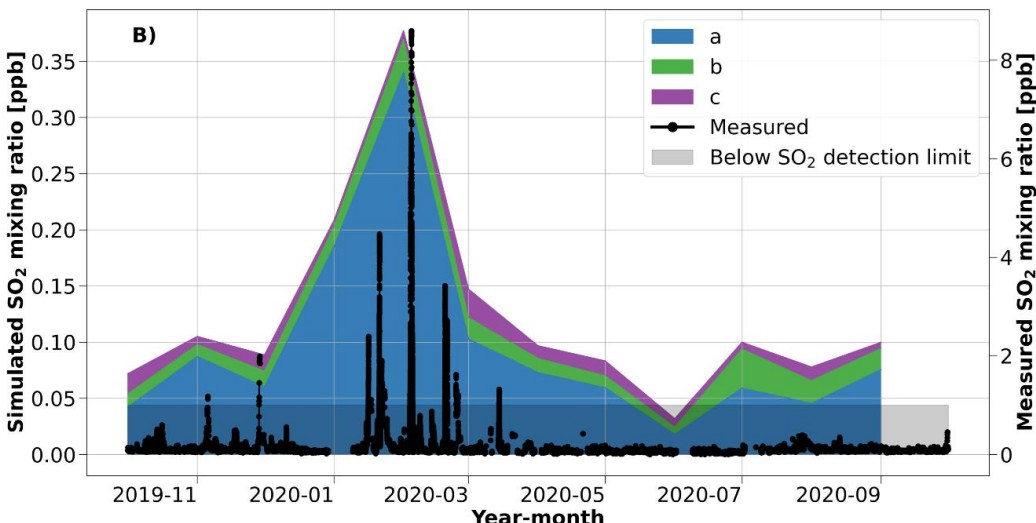


**Figure 5: Inverse model results for SO$_2$.** (A) The potential source regions identified using the FLEXPART air tracer data and the ambient SO$_2$ mixing ratio in the inverse model. The color bar shows the simulated annual mean contribution of the





source regions to the SO$_2$ mixing ratio during the year of the MOSAiC expedition. (B) A timeseries of the monthly median SO$_2$ mixing ratio contributions from the identified source regions, as simulated by the inverse model. The measured SO$_2$

mixing ratio, presented as a rolling 10-minute median during the year, is included for context.

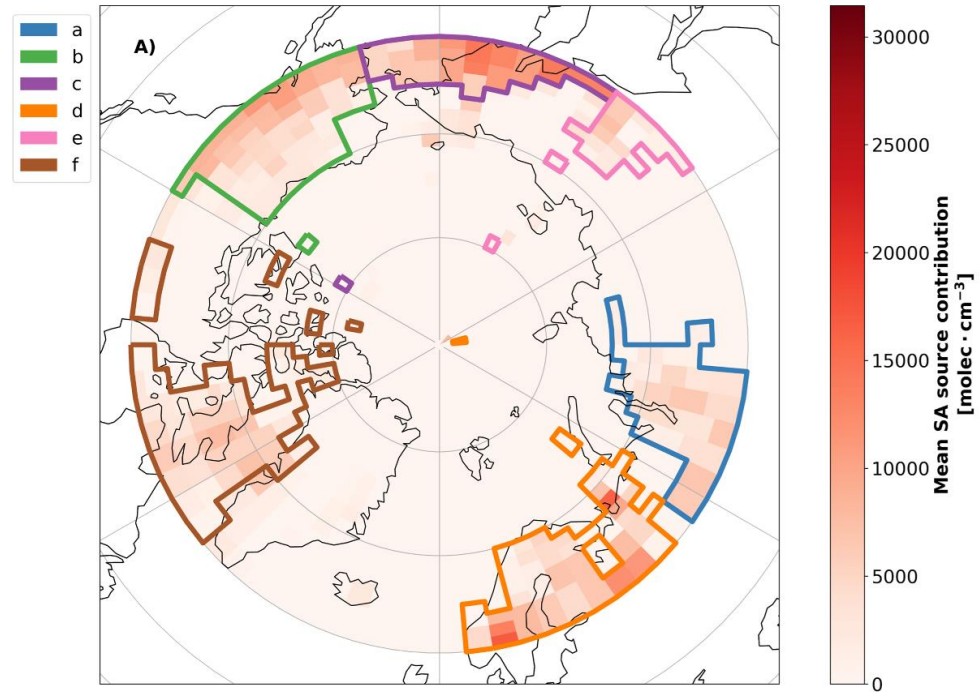

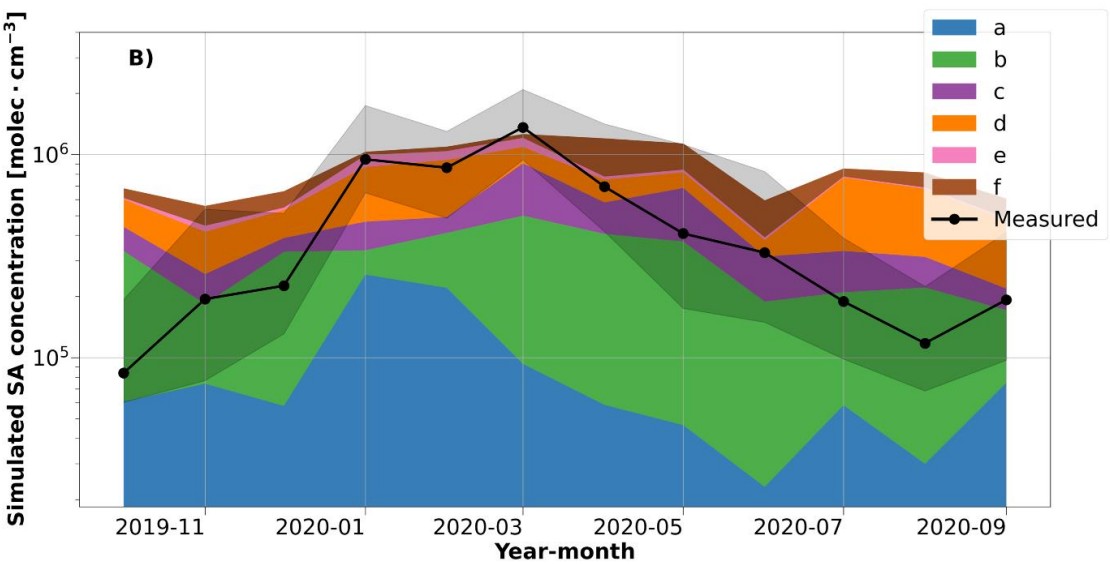



**Figure 6: Inverse modeling results for SA.** (A) The potential source contribution map of SA simulated by the inverse model. The red color bar shows the annual average of the potential source region contributions. The colored polygons show the source region clusters that were identified to have a significant contribution (> 5% annual average) on the simulated SA concentrations during the year, which are used in the time series analysis. (B) The simulated SA concentration timeseries according to the potential source region clusters identified by the inverse model. The time series shows the potential contribution of each source region polygon to SA concentrations throughout the year. Note that the source areas here are different from those shown in Fig. 5.

Despite lower anthropogenic $SO_4$-S influence from the North Asia sector during March and April, the SA concentration remained high and even reached a maximum monthly median concentration of $1.0 \times 10^6$ molec·cm$^{-3}$ during March (Fig. 2a). This result suggests that there is another source of the high SA observed during these months. As such, the inverse model was used to identify potential source regions for SA in March and April (Fig. 6). The inverse model results for SA also show that contributions from North Asia/Siberia decline in March and April, however, other source regions become more influential during this time. The more influential source regions correspond to polygons b and c in Fig 6a, which we attribute as the Aleutian Peninsula in Alaska and the Kamchatka Peninsula in Russia, respectively, both regions with volcanic activity. Polygon b likely also includes anthropogenic emissions from oil fields in Prudhoe Bay, Alaska. These source regions are consistent with major point source emissions of $SO_2$ (Fioletov et al., 2020), and hence SA production via photochemistry. The polygon associated with Alaska also agrees with the inverse model results for $SO_2$ in Fig. 5, which also identifies potential contributions from the Aleutian Peninsula region during March. It is also expected that the appearance of sunlight in the central Arctic during March and April contributes to enhanced oxidation of $SO_2$ to SA during these months compared to winter, resulting in the peak SA concentration in March.

SA is also a product of DMS oxidation, a process that requires biological activity and sunlight (i.e., during polar day), however, we do not observe enhanced SA concentrations in the summer. Instead, we observed that winter (i.e., polar night) and spring dominate the annual cycle of SA in the central Arctic. According to our measurements, the median SA concentration during the Arctic haze period (January – April) is between 3 – 4 times higher than the median SA concentration during periods of peak biological activity (May – July). This observation is contrary to typical patterns of SA behavior, where SA concentrations peak during daytime when global radiation is high (Baccarini et al., 2021; Quéléver et al., 2022). We can primarily attribute this difference in our measurements to the seasonal variation of sinks in the central Arctic. In winter and spring, Arctic haze builds up because there is little precipitation during air mass transport, leading to the accumulation of SA. In contrast, there is more convection during summer, and hence more precipitation, which limits northward transport of anthropogenic pollution (Klonecki et al., 2003). Based on this, our results demonstrate that SA concentrations from DMS emissions in summer are small compared to anthropogenic sulfur sources from Arctic haze. In addition, the cold air temperatures in the central Arctic during the year (Fig. 2b) suggest that DMS oxidation likely favors the





production of MSA rather than SA during summer (Shen et al., 2022), which could also help explain the lack of elevated SA concentrations during periods of higher biological activity over the summertime central Arctic Ocean (Fig. 2c).

SA has a well-known role in NPF processes (McMurry and Friedlander, 1979; Kulmala et al., 2013; Dunne et al., 2016; Gordon et al., 2017; Jokinen et al., 2018), and hence, higher concentrations in the central Arctic atmosphere could have a
notable impact on aerosol-related phenomena in the region, especially during summer months when SA concentrations are currently relatively low. Increased $SO_2$ emissions from ships could become a source of SA in the central Arctic as ship traffic could become more frequent with less sea ice extent, however, it is unclear how future shipping emissions might impact SA concentrations. For instance, when primary pollution from *Polarstern* influenced our data, we observed $SO_2$ concentrations exceeding 10 ppb, while only 0.6% of the not-*Polarstern*-influenced $SO_2$ data were above the detection limit of the
instrument (1 ppb). The periods with data above the detection limit only occurred in the winter months, as described above. In addition, recent observations show that $SO_2$ emissions, and subsequently SA, can lead to the formation of new particles even during low light conditions in winter/spring in the Arctic (Sipilä et al., 2021). Our observations highlight the combined influence of both natural and anthropogenic sources of atmospheric SA during the year in the central Arctic, where the highest concentrations occurred in winter/spring.

**3.2 Methanesulfonic acid: an oxidation product from biological activity**

We observed a minimum MSA concentration during the fall and winter months, particularly from October through February, with a median concentration of $2.9 \times 10^5$ molec·cm$^{-3}$ (SD = $4.8 \times 10^5$ molec·cm$^{-3}$) throughout this period (Fig. 2a). A clear and consistent increase was observed in the monthly MSA concentrations starting in March, when the global radiation starts to increase in the high latitudes of the Northern Hemisphere (Fig 2b). The MSA concentration continued to rise after March,
reaching a maximum monthly median concentration of $3.3 \times 10^6$ molec·cm$^{-3}$ in May, followed by high concentrations through August, with a median concentration of $1.5 \times 10^6$ molec·cm$^{-3}$ (SD = $3.0 \times 10^6$ molec·cm$^{-3}$) from June through August. The concentration then decreases again in September, completing the annual cycle. In general, our observed MSA concentrations are similar in magnitude to those reported previously in Ny-Ålesund, Svalbard, Norway, especially during the annual peak in May-June (Beck et al., 2021).

Chl-a, which is a well-known tracer of biological activity (e.g., Park et al., 2013), was used to evaluate the connection between MSA and biological activity in the ocean. While chl-a does not provide a direct measure of DMS production (Stefels et al., 2007; Uhlig et al., 2019), we used the chl-a concentrations to approximate the timing of biological activity in ocean regions between 60 and 90°N. The chl-a concentrations, presented in Fig. 2c, suggest that the MSA concentrations in the central Arctic are strongly linked to biological activity from regions south of the marginal ice zone, which agrees with
previous investigations (Beck et al., 2021). The annual minimum in the MSA concentrations in December and January also corresponds to the lowest chl-a concentrations in the Northern Hemisphere. This period of low MSA and chl-a concentrations occurs when the biological activity in the Arctic and surrounding ocean is limited due to lack of sunlight. Then, after February, the chl-a concentrations start increasing at the lower latitudes, progressing northward during the spring



and summer seasons. The MSA concentrations follow the same trend as chl-a, reaching a maximum during May when the chl-a concentration north of 75°N reaches its annual peak (Fig. 2c). The simultaneous increase in chl-a and MSA concentrations is unsurprising, as gaseous DMS, and subsequently MSA, is a product of biological activity in the ocean. The comparison of the chl-a and MSA timeseries also suggests that the MSA concentrations in the central Arctic are influenced by transport from oceanic regions further south. This is particularly clear during March and April when the MSA concentration in the central Arctic starts to increase despite the observation that chl-a concentrations are still low at the northernmost latitudes. The increase presumably occurs due to the onset of biological activity further south combined with transport. This is also consistent with more favorable transport of airmasses from southerly locations during spring (Bozem et al., 2019).

While the chl-a concentrations provide insight about potential regions of biological influence, they alone cannot describe the MSA observations. The source regions of the observed air masses in the central Arctic would need to correspond with the regions of enhanced biological activity to explain the MSA measured at the ship. Therefore, we coupled the FLEXPART air tracer simulations with the oceanic chl-a concentrations to calculate an index that quantifies potential air mass exposure to oceanic regions with biological influence, and hence potential DMS (MSA) emissions. The index, called the sea surface chl-a influence index, was obtained by convoluting the FLEXPART FES (residence time below 100 m) with the chl-a concentration maps. Hence, the index is proportional to the amount of time that the air masses have spent over regions with chl-a presence and to the concentrations of chl-a encountered in those regions. The sea surface chl-a index results are also shown in Fig 2c, and several maps of individual trajectories are included in Fig. S9 to demonstrate how the index was calculated. As previously stated above, we do not expect DMS emissions to correspond directly with chl-a, but chl-a serves as a tracer for air masses with potential influence from oceanic biological activity. The index clearly indicates that the influence of ocean biology increased during the summer, which is generally consistent with the seasonal enhancement in MSA.

We further examined source regions of MSA by using the inverse model. The key insight obtained from the inverse model results, shown in Figs. 7a and 7b, is that regions south of the marginal ice zone appear to be the most influential on MSA concentrations over the central Arctic. More specifically, the inverse model identifies several oceanic regions as potential sources of MSA in our observations, where the Kara, Barents, Norwegian, and Labrador Seas are the most prevalent source regions during spring and summer (polygons b, c, and d in Fig. 7a). These regions also agree well with individual trajectories from the coupled FLEXPART and chl-a concentration analysis (refer to Fig. S9). Note that due to the limited domain of the FLEXPART simulations (> 60°N), source regions polygons a and f in Fig. 7a may represent the contribution of MSA transport from regions further south than the polygons depicted on the map, which could be associated with oceanic regions on the western coast of North America and Bering Sea, respectively. Previous research has shown that the regions identified in Fig. 7a are biologically active or important sources of DMS, the precursor of MSA, from May to August (Leck and Persson, 1996a; Lana et al., 2011; Hulswar et al., 2022; Terhaar et al., 2021), which is consistent with the chl-a satellite data and again highlights the importance of air mass transport from biologically active source regions further south on our MSA



measurements. Our conclusions also agree with other studies that have also identified significant contributions of DMS to sulfate and MSA in the aerosol phase from biologically productive waters near the marginal ice zone and surrounding waters
(Leck and Persson, 1996b; Sharma et al., 2012; Becagli et al., 2016; Ghahreman et al., 2016; Ghahremaninezhad et al., 2017; Galí et al., 2021; Kurosaki et al., 2022), and our results further suggest that these regions are influential in MSA concentrations over the central Arctic as well.

To evaluate if transport from the regions in Fig. 7a is reasonable, we can consider the atmospheric lifetimes of MSA and DMS. Given the particle number size distributions in the central Arctic during MOSAiC (Boyer et al., 2023), we estimate the
lifetime of MSA against condensation onto particle surfaces to range from ~0.5 to 3 hours in our observations, using the simplifying assumption that MSA condenses irreversibly onto particle surfaces. This lifetime is too short to explain the effect of transport on our MSA measurements. The lifetime of gaseous DMS in the Arctic, however, is longer. The chemical conversion of DMS in the Arctic atmosphere is limited by the presence of available oxidants, and the resulting DMS lifetime is estimated to range from 1 – 5 days at latitudes < 70°N and 5 – 20 days > 80°N (Ghahremaninezhad et al., 2019). The
inverse model and FLEXPART simulations do not account for chemical processing, but from the chemical lifetime of DMS in the Arctic atmosphere, we can infer that MSA production from DMS occurs on timescales that are consistent with air mass transport into the central Arctic during summer, which is between ~5 – 15 days (Stohl, 2006). Therefore, we conclude that transport of DMS from the regions > 60°N in Fig. 7a, followed by subsequent chemical processing during transport, could explain our MSA measurements from the central Arctic during MOSAiC, which is also consistent with the convoluted
FLEXPART trajectories with the oceanic chl-a concentrations on timescales between 1 – 5 days.







**Figure 7: Source regions and their contributions to MSA from the inverse model.** (A) Potential source regions and their mean contribution to MSA between March – September, as identified by the inverse model, are shown in red. (B) A



timeseries of the simulated monthly median MSA concentrations from the source region polygons identified by the inverse
model. The MSA time series measured at the ship is included for context, and the shaded region shows the interquartile
range. Due to the limited domain of the FLEXPART simulations (> 60°N), source regions polygons "a" and "f" may
represent the contribution of MSA transport from regions further south than the polygons depicted on the map, such as the
oceanic regions on the western coast of North America and Bering Sea, respectively.

It is important to highlight the influence of transport from ocean regions south of the marginal ice zone on our observed
MSA concentrations. We cannot comment on the local sulfur emissions from biological activity below the sea ice, however,
there is mounting evidence that a stable meltwater layer forms on exposed leads within the sea ice during the melt season
(Rabe et al., 2022; Smith et al., 2023). The meltwater layer could act as a barrier that limits ocean-atmosphere gas exchange
(Nicolaus et al., 2022; Smith et al., 2023), hence limiting the release of DMS during the melt season in regions with sea ice
coverage. The effect of this meltwater layer is not explored further in this study, however, its occurrence and potential
implications for Arctic processes will be evaluated in future studies. There is also evidence to suggest that sea water contact
with sea ice limits biological processes associated with DMS production (Uhlig et al., 2019). Therefore, as the summertime
sea ice extent continues to decline, emissions of these biologically produced gases that are relevant for secondary aerosol
processes could become more prevalent in the central Arctic, leading to large scale changes in climate-relevant aerosol
processes. Indeed, long-term observations of gas-phase MSA concentrations in the Arctic show a strong positive association
with ambient air temperature (Moffett et al., 2020), and DMS emissions are increasing across the Arctic and are expected to
increase further under future scenarios with less sea ice coverage (Galí et al., 2019; Kurosaki et al., 2022).

While MSA emissions are expected to increase in the future, the subsequent changes in aerosol processes are not
straightforward. Based on our observations from MOSAiC, we note that periods of high MSA concentrations coincide with
the time of year when particles < 100 nm in diameter dominate the aerosol number size distribution (Boyer et al., 2023). Our
results agree with previous investigations that have found that enhanced NPF and subsequent Aitken mode sulfate aerosol
composition in the Arctic during summer can be directly linked to biogenic sulfur sources (i.e., MSA) (Leaitch et al., 2013;
Ghahreman et al., 2016; Willis et al., 2017; Abbatt et al., 2019) caused by retreating sea ice (Dall´Osto et al., 2017). On the
other hand, several other studies have observed a decrease in aerosol-phase MSA in the Arctic despite increases in DMS
emissions (Sharma et al., 2019; Moffett et al., 2020; Schmale et al., 2022), suggesting that higher gas-phase MSA
concentrations do not directly lead to increased MSA mass in particles.

The discrepancy among these observations may be due to the fate of MSA in aerosol processes in the marine atmosphere,
which are complex and not yet fully understood (e.g., Hodshire et al., 2019). MSA can condense onto aerosol surfaces to
grow nucleation and Aitken mode particles (Beck et al., 2021), however, the partitioning of MSA to the particle phase is
linked to aerosol acidity and relative humidity (Baccarini et al., 2021; Dada et al., 2022 and references therein), which varies
over time and space across the greater Arctic region (Fisher et al., 2011). Once in the aerosol phase, secondary particles
containing MSA are sufficiently hygroscopic such that they may enhance CCN concentrations in the often CCN-limited





conditions in the summertime Arctic atmosphere (Mauritsen et al., 2011). As a result, particles containing condensed MSA may be effectively removed by wet deposition during regional transport in the Arctic, which is supported by recent field and
modelling studies (Pernov et al., 2022; Mahmood et al., 2019). Conversely, multiphase chemistry on aerosol and cloud droplet surfaces has also been demonstrated as an important process in the formation of gas-phase MSA (Hoffmann et al., 2016; Baccarini et al., 2021). Moreover, the wet deposition processes that remove biogenic sulfur-containing aerosols may also result in periods of very low particle concentrations during summer in the Arctic, creating favorable conditions for NPF, a process that Pernov et al. (2022) reported to correspond with increased gas-phase MSA concentrations in their observations
from Greenland. These aerosol processes involving MSA likely vary temporally and spatially across the Arctic region, and therefore, it is necessary to continue monitoring the concentrations of MSA, in both the gas and aerosol phase, to resolve its contribution to the aerosol, CCN number concentrations, and ultimately the surface energy budget in the central Arctic as sea ice declines. Note that the seasonal analysis presented here is not sufficient to resolve the precise role of MSA in such processes, however, event-level analyses of the mechanism of NPF and the role of MSA during MOSAiC will be presented
in future studies.

### 3.3 Iodic acid: a halogen gas phase aerosol precursor

Figure 2a shows the annual record of IA concentrations during MOSAiC. Two significant peaks in IA concentration were observed during the year, the largest of which occurred in spring followed by a secondary peak in early autumn. Peak monthly median IA concentrations of $2.1\times10^6$ molec·cm$^{-3}$ and $3.7\times10^5$ molec·cm$^{-3}$ were measured in May and September,
respectively. In contrast, concentrations in late fall and winter were lower, with an average concentration of $9.7\times10^4$ molec·cm$^{-3}$ (SD = $1.3\times10^5$ molec·cm$^{-3}$) from October to February. Interestingly, there is also a low median concentration of IA during July ($1.1\times10^5$ molec·cm$^{-3}$), which is similar to the low concentrations observed during late autumn and winter. The seasonal cycle in IA concentrations correspond with the findings of Sharma et al. (2019) who observed two peaks in iodine constituents in the aerosol phase between March – May and August – September using a long time series of aerosol filter
measurements at Alert, Canada. Moreover, our IA concentrations agree well with the range of concentrations reported at both Villum Research Station, Greenland, Denmark and Ny-Ålesund, Svalbard, Norway, especially the peak observed during Spring (Sipilä et al., 2016; Beck et al., 2021). The IA concentrations measured during the autumn peak also agree with the findings of Baccarini et al. (2020) during a research cruise in the central Arctic in 2018; they observed monthly median IA concentrations of $8.6\times10^4$ molec·cm$^{-3}$ (SD = $1.2\times10^6$ molec·cm$^{-3}$) and $6.3\times10^5$ molec·cm$^{-3}$ (SD = $1.8\times10^6$
molec·cm$^{-3}$) during August and September, respectively (Baccarini et al., 2020).

The early spring peak of IA concentration, the largest during the year, coincides with increasing solar radiation that starts during March in the central Arctic (Fig. 2b). There are two possible explanations for this peak in spring, from biotic and abiotic processes, both of which are driven by the appearance of solar radiation. When solar radiation increases, it boosts the photolysis of molecular iodine as well as the biological activities of micro algae and phytoplankton, which are known
sources of iodine (O'Dowd et al., 2002; Ashu-Ayem et al., 2012; Allan et al., 2015). Interestingly, the increasing IA



concentrations occur during the same period of increasing MSA concentrations between March and May. As previously discussed in Section 3.2, increasing MSA concentrations during this time were associated with the onset of biological activity at high latitudes, which might suggest a link between IA and biological activity as well (Fig. 2c). However, the types of organisms that produce DMS (MSA) are different than those that produce iodine (IA), and the chemical mechanisms

governing the production of MSA or IA differ as well. Thus, the concurrent increase of MSA and IA during spring may be coincidental and more generally associated with the appearance of solar radiation during this time of year in the high Arctic. We must also consider abiotic processes that could contribute to the spring peak in IA. The introduction of sunlight in early spring initiates heterogenous photochemistry and emission of iodine compounds from sea salts deposited on sea ice/snow surfaces by blowing snow or the upward migration of brine through the sea ice (Domine et al., 2004; Raso et al., 2017;

Spolaor et al., 2019), which may proceed even with very low levels of light (He et al., 2021). Therefore, it is possible that the large increase of IA concentrations in March – May, when solar radiation returns to the Arctic, includes both biotic and abiotic processes. Sharma et al. (2019) also proposed that the iodine peaks in the aerosol constituents result from both biogenic activity and photochemical iodine processes. We are unable to resolve the contribution of these processes with the seasonal analysis presented herein.

Our observations show that IA concentrations are also strongly linked to seasonal changes in sea ice conditions. We observed that IA concentrations decrease during summer, possibly due to thinning of the sea ice and the reduced brine layer (Saiz-Lopez et al., 2015). It is likely that warming of the sea ice surface in the early season promotes the emission of iodine by restructuring the brine channel network and promoting the diffusion of iodine species to the surface through increases in the volume of brine veins, but only up to the point where the brine vein network breaks down due to more advanced stages

of melting resulting in unfavorable conditions for the emission of reactive iodine precursors (Saiz-Lopez et al., 2007). This process provides evidence that abiotic processes could be important during the spring peak in IA concentrations and may also explain the low IA concentrations observed between June and August, which corresponds to the annual peak in ambient air temperatures, global radiation, and sea ice melt. Additionally, it is important to consider the parallel competitive consumption of $O_3$ for $HIO_3$ formation (He et al., 2021; Finkenzeller et al., 2023), but also by other reactive halogens, such

as chlorine or bromine, that are well known actors of ozone depletion phenomena at the poles (Barrie et al., 1988; Pratt, 2019; Benavent et al., 2022). On the other hand, bromine emissions are most active in the upper layer of the snowpack that is thinning and disappearing well before the rest of the ice pack during the melting season (Custard et al., 2017).

In addition, we propose that the secondary peak in IA during the year, which occurred at the end of the summer melt season, is also associated with seasonal changes in sea ice processes and solar radiation. During this time of year, global radiation

and ambient air temperature decline in the central Arctic (Fig. 2b), and the sea ice undergoes various freeze/thaw cycles. These freeze/thaw cycles could cause restructuring of the brine channel networks in a similar way as in the spring. The freezing onset during MOSAiC occurred during late August and early September, which corresponds with the timing of the secondary peak in IA concentrations. Baccarini et al. (2020) also proposed that the increase of the IA concentration that they observed in the fall was linked to elevated $O_3$ concentrations combined with the formation of new sea ice during the freezing





onset observed in late August. Additionally, several other studies have shown that ozone enhances the emission of iodine from saline surfaces undergoing multiple freeze-thaw cycles during the autumn transition period (Halfacre et al., 2019; Abbatt et al., 2012; Carpenter et al., 2013). Moreover, the diurnal cycle of IA observed in Antarctica and over the Southern Ocean suggests that solar irradiance plays a role in atmospheric IA concentrations; higher IA concentrations were favored during periods of lower solar irradiance in the early morning and evening compared to mid-day (Jokinen et al., 2018;

Baccarini et al., 2021). We also observed that IA reaches its maximum concentrations during the months where sunlight returns to the Arctic and freeze/thaw cycles occur. Such a result suggests that emissions from the sea ice and ocean regions are influential on the IA concentrations during seasonal transitions in sea ice processes and under conditions with low solar radiation during the seasonal transitions from polar day/night, which further supports the results of these previous studies. However, our analysis focuses on the seasonal cycle, which is not sufficient to resolve the relative contributions from these

processes on IA concentrations. As such, atmospheric iodine processes, especially in the Arctic, require further investigation. A more detailed analysis of atmospheric IA formation mechanisms during the MOSAiC expedition will be given in a separate study.

IA has a demonstrated role in NPF (Allan et al., 2015; Sipilä et al., 2016). Baccarini et al. (2020) also identified that IA has an important role in NPF processes in the central Arctic, particularly during autumn when the sea ice refreezes. In contrast,

another study of the mechanism of NPF at two Arctic sites identified different particle formation pathways that were dependent on season and location. The study, conducted by Beck et al. (2021), showed that IA, SA, MSA, and ammonia were all identified to play different roles in the NPF process due to changes in the surrounding environment in Greenland and Svalbard. IA-induced NPF was found to be the most relevant pathway in Greenland during spring due to its proximity to sea ice, whereas NPF proceeded with the participation of SA, MSA, and ammonia during the summer months in Greenland

and Svalbard (Beck et al., 2021). During MOSAiC, the peaks in the IA concentration that we observed in spring and fall correspond with the results of both Baccarini et al. (2020) and Beck et al. (2021) that show IA NPF occurs during seasonal transitions in sea ice processes while near the sea ice. On the other hand, previous observations show that secondary particles in the nucleation and Aitken modes dominate the aerosol size distribution throughout summer (Tunved et al., 2013; Freud et al., 2017; Croft et al., 2016; Collins et al., 2017; Pernov et al., 2022; Boyer et al., 2023), even during the months where IA

concentrations are relatively low (e.g., July). This observation allows us to speculate that IA is not the only compound forming particles in the central Arctic during all seasons and that the chemistry of clusters and newly formed particles is more complex, as witnessed at land-based stations by Beck et al. (2021) (Schmale and Baccarini, 2021). The chemical mechanisms of NPF observed during MOSAiC will be evaluated in a dedicated study.



## 4 Conclusions

In this study, we present the annual cycles of MSA, SA, SO$_2$, and IA measured in the central Arctic during the MOSAiC expedition. These measurements represent the first continuous annual timeseries of these aerosol precursor vapors ever collected over the sea ice in the central Arctic Ocean, which offers new insights into the seasonal cycles of these vapors. Although the detailed sulfur chemistry is not explored herein, our results show the influence of both natural and anthropogenic sources on SA concentrations in the central Arctic. Most notably, we show that these sources yield the highest

SA concentrations in winter and spring, associated with Arctic haze and enhanced transport from continental sources further south. Comparatively, DMS oxidation from biogenic sources contributes less to SA concentrations during summer, as MSA production is likely favored in the Arctic region (Shen et al., 2022). FLEXPART simulations with emission inventories and inverse modeling results show that anthropogenic point-source emissions, especially from the region of Norilsk in Northern Russia, contribute substantially to the SA concentrations during winter. The short-lived peaks in SA concentrations and SO$_2$

mixing ratios during the Arctic haze period further confirm the anthropogenic origin of the high SA concentrations from point source emissions in Siberia/Northern Russia. These observations also agree with enhanced pollution transport due to the positive phase of the AO during the MOSAiC year (Lawrence et al., 2020). Natural sources of atmospheric sulfur, including the volcanically active regions of Aleutian Peninsula in Alaska and the Kamchatka Peninsula in Russia, also contribute to high SA concentrations during spring when transport of air masses from continental regions remains favorable

and solar radiation increases in the Arctic region. Processes controlling SA concentrations are subject to change as the Arctic becomes more accessible as sea ice continues to decline and anthropogenic activities, such as ship traffic, become more common in the central Arctic throughout the year (Ferrero et al., 2016).

Our analyses additionally show that biological activity in the open ocean areas near the marginal ice zone within the Arctic region contributes to enhanced MSA concentrations, an important component of aerosol formation and growth, during late

spring through summer in the central Arctic. The timing of the annual maximum in MSA corresponds to elevated chl-a concentrations north of 75 °N during May and June. In the current Arctic environment, transport from the marginal ice zone and regions further south appears to be the primary driver of MSA concentrations in the central Arctic over the sea ice. Inverse model results suggest that influential regions include the Labrador, Norwegian, Barents, and Kara Seas. We do not expect that biological activity in the surface ocean within the sea ice pack is a strong source of MSA concentrations during

the summer, but this may be subject to change in the future as sea continues to decline. MSA concentrations may increase as a result. The peak in MSA concentrations is also consistent with previous observations of enhanced nucleation and Aitken mode particle concentrations in the aerosol size distributions (e.g., Leaitch et al., 2013).

We observed peaks in atmospheric IA concentrations during seasonal transitions in spring and fall. IA has an apparent source from sea ice thawing/freezing processes during periods with low solar irradiance, which is in concordance with

previous observations of IA. As there are iodine compounds sourced from marine biological activities, the spring IA peak could be partly explained by the increased production of algae (from both the sea and the ice) with the appearance of





sunlight. In addition, the thinning sea ice, and hence increased brine channel network, could facilitate the exchange of iodine into the atmosphere and further reaction with $O_3$ to form IA. Once solar radiation intensity increases and sea ice experiences more advanced stages of melting, however, the concentrations of IA decline. Therefore, we suggest that the peak IA

concentrations may have biotic and abiotic origins that are strongly linked to low levels of solar radiation, however future work is necessary to resolve the details of these processes.

Our observations suggest that the current seasonal cycles of SA, MSA, IA, and $SO_2$ in the central Arctic Ocean are linked to sea ice conditions and solar radiation due to their role in biological activity and air mass transport from southern regions. Given that sea ice is in a state of decline in the central Arctic, the concentrations of the vapors presented herein, and their

influence on aerosol processes, will likely change as a result. Anthropogenic activities around the Arctic and in the central Arctic Ocean may increase as the Arctic becomes more accessible with less sea ice, which can also influence these gas phase species. The magnitude and implications of such changes on climate relevant processes remain uncertain. While our findings offer new insights that can improve climate model predictions in the remote Arctic, it is imperative to continue monitoring these aerosol precursor vapors and to further understand their role in atmospheric processes to evaluate their climate-relevant

effects on aerosol formation, growth, and subsequent CCN activation in the future in the central Arctic.

**Data availability**

All datasets used in this work that were obtained during the MOSAiC campaign are publicly available via Pangaea ([https://www.pangaea.de/](https://www.pangaea.de/)).

Data from the Pangaea archive includes:

- Meteorological observations from *Polarstern*:

    o Schmithüsen, Holger (2021): Continuous meteorological surface measurement during POLARSTERN cruise PS122/1. Alfred Wegener Institute, Helmholtz Centre for Polar and Marine Research, Bremerhaven,
PANGAEA, [https://doi.org/10.1594/PANGAEA.935221](https://doi.org/10.1594/PANGAEA.935221)

    o Schmithüsen, Holger (2021): Continuous meteorological surface measurement during POLARSTERN cruise PS122/2. Alfred Wegener Institute, Helmholtz Centre for Polar and Marine Research, Bremerhaven, PANGAEA, [https://doi.org/10.1594/PANGAEA.935222](https://doi.org/10.1594/PANGAEA.935222)


    o Schmithüsen, Holger (2021): Continuous meteorological surface measurement during POLARSTERN cruise PS122/3. Alfred Wegener Institute, Helmholtz Centre for Polar and Marine Research, Bremerhaven, PANGAEA, [https://doi.org/10.1594/PANGAEA.935223](https://doi.org/10.1594/PANGAEA.935223)





680       o Schmithüsen, Holger (2021): Continuous meteorological surface measurement during POLARSTERN cruise PS122/4. Alfred Wegener Institute, Helmholtz Centre for Polar and Marine Research, Bremerhaven, PANGAEA, https://doi.org/10.1594/PANGAEA.935224

685       o Schmithüsen, Holger (2021): Continuous meteorological surface measurement during POLARSTERN cruise PS122/5. Alfred Wegener Institute, Helmholtz Centre for Polar and Marine Research, Bremerhaven, PANGAEA, https://doi.org/10.1594/PANGAEA.935225

- Chemical Ionization Mass Spectrometer ($NO_3$-CIMS):

690       o Boyer, Matthew; Quéléver, Lauriane; Beck, Ivo; Laurila, Tiia; Sarnela, Nina; Schmale, Julia; Jokinen, Tuija (publication year): Ambient concentrations of aerosol precursor vapor concentrations (sulfuric acid, methanesulfonic acid, and iodic acid) in 5-minute resolution measured by a nitrate chemical ionization mass spectrometer. PANGAEA, https://doi.org/10.1594/PANGAEA.963321

- Sulfur dioxide (SO2):

       o Angot, Hélène; Beck, Ivo; Jokinen, Tuija; Laurila, Tiia; Quéléver, Lauriane; Schmale, Julia (2022): Ambient air sulfur dioxide mole fractions measured in the Swiss container during MOSAiC 2019/2020. PANGAEA, https://doi.pangaea.de/10.1594/PANGAEA.944270


- Particle number concentration (CPC3025):

       o Beck, Ivo; Quéléver, Lauriane; Laurila, Tiia; Jokinen, Tuija; Schmale, Julia (2022): Continuous corrected particle number concentration data in 10 sec resolution, measured in the Swiss aerosol container during
705       MOSAiC 2019/2020. PANGAEA, https://doi.org/10.1594/PANGAEA.941886

The Ocean Colour Climate Change Initiative dataset, Version 5.0, European Space Agency, is publicly available online at http://www.esa-oceancolour-cci.org/:

- Sathyendranath, S., Jackson, T., Brockmann, C., Brotas, V., Calton, B., Chuprin, A., Clements, O.,
Cipollini, P., Danne, O., Dingle, J., Donlon, C., Grant, M., Groom, S., Krasemann, H., Lavender, S., Mazeran, C., Mélin, F., Müller, D., Steinmetz, F., Valente, A., Zühlke, M., Feldman, G., Franz, B., Frouin, R., Werdell, J., Platt, T.: ESA Ocean Colour Climate Change Initiative





(Ocean_Colour_cci): Version 5.0 Data, NERC EDS Centre for Environmental Data Analysis, 19 May 2021. http://dx.doi.org/10.5285/1dbe7a109c0244aaad713e078fd3059a, 2021.


An archive of the FLEXPART model output and quick looks for the whole campaign can be found at https://img.univie.ac.at/webdata/mosaic.

**Author contributions**

MB led the analysis and writing process of the manuscript with the assistance of LQ and TJ.

LQ, IB, JS, TJ, and TL conducted field measurements reported in this work.

SB, MD, and AS performed the FLEXPART simulations.

DA performed inversion modelling and source region identification work.

HA, LD, IB, MDO, RT, LB, NS, BM, and JS provided useful insights and assisted with the interpretation of various datasets.

MK, TP, JS, and MS provided funding for the campaign and data analysis.

All authors provided comments/revisions on the manuscript.

**Competing interests**

At least one of the (co-)authors is a member of the editorial board of Atmospheric Chemistry and Physics. The authors also have no other competing interests to declare.


**Acknowledgments**

Data reported in this manuscript were produced as part of the international Multidisciplinary drifting Observatory for the Study of Arctic Climate (MOSAiC) expedition with the tag MOSAiC20192020, with activities supported by *Polarstern* expedition AWI_PS122_00. We acknowledge funding from the Swiss National Sciences Foundation grant No. 188478 and

the Swiss Polar Institute (grant no. DIRCR-2018-004), the US DOE grant No. DE-SC0022046, European Union's Horizon 2020 research and innovation programme under grant agreement No. 856612 and the Cyprus Government. JS holds the Ingvar Kamprad Chair, sponsored by Ferring Pharmaceuticals. Part of this project was funded by ERC grant (GASPARCON, 714621) and the Academy of Finland funding (grant No. 337552, 296628, 328290, 346372, 1346372, and 335844). Data analysis was partly funded by the European Union ERC-2022-STGERC-BAE-Project: 101076311. This

project has received funding from the European Union's Horizon 2020 research and innovation program under grant agreement No 101003826 via project CRiceS (Climate Relevant interactions and feedbacks: the key role of sea ice and Snow in the polar and global climate system). Views and opinions expressed are however those of the author(s) only and do not necessarily reflect those of the European Union or the European Research Council Executive Agency. Neither the European Union nor the granting authority can be held responsible for them. The authors thank the Laboratory of Atmospheric

Chemistry at the Paul Scherrer Institute and ACTRIS CiGas-UHEL for their support, and we thank Janne Lampilahti, Markus Lampimäki, Tommy Chan, Katrianne Lehtipalo, Federico Bianchi, Anton Rusanen, Heikki Junninen, and the INAR



technical staff for their assistance with measurement support during the campaign. We also extend a special thank you to all personnel who made the expedition possible through the operation of the R/V *Polarstern* during MOSAiC in 2019–2020 (AWI_PS122_00) (Nixdorf et al., 2021).

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
