# Peer review of "The annual cycle and sources of relevant aerosol precursor vapors in the central Arctic during the MOSAiC expedition"

_EGUsphere, 2023_

## Referee Comment (RC1)

Comments to the manuscript egusphere-2023-2953

**Boyer et al. The annual cycle and sources of relevant aerosol precursor vapors in the central Arctic.**

**Reviewer's comments**

The article presents and analyzes the first time series of aerosol precursor vapors (sulfuric acid (SA), methanesulfonic acid (MSA), and iodic acid (IA)) that are relevant to the central Arctic during MOSAiC. The authors conducted the measurements with state-of-the-art instruments, and the results are crucial in assessing the impact of anthropogenic emissions in the Arctic. The article is well structured, and well written, and only requires minor corrections before acceptance.

**Minor comments**

- The article discusses the crucial findings related to the concentration of atmospheric gases in the Arctic. However, the information is not presented in a concise format such as a table, which would enable the reader to easily examine the data. It is recommended to include a table that displays the monthly and seasonal averages of the primary anthropogenic pollutants.
- The date in Figure 4 is confusing. It is recommended to improve the date and use a single format.
- Figure 1 presents the gas phase time series for SA, MSA, and IA during clean and polluted periods using a violin-type scheme. This type of schematic can be a bit confusing. It is recommended to present the information in another format, for example, box-and-whisker diagrams. Could you expand the description of these results? Perhaps showing a table with the main statistical results.
- Figure 2c shows several results that are difficult to interpret. Is it possible that you could improve or present these results in another format?
- The message describing Figure 2c mentions the dashed white line denoting the monthly median latitude of Polarstern during the campaign. However, in Figure 2c no dotted line is shown. The only one observed is a continuous white line.

---

## Author Response (AR1)

Authors' response to Reviewer #1

Reviewer comments in black, Author responses in Red

Please note that all line number references correspond to the line numbers in the original preprint.

The article presents and analyzes the first time series of aerosol precursor vapors (sulfuric acid (SA), methanesulfonic acid (MSA), and iodic acid (IA)) that are relevant to the central Arctic during MOSAiC. The authors conducted the measurements with state-of-the-art instruments, and the results are crucial in assessing the impact of anthropogenic emissions in the Arctic. The article is well structured, and well written, and only requires minor corrections before acceptance.

We thank the reviewer for their positive feedback and constructive comments on the manuscript.

Minor comments

- The article discusses the crucial findings related to the concentration of atmospheric gases in the Arctic. However, the information is not presented in a concise format such as a table, which would enable the reader to easily examine the data. It is recommended to include a table that displays the monthly and seasonal averages of the primary anthropogenic pollutants.

  Thank you for this suggestion. A table could be helpful to present the data in a simplified format that is accessible for the readers. We have added a table to the SI with the monthly statistics for the aerosol precursor vapors presented in this work (SA, MSA, IA) that includes the monthly median concentrations, the interquartile ranges, the percentage of data identified as pollution, and the minimum and maximum values. This table is as follows:

  "**Table S1. Monthly statistics for the aerosol precursor vapors measured during MOSAiC.** The statistics include the monthly median, 25th and 75th percentiles, and the maximum values for SA, MSA, and IA. The monthly percentage of data influenced by local pollution sources is also included. For readability, all units in concentration (molec·cm$^{-3}$), are scaled by 1e+05, except for pollution, which is given as a percentage.

| Time | SA | | | | MSA | | | | IA | | | | Percentage pollution |
|---|---|---|---|---|---|---|---|---|---|---|---|---|---|
| | Median | 25th percentile | 75th percentile | Maximum | Median | 25th percentile | 75th percentile | Maximum | Median | 25th percentile | 75th percentile | Maximum | |
| October 2019 | 0.84 | 0.6 | 1.93 | 34.9 | 3.4 | 1.95 | 4.58 | 11.5 | 1.11 | 0.51 | 2.22 | 16.7 | 75 |
| November 2019 | 1.94 | 0.77 | 5.38 | 128 | 2.71 | 1.5 | 4.09 | 37.4 | 0.54 | 0.32 | 0.96 | 13.5 | 56 |
| December 2019 | 2.26 | 1.31 | 5.13 | 674 | 2.48 | 1.96 | 3.2 | 159 | 0.8 | 0.5 | 1.17 | 6.85 | 38 |
| January 2020 | 9.47 | 6.48 | 17.4 | 157 | 2.59 | 2.06 | 3.26 | 19.7 | 0.44 | 0.31 | 0.6 | 3.51 | 47 |
| February 2020 | 8.61 | 4.95 | 13.5 | 98.9 | 3.11 | 2.37 | 3.9 | 22.6 | 0.36 | 0.28 | 0.49 | 4.37 | 34 |
| March 2020 | 13.6 | 9.42 | 20.9 | 143 | 6.88 | 5.23 | 10.4 | 75.2 | 3.27 | 1.46 | 9.68 | 58.8 | 73 |
| April 2020 | 6.95 | 4.18 | 14.2 | 164 | 14.9 | 8.15 | 34.6 | 294 | 11.4 | 4.4 | 30.4 | 373 | 77 |
| May 2020 | 4.08 | 1.75 | 11.2 | 82 | 32.1 | 16.3 | 52.7 | 712 | 20 | 3.95 | 48 | 305 | 79 |
| June 2020 | 3.29 | 1.49 | 8.25 | 71.2 | 15.4 | 2.35 | 42.9 | 306 | 2.04 | 0.62 | 4.49 | 22 | 86 |
| July 2020 | 1.89 | 0.98 | 3.88 | 29.3 | 19.5 | 7.12 | 45.8 | 164 | 0.39 | 0.28 | 0.79 | 11.5 | 80 |
| August 2020 | 1.18 | 0.69 | 2.24 | 30.1 | 9.51 | 3.46 | 22.4 | 158 | 0.68 | 0.36 | 2.01 | 41.1 | 69 |
| September 2020 | 1.93 | 0.97 | 4.15 | 96.1 | 3.47 | 1.65 | 7.55 | 82.1 | 2.63 | 0.9 | 7.55 | 188 | 66 |

"

We would like to highlight that the dataset used in this study is publicly available, with the data citation given in the "Data availability" section on line 690, if readers wish to explore the data further.

- The date in Figure 4 is confusing. It is recommended to improve the date and use a single format.
  Thank you for this helpful suggestion. The date format in Figure 4B (now Figure 3B) has been updated in the text as follows:

[Figure]

- Figure 1 presents the gas phase time series for SA, MSA, and IA during clean and polluted periods using a violin-type scheme. This type of schematic can be a bit confusing. It is recommended to present the information in another format, for example, box-and-whisker

diagrams. Could you expand the description of these results? Perhaps showing a table with the main statistical results.

We have considered this comment but have chosen to keep the violin plot in Figure 1 (now Figure S2). The violin plot features the same information as a box-and-whisker diagram, including the median values, interquartile ranges, and extremes of the data. However, a violin plot also shows the distribution of the data. Given that the datasets presented in Figure 1 (now Figure S2) are very large (~90,000 data points), we think that the violin plot gives a better representation of the data by also providing information about the data distribution.

Furthermore, with the violin plots, we are also able to directly compare the shape of the data distribution between the clean and polluted periods. This comparison is relevant for the interpretation of the possible pollution influence in the data, as described in the main text. As an additional note, this method has been used previously to evaluate gas phase data between clean and polluted periods from ship-based measurements (Baccarini et al., 2021).

Additional figures showing the timeseries of the clean/polluted data are given in Figs S3, S4, S5, and S6 in the supplemental information for SA, MSA, and IA, respectively. These figures complement Figure 1 (now Figure S2) by showing information about the timing of pollution that is not captured by the violin plots. For further clarification of the timing and extent of pollution, we have also included the percentage of data identified as pollution in the monthly data table (Table S1). Likewise, Figs S4, S5, and S6 show the comparison of the monthly median gas phase concentrations and the corresponding interquartile ranges between clean and polluted periods. The monthly comparisons clearly show that the data are very similar for the clean and polluted periods, and hence our interpretation of the seasonal cycles are largely unaffected by including the polluted data.

- Figure 2c shows several results that are difficult to interpret. Is it possible that you could improve or present these results in another format?

It is true that Figure 2C (now Figure 1C) includes data that is described in a later section of the manuscript, which was not adequately described in the caption, namely the sea surface chl-a index. We have presented the data in this way to so that the chl-a data can be compared to the timeseries of each species (SA, MSA, and IA) in one figure to avoid redundancy. To help clarify the data presented in Figure 2C (now Figure 1C), we have added descriptive text to the figure caption and a reference the section where the sea surface chl-a index is explained in further detail. The description of Figure 2C (now Figure 1C) is updated as follows:

"(C) The chl-a concentrations are shown as 8-day averages integrated for every 5° of latitude between 60 – 90°N from the OC-CCI satellite dataset. The color bar depicts the chl-a concentrations, where the lack of color indicates missing data due to cloudiness, algorithm failure, or low-incidence sun angles in winter. The solid black line denotes the monthly median latitude of *Polarstern* during the campaign. The orange circles show the sea surface chl-a influence index, a timeseries that quantifies potential air mass exposure to ocean regions with chl-a derived from the convolution of the FLEXPART FES with the satellite chl-a concentrations using air mass ages of 10 days, as described in section 3.2."

- The message describing Figure 2c mentions the dashed white line denoting the monthly median latitude of Polarstern during the campaign. However, in Figure 2c no dotted line is shown. The only one observed is a continuous white line.

Thank you for finding this error. In addition, according to our response to other reviewer comments, the line indicating the ship's median latitude was now changed to black. We have corrected the figure caption on line 280 as follows:

"The solid black line denotes…"

References

Baccarini, A., Dommen, J., Lehtipalo, K., Henning, S., Modini, R. L., Gysel-Beer, M., Baltensperger, U., and Schmale, J.: Low-Volatility Vapors and New Particle Formation Over the Southern Ocean During the Antarctic Circumnavigation Expedition, J. Geophys. Res. Atmospheres, 126, https://doi.org/10.1029/2021JD035126, 2021.

Authors' response to Reviewer #2

Reviewer comments in black, Author responses in Red

Boyer et al. describe one year of relevant aerosol precursor in the gas phase from the central Arctic during the MOSAiC expedition from Sep 2019 to Oct 2020. The data include sulfuric acid (SA), methanesulfonic acid (MSA), and iodic acid (IA). They used the FLEXPART simulations, inverse modeling, and satellite chlorophyll-a (chl-a) to interpret the seasonal variability of the studied parameters and identify their source regions. The results conclude that natural and anthropogenic sources contribute to different aerosol types depending on the season, biotic and abiotic factors, and air mass movement. The information provided and the study's conclusions could be helpful to the scientific community. The manuscript does, however, require more explanation in some sections. The results visualizations, as I explain below, are my major concern.

Thank you for your general comments and constructive feedback on this work. We have implemented your suggestions accordingly, which have improved the clarity of our results, as described below in our responses to your specific comments.

General comments:

Section 2.6, Line 207: The FLEXPART and inverse modeling need further explanation on how they work. For instance, what are the frequency, timing and length of back-trajectories used? Where do the main sources of air mass dominate during the measurement period? A figure showing the air mass back-trajectories in the supplement would be beneficial. While ECLIPSE is a fundamental tool in presenting the results, no details on how It works, references and previous applications.

Thank you for this suggestion. You are correct, additional details about the length and frequency of the FLEXPART trajectories will help clarify our analysis. The description of the FLEXPART simulations on line 204 now reads as follows:

"To infer the air masses origin, we use the lagrangian particle model FLEXPART v10.4 (Pisso et al., 2019) driven by ERA5 reanalysis data. Backward simulations, up to 30 days backward in time, were calculated releasing a cluster of 100 000 air tracer particles every 3 hours from Polarstern's location during the MOSAiC expedition. These simulations were then used to determine the "Footprint Emission Sensitivity" (FES), or the emission sensitivity of air masses < 100 m altitude during transport, according to the simulated "source-receptor relationship" (SRR) (Seibert and Frank, 2004). Refer to Fig. S7, taken from Boyer et al. (2023), for the FES observed during the MOSAiC expedition.."

In the text, we previously referred to Fig. 2 in Boyer et al. (2023) to show the dominant sources of air mass trajectories from the FES. It is a helpful suggestion to also include this figure in the supplement. Therefore, we added the following figure, Fig. S7, to the supplement (taken from Boyer et al., 2023):

[Figure]

**Figure S7. The seasonality of surface influence from air mass source regions using FLEXPART.** A geographical mask (a) was applied to the FLEXPART air tracer data to quantify the FES associated with each geographical source region (b). The FES was determined from the FLEXPART air tracer data within the lowest 100 meters of the atmosphere and was used to identify the influence of different source regions on the observed aerosol throughout the year. Taken from Boyer et al. (2023).

You are correct that additional details could be provided for the description of the ECLIPSE emission inventory. The description has been expanded on line 211 as follows:

"The FES can then be coupled with an emission inventory to simulate concentrations at the ship, which is an established method to evaluate source regions using to the SRR (e.g., Sauvage et al., 2017). Specifically, we used the ECLIPSE v6b (Evaluating the Climate and Air Quality Impacts of Short-Lived Pollutants) emission inventory to estimate source regions of anthropogenic sulfur, from $SO_2$ emissions, as described in section 3.1. For Additional details on $SO_2$ in the emission inventory, we refer to Klimont et al. (2013)."

The conclusion section is long and has discussion sentences/cited references that might be moved to Results and Discussions. The Section shall summarize the major findings along with their implications.

Thank you for this constructive feedback. The conclusion section has been revised, particularly with respect to the lengthy summaries of the SA and MSA results. The summaries have been shortened considerably by reducing redundant statements and citations that are also included in the discussion sections.

The first three paragraphs of the conclusions, starting on line 615, are now written as follows:

"In this study, we present the annual cycles of SA, MSA, and IA measured in the central Arctic during the MOSAiC expedition. These measurements represent the first continuous annual timeseries of these aerosol precursor vapors ever collected over the sea ice in the central Arctic Ocean, which offers new insights into their seasonal cycles.

Our results show the influence of both natural and anthropogenic sources on SA concentrations in the central Arctic. Most notably, we show that these sources yield the highest SA concentrations in winter and spring, associated with Arctic haze and enhanced transport from continental sources further south. Comparatively, DMS oxidation from biogenic sources contributes less to SA concentrations during summer. Anthropogenic point-source emissions in Siberia/Northern Russia, especially from the region of Norilsk in Northern Russia, contribute substantially to the SA concentrations during winter. Natural sources of atmospheric sulfur, including volcanically active regions, also contribute to high SA concentrations during spring when transport of air masses from continental regions remains favorable and solar radiation increases in the Arctic region. Processes controlling SA concentrations are subject to change as the Arctic becomes more accessible as sea ice continues to decline and anthropogenic activities, such as ship traffic, might become more common in the central Arctic throughout the year.

Our analyses additionally show that biological activity in the open ocean areas near the marginal ice zone within the Arctic region contributes to enhanced MSA concentrations, an important component of aerosol formation and growth, during late spring through summer in the central Arctic. The timing of the annual maximum in MSA corresponds to elevated chl-a concentrations north of 75 °N during May and June. In the current Arctic environment, transport from the marginal ice zone and regions further south appears to be the primary driver of MSA concentrations in the central Arctic over the sea ice. We do not expect that biological activity in the surface ocean within the sea ice pack is a strong source of MSA concentrations during the summer, but this may be subject to change in the future as sea ice continues to decline. MSA concentrations may increase as a result."

Specific comments:

Line 94: Between brackets, define the HOx, for example, (OH and H2O).

You are correct. We have added the definition of HOx, "(OH and $HO_2$)", to the text on line 94 as suggested.

Line 97: At least define what IxOy refers to at the 1st mention.

Thank you for pointing this out. According to your suggestion, the text at line 97 is changed as follows:

"...or from the hydrolysis of intermediate iodine oxide compounds (e.g., $I_2O_3$, $I_2O_4$, $I_2O_5$)."

Figure 1: The data is misleading and the word "time series" in the caption is not appropriate. From the 1st glance, the readers may interpret that the colors distinguish between the input from biogenic (clean) and anthropogenic (polluted) sources. While I see the polluted cases represent the excess concentration to the biogenic one (polluted + biogenic). Indeed, I think this figure after a suitable explanation could be transferred to the supplementary material since it doesn't contribute to the main conclusion of the study.

The reviewer is correct. We have updated "timeseries" to "concentrations" in the title of Figure 1 (now Figure S2).

To clarify that the figure is referring to local pollution and not anthropogenic emission associated with Arctic Haze, the figure title was changed to "The influence of local pollution on the gas phase concentrations of SA, MSA, and IA." In addition, we updated the figure legend and figure caption to clarify that "polluted" refers to local pollution from the ship. In addition, the figure was moved to the supplemental information, as suggested by the reviewer. Here is the updated figure, now located in the supplemental information:

[Figure]

**"Figure S2. The influence of local pollution on the gas phase concentrations of SA, MSA, and IA.** The violins show the frequency distribution of the gas phase concentration for the clean (blue) and polluted (red) periods to evaluate the influence of local pollution, from the ship and logistical activities, on each species during the campaign. The medians and interquartile ranges of each data subset are represented by the solid and dashed black lines, respectively."

Line 178: A statistical test would be more robust to evaluate that there is no significant difference in medians.

Thank you for this suggestion. We have tried using statistical tests on the concentration data between the "clean" and "polluted" periods. Interestingly, the tests suggest that there is a significant difference in the medians between the two groups (Mann-Whitney U test, p<0.01 for SA, MSA, and IA). Given the large number of data points in the two groups (~25 000 "clean" and ~50 000 "polluted" after removing data below the limit of detection), the medians presented in Figure 1 (now Figure S2) are already quite robust, and therefore, statistically significant differences between the two groups are not surprising.

Despite these differences, we have concluded that the variations between the clean and polluted periods is coincidental by interpreting other aspects of the data, as explained in the text in section 2.3. First, the differences between the clean and polluted data occur for all three vapors species. While we understand how local pollution could affect SA, we are not aware of any mechanisms where local pollution could affect the concentrations of MSA and IA, suggesting that the higher concentrations of each species during polluted periods could be coincidental due to temporal changes in concentrations. Second, the shapes of the distributions, as shown in Figure 1 (now Figure S2), are quite similar for each vapor between clean and polluted periods, even at the lower concentration ranges. If there were a systematic effect of pollution on the concentrations, we would expect the shapes of the data distribution to show more variability. The most notable exception in distribution shape occurs in the SA concentrations > $1e^7$ molec·cm$^{-3}$ in the "polluted" group, however, our analysis suggests that these high SA concentrations are associated with episodic transport of anthropogenic pollution during winter, as described in section 3.1.

Furthermore, we concluded the seasonal analysis in this work remains unaffected by the inclusion of the "polluted" data. Figures S4, S5, and S6 in the supplement show the concentration timeseries between the "clean" and "polluted" periods for SA, MSA, and IA, respectively. Panel B in these figures directly compares the monthly median data between "clean" and "polluted" periods, where the seasonal cycle remains consistent between the two groups. This result demonstrates that our conclusions on the seasonal cycles are not biased by the "polluted" data.

We adjusted the wording in the text on line 171, where we called the effect of local pollution "negligible." Line 171 now reads as follows:

"Overall, the comparisons presented in Figs. S2, S3, S4, S5, and S6 suggest that local pollution does not affect our analysis of the seasonal cycles for all three aerosol precursor vapors during the MOSAiC expedition."

Line 200: Not only cloudiness but also the low-incidence sun angle in winter hinders the measurement of chl-a from satellites at high latitudes.

This is a helpful comment. We have included this information in the text accordingly on line 201:

"Cloudiness, algorithm failure, or low-incidence sun angles in winter can result in missing data in the Arctic, however, overall, there are typically enough cloud-free data available to evaluate chl-a in the Arctic region during summer (e.g., Becagli et al., 2016)."

Line 270: do you think the word "diurnal" fits the explanation here? Diurnal means 24-hour variations from day to night.

For the reason mentioned, we put the term "diurnal" in quotes and then further described it as the "cycle between polar day and polar night." It's true that this is not a traditional diurnal cycle in the 24-hour sense, but the change in global radiation during the year in the central Arctic (Fig. 2b) (now Figure 1b) can be likened to one long transition between polar day and polar night. To avoid confusion for the readers, we removed the word "diurnal" and added a reference to the global radiation in Fig. 2b (now Figure 1b). The sentence now reads as follows on line 270:

"Given the high latitude of these measurements and the changes in global radiation during the year (Fig. 1b), the seasonal cycle that we present is essentially a cycle between polar day and polar night."

Figure 2: I don't see the advantage of adding the 5-minute time resolution data in the background. The figure presents the annual cycle, so the monthly median with the shaded area or monthly box charts that describe the whole statistics is enough. Panel-A: the left y-axis is hidden. Panel C: It is better to present the missing data in chl-a as blank (white) to differentiate low values and non-measured times. Accordingly, the dotted white line must be modified.

Initially, we chose to include the monthly median data as well as the higher time resolution data to show the variability and extremes for each species during the year, which is a more descriptive presentation of the data overall. The high time resolution data are also represented in figures S4, S5, and S6 in the supplemental information, however, so we have taken the reviewers suggestion to remove the 5-minute time resolution data from Figure 2 (now Figure 1) in the main text.

Thank you for your comment on the missing chl-a data. You are correct that it better to represent the missing data in white, so we have changed the figure 2c (now figure 1c) accordingly, including the description of the chl-a data in the figure caption. The ship's latitude is now denoted by a solid black line to remain visible after making the missing data white.

Taking these changes into account, Figure 2 (now Figure 1) is now as follows:

[Figure]

**Figure 1: The annual cycle of aerosol precursor vapor concentrations and related parameters during MOSAiC**. (A) The monthly median vapor concentrations and the corresponding interquartile ranges of iodic acid (IA), methanesulfonic acid (MSA), and sulfuric acid (SA) are presented as the solid

lines and shaded regions, respectively. The concentration data at a 5-minute time resolution are also shown (thin lines). Note that concentrations below the LODs for each vapor species were removed. (B) The temperature and global radiation measurements are shown at 1-minute resolution with the monthly averages and interquartile ranges overlayed for context. (C) The chl-a concentrations are shown as 8-day averages integrated for every 5° of latitude between 60 – 90°N from the OC-CCI satellite dataset. The color bar depicts the chl-a concentrations, where the lack of color indicates missing data due to cloudiness, algorithm failure, or low-incidence sun angles in winter. The solid black line denotes the monthly median latitude of *Polarstern* during the campaign. The orange circles show the sea surface chl-a influence index, a timeseries that quantifies potential air mass exposure to ocean regions with chl-a derived from the convolution of the FLEXPART FES with the satellite chl-a concentrations using air mass ages of 10 days, as described in section 3.2.

Figure 3: Same comment as Figure 2. Presenting a time average (median) may be better to highlight the main variations throughout the time series.

In Figure 3 (now Figure 2), we specifically chose to present the timeseries at a higher time resolution to show the consistency of short-lived spikes for all three sulfur species during winter, as we described on line 332:

"Overall, the SO4-S simulations agree well with the measured gas phase sulfur species, especially in January and February when we observed temporal spikes of each species (Fig. 2)."

If we presented the data as a longer time average, Figure 3 (now Figure 2) would lose the information concerning these spikes, which is pertinent to our analysis. Additionally, the low amount of $SO_2$ data above the detection limit restricts our ability to present the data as a meaningful time average. Therefore, we feel that it is best to keep the high time resolution in Figure 3 (now Figure 2).

Figure 4: The presentation of the data is hard to interpret. For example, the oceanic contribution in July is about 0.1 µg/m3 or (0.18 – 0.1 µg/m3). Indeed, I see Fig. S8 is worth presenting in the main manuscript rather that Fig. 4.

While we acknowledge that is difficult to interpret the calculated values from several of the source regions in Figure 4 (now Figure 3), the key point of the figure is to show the timeseries of influence from geographic regions that were most relevant on our measured sulfur compounds during a given time of year, especially the pronounced influence from the North Asia region during winter. To do this, Figure 4 shows the coupling of the FLEXPART simulations with the emission inventory. Conversely, Figure S8 simply shows the emission characteristics from the emission inventory according to the source regions included in the source region mask (Fig. 4a) (now Fig. 3a), which is presented as the annual average of anthropogenic emissions from these regions. Therefore, Figure S8 (now Figure S11) is only a description of the emission inventory data. Figure S8 (now Figure S11) supplements Figure 4 (now Figure 3), but it completely lacks information related to the timing of air mass influence from these source regions on our measurements that are relevant for our analysis. As such, we conclude that Figure 4 (now Figure 3) should remain in the main text as is, and Figure S8 (now Figure S11) should remain in the supplement.

Figures 5, 6, and 7: In my view, the colors show the contribution of each polygon, which, when added together, gives the concentration overall.

Thank you for your comment. In panel A of Figures 5, 6, and 7 (now figures 4, 5, and 6), the color bar shows the annual mean of the source regions, as identified by the inverse model. This is presented in a similar way as the source contribution maps from the ECLIPSE emission inventory in Figure S8 (now Figure S11). The annual mean contribution was used in this figure to show the overall contribution from the different regions during the year on a single figure. Otherwise, as the source contributions change temporally, different maps would be required for each month. The temporal changes are presented in Panel B, which does indeed show the contribution of each polygon and their collective simulated concentration as it changes during the year, as suggested by the reviewer.

The description of the annual mean source region contribution and the polygons varied slightly between the captions in Figures 5, 6, and 7 (now figures 4, 5, and 6). To make it more clear to the reader, we change the caption of figures 5 (4) and 7 (6) to be more consistent with the figure 6 (now figure 5), which gives the clearest description. The figure captions for Figures 5 & 7 (now figures 4 & 6) now read as follows:

> "Figure 4: Inverse model results for SO2. (A) The potential source regions identified using the FLEXPART air tracer data and the ambient SO2 mixing ratio in the inverse model. The red color bar shows the simulated annual mean contribution of the source regions to the SO2 mixing ratio during the year of the MOSAiC expedition. The colored polygons show the source region clusters that were identified to have a significant contribution (> 5% annual average) on the simulated SO2 concentrations during the year, which are used in the time series analysis. (B) A timeseries of the monthly median SO2 mixing ratio contributions from the identified source regions, as simulated by the inverse model. The measured SO2 mixing ratio, presented as a rolling 10-minute median during the year, is included for context."

> "Figure 6: Source regions and their contributions to MSA from the inverse model. (A) The potential source regions and their mean contribution to map of MSA between March – September, as identified by the inverse model, are shown in red. The red color bar shows the annual average of the potential source region contributions. The colored polygons show the source region clusters that were identified to have a significant contribution (> 5% annual average) on the simulated MSA concentrations during the year, which are used in the time series analysis. (B) A timeseries of the simulated monthly median MSA concentrations from the source region polygons identified by the inverse model. The MSA time series measured at the ship is included for context, and the shaded region shows the interquartile range. Due to the limited domain of the FLEXPART simulations (> 60°N), source regions polygons "a" and "f" may represent the contribution of MSA transport from regions further south than the polygons depicted on the map, such as the oceanic regions on the western coast of North America and the Bering Sea, respectively."

Line 375: Refer to the polygon number.

The reviewer is correct. This is a useful comment to improve the clarity of the presented results. The reference to the polygon in line 375 has been added:

"…contributions from North Asia/Siberia (polygon a) decline…"

Line 420: Citing Park et al. is not enough here because it handles the DMS rather than MSA. There are studies in the literature that have reported the link between marine biological activity and aerosol chemical composition (or MSA) in different marine environments (e.g., North Atlantic, Mediterranean and Eastern China Seas).

Thank you for this comment. We agree that, as written, the statement in line 420 could be confusing since we did not specify that we were using chl-a to estimate source regions of DMS, which would subsequently impact the measured MSA concentrations. Line 420 has been rephrased to state this explicitly:

"Chl-a, which is a well-known tracer of biological activity (e.g., Park et al., 2013), was used to evaluate the connection between DMS emissions (and subsequent formation of MSA) with biological activity in the ocean."

As our study is focused on the gas phase concentration of MSA, which is sourced from DMS oxidation, we feel that the corrected statement is now sufficient and does not require citing further studies related to MSA's contribution to aerosol composition.

Line 443: Introducing an equation or more explanation on the influence index calculation and the unit used is required to make it clearer.

The reviewer is correct. For improved clarity of the sea surface chl-a influence index, line 443 was updated to be more descriptive of the calculation, as follows:

"The index, called the sea surface chl-a influence index, was obtained by multiplying the residence time (in seconds) of the FLEXPART air tracer below 100 m altitude with the corresponding chl-a concentration maps (in mg·m-3)."

Line 447: It is worth highlighting that DMS emissions are mostly related to senescent phytoplankton cells rather than healthy cells.

This is a helpful comment. It is indeed worth mentioning that DMS emissions result from phytoplankton cells at the end of their lifecycle, whereas chl-a is more representative of healthy cells. Line 447 is updated as follows:

"As previously stated, we do not expect DMS emissions to correspond directly with chl-a, especially since DMS emission are primarily associated with senescent phytoplankton cells and chl-a is indicative of healthier cells, but chl-a serves as a tracer for air masses with potential influence from oceanic biological activity."

Line 723: Who is MDO? What is the role of BH?

Thank you for identifying this error in the author contributions. The description of author contributions has been updated.

Typos:

Line 52: "product" instead of "byproduct"

Corrected.

Line 60: "sulfur" instead of "sulfate"

Corrected.

Line 151: -50% to …

Corrected.

Line 255: remove "or"

Corrected.


Authors' response to Reviewer #3

Reviewer comments in black, Author responses in Red

The manuscript presents very interesting and useful data. It is well organized and clearly written and the elaborations and conclusions are scientifically sound. I recommend publication once the following minor issues are addressed.

Thank you for your general comments regarding this work, as well as your suggestion for improvement below.

Sect. 2.3. This Paragraph deals with a key issue that should be addressed more quantitatively. For the provided plots it is quite evident that pollution from the ship or ongoing surrounding activities did not spoil the measurements, nevertheless the differences should be presented more quantitatively: a statistics test should be employed to demonstrate that the data distribution are not statistically different between the "clean" and "polluted" data subsets. Furthermore, I find interesting the fact that, judging from Figure 1, MSA is more affected from pollution than SA (the difference between the "clean" and "polluted" medians is higher for the MSA case than for SA): have the authors an explanations for this? Which activity can be a source of MSA?

Thank you for this suggestion. Please note that we have used statistical tests on the concentration data between the "clean" and "polluted" periods, as we also described in our response to Reviewer #2. Here is our response from above:

"We have tried using statistical tests on the concentration data between the "clean" and "polluted" periods. Interestingly, the statistical tests suggest that there is a significant difference in the medians between the two groups (Mann-Whitney U test, p<0.01 for SA, MSA, and IA). Given the large number of data points in the two groups (~25 000 "clean" and ~50 000 "polluted" after removing data below the limit of detection), the medians presented in Figure 1 (now Figure S2) are already quite robust, and therefore, statistically significant differences between the two groups are not surprising.

Despite these differences, we have concluded that the variations between the clean and polluted periods is likely coincidental by interpreting other aspects of the data, as suggested by the reviewer, and as explained in the text in section 2.3. First, the differences between the clean and polluted data occur for all three vapors species. While we understand how local pollution could affect SA, to our knowledge, there is no specific activity or local pollution source that can explain the slightly higher observed MSA or IA concentrations during periods of "polluted" data. Thus, since all had slightly higher concentrations during "polluted" periods, we conclude this is likely coincidental due to temporal changes in concentration. Second, the shapes of the distributions, as shown in Figure 1 (now Figure S2), are quite similar for each vapor between clean and polluted periods, even at the lower concentration ranges. If there were a systematic effect of pollution on the concentrations, we would expect the shapes of the data distribution to show more variability. The most notable exception in distribution shape occurs in the SA concentrations > 1e7 molec·cm-3 in the "polluted" group, however, our analysis suggests that these high SA concentrations

are associated with episodic transport of anthropogenic pollution during winter, as described in section 3.1.

Furthermore, we concluded the seasonal analysis in this work remains unaffected by the inclusion of the "polluted" data. Figures S4, S5, and S6 in the supplement show the concentration timeseries between the "clean" and "polluted" periods for SA, MSA, and IA, respectively. Panel B in these figures directly compares the monthly median data between "clean" and "polluted" periods, where the seasonal cycle remains consistent between the two groups. This result demonstrates that our conclusions on the seasonal cycles are not biased by the "polluted" data.

To address this conclusion more appropriately in the text, we adjusted the wording on line 171, where we called the effect of local pollution "negligible." Line 171 now reads as follows:

"Overall, the comparisons presented in Figs. S2, S3, S4, S5, and S6 suggest that local pollution does not affect our analysis of the seasonal cycles for all three gas phase species during the MOSAiC expedition."

Sect. 2.5. Which time resolution have the CHL data?

On line 196, we specify that we used "Eight-day averages of chlorophyll-a (chl-a) concentrations..." in our analysis, which was one of the merged data products available from the Ocean-Colour Climate Change Initiative. We selected the eight-day merged data product since it has better spatial coverage than the one-day merged product (due to cloudiness or algorithm failure), while also providing a high enough time resolution for our 1-year analysis.

Sect. 2.6. Please provide more information on the back trajectories: time resolution, length, frequency. Moreover, was the travelling height of the back-trajectories taken into consideration for the elaborations?

Thank you for this helpful comment. We have updated the description on line 204 to include the time resolution and length of the FLEXPART simulations used in our study. FLEXPART does account for air mass heights during transport. On line 206, we specified that our analysis only considered the amount of time that air masses spent in the lower 100 m of the atmosphere to calculate the Footprint Emission Sensitivity (FES), since we are interested in emissions from the surface. To add clarity to this statement, the description of the FLEXPART simulations and the FES, starting on line 204, now reads as follows:

"FLEXPART is a Lagrangian particle dispersion model that can be used to simulate air mass origins using reanalysis meteorological data. Backward simulations, up to 30 days backward in time, were calculated every 3 hours from Polarstern's location during the MOSAiC expedition using a passive air tracer with a cluster of 100 000 particles in FLEXPART v10.4 (Pisso et al., 2019). These simulations were then used to determine the "Footprint Emission Sensitivity" (FES), or the emission sensitivity of air masses < 100 m during transport, according to the simulated "source-receptor relationship" (SRR) (Seibert and Frank, 2004). Refer to Fig. S7, taken from Boyer et al. (2023), for the FES observed during the MOSAiC expedition."

L265. Why "including SA, MSA, and IA"? Fig. 2a presents precisely SA, MSA and IA.

We included the clause "…, including SA, MSA, and IA,…" since there could be other vapors that could contribute to aerosol formation in remote environments, however, our analysis only considered SA, MSA, and IA. We agree that the sentence was phrased in a way that made this information seem irrelevant. Line 265 was rephrased accordingly:

"The monthly median concentrations of SA, MSA, and IA – vapors that are known to participate in aerosol formation in remote marine environments – are presented in Fig. 1a."

L280. No "dotted white line" is present in Figure 2c.

Thank you for identifying this error. We have corrected the caption for Figure 2c (now Figure 1c) on line 280 to describe the line as "solid" rather than "dotted." Also, according to our response to reviewer 2, the line is now black, as described above.

L317. "To further evaluate the sources of SO2 and SA in our measurements, we examined emissions of anthropogenic sulfate…". I do not think this expression to be correct: by coupling the ECLIPSE v6b with flexpart trajectories the authors are not evaluating the emissions. Please reformulate the sentence.

The reviewer is correct. We are not examining emissions of anthropogenic sulfate, but rather the known source regions. Line 317 has been corrected to properly reflect our analysis:

"To further evaluate the sources of $SO_2$ and SA in our measurements, we examined source regions of anthropogenic sulfate…"

L371. "Despite lower anthropogenic SO4-S influence from the North Asia sector during March and April": refer to the appropriate Figure here, for major clarity.

Thank you for this helpful suggestion to improve clarity. We have updated Line 371 to include the reference to Figure 4 (now Figure 3):

"Despite lower anthropogenic $SO_4$-S influence from the North Asia sector during March and April (Fig. 3), the SA concentration remained high and even reached an annual maximum monthly median concentration of $1.36 \times 10^6$ molec·cm$^{-3}$ during March (Fig. 1a)."

L640. Correct in "sea-ice continues to decline".

Thank you for finding this word omission. We have added "ice" to line 640 to correct this error.

Authors' response to Reviewer #4

Reviewer comments in black, Author responses in Red

General:

This paper is potentially dangerous as it could impede our process understanding of the role of precursor gases in aerosol formation and growth. This could have long-term implications for our knowledge of cloud formation and climate change in the inner Arctic basin, affecting future generations.

This is a very strong statement about our manuscript, which we feel is not supported by the arguments presented by the reviewer. The reviewer's critical comments do not describe how our discussion is "potentially dangerous," and instead, they suggest that the paper "could impede our process understanding of the role of precursor gases in aerosol formation and growth." Please note that this paper is not focused on the role of precursor gases in aerosol formation and growth, and any mention of these processes in the introduction is used to give context for our measurements and analysis. The scope of this work focuses on understanding the seasonal cycle of vapor that are know to be potential aerosol precursors and the observed source regions that contributed to their concentrations in our measurements.

The reviewer does offer critical comments on our interpretation of pollution, which we address below, however, the majority of the reviewer's comments focus on discussing the cited literature in the introduction and suggesting new citations. The reviewer mainly criticized the methods from sources that were used in the introduction, with an emphasis on pointing out the limitations of previous studies. Our response to each comment on the cited literature raised by the reviewer is addressed below.

Interestingly, the reviewer's concluding remark to their general comments contradict their statement that this paper is "dangerous" by stating that "after a convincing decontamination, the study could have made an important descriptive contribution to the sparsely sampled inner Arctic." We would like to point out that we do not propose any large claims that conflict with the existing literature, and we were transparent with our approach to dealing with issues such as local pollution, the description of which is now improved after considering the reviewer's concerns, as described in our responses to specific comments below.

The manuscript references several relevant articles from the field, but the literature survey is not comprehensive. Over the past decades, a substantial amount of observational data has been collected over the summer and early autumn pack ice, which is essential to this study's results and conclusions and, therefore, deserves a discussion and comparison. It would be beneficial to mention or learn from the previous work by Tjernström, Leck, and their colleagues over the last 30 years on the inner Arctic pack ice area, including the marginal ice zone.

We thank the reviewer for suggesting additional sources to improve the manuscript. Our response to the suggestion to add many of these sources is discussed below in the detail comments.

The paper gives the impression that "continuous" sampling had occurred at one Arctic location. All measurement plots, except for Fig. 2, show only time series, disregarding the variation of latitude and longitude during sampling while on a moving ship.

We understand the reviewer's concerns. It was not our intension to give the impression that no drift has taken place, and we tried to be transparent about this. To ensure that there is no misunderstanding we included Figure S1 with the drift track of the expedition in the original manuscript. It is referred to on line 120. In addition, as the reviewer correctly pointed out, Figure 2 (now Figure 1) shows the changes in latitudinal position of the ship. To add additional clarity, we also added a statement in the methods referring to the ship's location in Figure 2 (now Figure 1) on line 121 as follows: "Refer to Fig. S1 in the supplemental information (SI) for a map of the drift track and Fig. 1c for the monthly median location of Polarstern during the campaign."

Furthermore, the publicly available data set that accompanies this work includes concurrent measurements of the ship's latitude and longitude such that they can be considered alongside the gas phase concentration data. We did not include the latitude of the ship in every figure as this would be redundant, and the readers can refer to the figures denoting the ship's location (including Figure 1c in the main text). Moreover, the FLEXPART and inverse model simulations took the ship's location into account, such that relevant source regions that influenced our measured gas phase concentrations during different times of year could be evaluated despite the ship's drift, which as the reviewer pointed out below leads to "important insights into the source regions of the aerosol precursors in different seasons". We only claim that the year-long data set was collected "north of 80°N for almost the entire expedition" (line 120) and includes novel observations of parameters measured during polar night.

The inverse model uses the FLEXPART source-receptor-relationship and the measurement time series from the ship to identify potential source regions. It is unclear whether and which DMS emissions were used in the FLEXPART runs. While the inverse modeling technique results in important insights into the source regions of the aerosol precursors in different seasons, the time series analyses in the paper are superficial.

We thank the reviewer for this helpful comment. The descriptions of FLEXPART and the inverse model have been updated to add clarity to the text. Specifically, we added more detail to clarify that the inverse model simulations, which used our measurement time series data, were performed on the SA, $SO_2$, and MSA data. DMS was not described, as it was not used in the inverse model or FLEXPART simulations, as the inverse model does not account for chemical processing during transport, assumes that the emission rates are constant from the identified source regions, and requires our measurement timeseries, as described in section 2.6.

In general, analyses of atmospheric transport using FLEXPART and inverse modelling are qualitative in nature, but can still be used to obtain scientific insights, as the reviewer suggested. In our analysis, we used the simulated concentration time series of SA, $SO_2$, and MSA to compare the relative contributions from the different source regions during different times of year. We explicitly stated that these simulated time series data are qualitative in Section 2.6 of the methods in our description of how the FLEXPART and inverse model simulations were used to obtain the simulated time series:

Line 249: "Finally, we obtained a timeseries of simulated influence from the source region polygons, which we used to provide qualitative insights on the seasonal contributions from different source regions."

It's further unclear why no observation of particulate MSA (MSAp) or nss-SO4 was included in the manuscript. Were these parameters not measured during MOSAiC? The conclusions remain vague without an analysis of MSAp/nss-SO4 molar ratios, especially since this study's high-MSA concentration periods provide little explanation for the low levels of particulate MSA reported from previous studies over the summer early autumn pack ice area.

We have addressed this question raised by the reviewer in response to a specific comment below.

The paper should be rejected foremost because of grave methodological uncertainties, which must be specified to be convincing, such as demanding a cleaned CPC record showing that decontamination resulted in CPC levels comparable to previous results over the pack ice when existing. After a convincing decontamination, the study could have made an important descriptive contribution to the sparsely sampled inner Arctic.

The grounds on which the reviewer suggests that the paper should be rejected are vague here. Their comment on the methods suggest that pollution is their primary concern. Please note that the pollution detection algorithm used in our study is based off condensation particle counter (CPC) measurements, which has been explored in detail previously and was the topic of a dedicated study presented in Beck et al. (2022). We cited the relevant source discussing the pollution detection algorithm and specified that the measurements were based off the CPC data in line 172:

"Polluted periods were determined by applying the pollution detection algorithm developed by Beck et al. (2022) to particle concentrations from a condensation particle counter (TSI, model 3025) located in the Swiss container."

The method by Beck et al. (2022) is a further development of the method applied in Baccarini et al. (2020), which was widely applied to aerosol observations during the Arctic Ocean 2018 expedition (e.g., Karlsson et al., 2022). The Beck et al. (2022) has been used widely for publications of the MOSAiC expedition (see here for references: https://scholar.google.com/scholar?oi=bibs&hl=de&cites=13177926182926935824).

Given the establishment of the method in the peer-reviewed literature, we are confident that we have applied state of the art data pollution cleaning.

Further discussion of the influence of pollution on our data are addressed in our response to the detailed comments below.

Detailed comments:

Line 36: Meier et al., 2014 could be replaced with a more suitable citation. Its focus is not on the air-sea exchange of particles.

The reviewer is correct, Meier et al. (2014) does not focus on air-sea exchange of particles. However, we cited Meier et al. (2014) here due to their summary of how sea ice extent is expected to affect the atmospheric processes mentioned, including "changing atmospheric transport patterns, increasing anthropogenic activities such as ship traffic, and increasing ocean-atmosphere exchange of gases." According to the reviewer's suggestion, we have removed "and particles" from line 36.

Line 37: Define "in the Arctic", "the central Arctic Ocean, "high Arctic". Do you mean north of 80°?

We thank the reviewer for their comment. The reviewer is correct: the discussion in this study typically pertains to the high Arctic, north of 80°N, which is also the central Arctic Ocean where our data was collected. To make this clearer to the reader, we added "north of 80°N" in the introductory on lines 33 and 37, in the abstract on line 18, and in the conclusions on line 617.

Line 40: Discussing past observations made over the pack ice during summer and early autumn would be worthwhile. These observations have shown that particulate organics present in Aitken and accumulation mode aerosol and cloud water had properties like marine polymer gels. These gels were found to originate from the surface microlayer on leads. This behavior was attributed to the activity of ice microalgae, phytoplankton, and possibly bacteria. Some studies exploring these observations include Leck and Bigg 2005, Bigg and Leck 2008, Orellana et al. 2011, Karl et al. 2013, Hamacher-Barth et al. 2016, and Lawler et al. 2021.

We thank the reviewer for their suggestion to include this literature in the introduction. Given that these papers are focused on primary emissions of aerosols in the Aitken and accumulation mode, and our study is focused on specific trace gases and aerosol precursor vapors that participate in new particle formation (NPF), our original manuscript did not include any references to sea spray and marine nanogels. Please also note that our introduction is not implying that sea spray or marine nanogels do not contribute to Arctic aerosol, it is simply not discussed as it is beyond the scope of the current study. We did indeed cite other studies from summer and autumn in the high Arctic that are relevant in our study, which the reviewer also commented on; our responses to each are given in the proceeding comments. We have added the following statement at line 40 according to the reviewer's suggestion:

"Other relevant sources of particles in the high Arctic during summer could include sea spray and marine nanogels formed by bubble bursting (Leck and Bigg, 2005; Bigg and Leck, 2008; Karl et al., 2013; Lawler et al., 2021)."

Line 42: Please provide the exact citation indicating that 90% of CCN in the Arctic during summer is explained by NPF. Please include the latitude range and period used to define the summer Arctic.

Gordon et al. (2017) implemented NPF parameterizations obtained from the CERN-CLOUD experiments into a global aerosol model to simulate the contribution of NPF to CCN concentrations. The results, shown in Figs. 3d and 4d in Gordon et al. (2017), suggest that that up to 80% of CCN at

0.2% supersaturation could originate from NPF, and up to 90% of CCN at 1.0% supersaturation could originate from NPF north of ~80°N. Based on their presentation of the data, we included the statement that "up to 90%" of the CCN could be "estimated" to originate from NPF, and we did not definitively state that 90% of CCN in the Arctic is explained by NPF. This estimate is, of course, based on model simulations and subject to uncertainty. Please also note that on line 43 we explicitly stated that this contribution "up to" 90% is an estimate and remains "poorly constrained", owing to limited observations of the contribution of NPF to CCN in general.

Line 43-44: The region of the Arctic that is situated north of 80° witnesses a significant seasonal disparity all year round. The cause of this contrast can be attributed to the temperature and sea ice conditions, the uninterrupted sunlight in summer, and the prolonged polar darkness in winter. These diverse circumstances bring about significantly different atmospheric transport dynamics, atmospheric aerosol precursor gases, and their chemical reactions, which will affect the life cycle of aerosol particles over the pack ice area in distinct ways. Thus, the literature cited on a global evaluation of CCN formation by direct sea salt and ultrafine particles or CERN-CLOUD measurements of global particle formation or present and pre-industrial new particle formation does seem unrelated to this study's observations over the pack ice area and as such, should be omitted. The only citation with some relevance is Kecorius et al., 2019.

We thank the reviewer for their summary of the changing Arctic conditions over the year that influence aerosol precursor vapors and aerosol processes. Please note that this summary is also in concordance with our discussion and conclusions presented in this study.

We understand the reviewer's concern about the relevance of the cited papers to this work. We chose to include the references that use the mechanisms of NPF, obtained from the CERN-CLOUD measurements, in global aerosol models to predict CCN because direct observations are limited. In general, it is a challenge to observe of NPF and subsequent growth to CCN sizes from field measurements due to changing meteorological conditions or air mass exchange, during which the growing nucleation mode particles are advected away from the measurement location. Modelling can help in this regard. In addition, the measurements at CERN-CLOUD provide detailed insights on the NPF mechanism by operating under controlled circumstances that are difficult to obtain in the field. Hence, the application of the CERN-CLOUD mechanisms in the models can also yield estimates on the role of NPF on CCN concentrations in various environments, including the Arctic, that are based on the current understanding of the NPF mechanisms.

As stated above, we acknowledge that there are a range of estimates on the contribution of NPF to CCN concentrations in the Arctic, which are uncertain. This information was presented in our original statement. According to the reviewer's suggestions, we have removed Pierce and Adams (2006) and Dunne et al., (2016) from the cited references. The citation of Gordon et al. (2017) was not removed according to the reasons given in the previous comment. The statement was rephrased to be more clear about the information from the given sources, as follows:

"NPF, which can yield high number concentrations of particles, is estimated to contribute to cloud condensation nuclei (CCN) number concentrations in the Arctic, however, these estimates remain poorly constrained. Global model estimates range up to 90% contribution of CCN concentrations from NPF (Gordon et al., 2017), and field observations suggest that newly formed particles can increase background CCN concentrations by up to a factor of 5 (Kecorius et al., 2019)."

Line 45-48: It should be clarified that the cited studies are performed in summer to early autumn, but Koike et al., 2019, report on two years of continuous in situ measurements at the Mount Zeppelin Observatory (78°56'N, 11°53'E), in Ny-Ålesund, Spitsbergen. The relevance of the inner Arctic of Koike et al.'s 2019 study is thus questioned; please omit.

We had included Koike et al. (2019), as measurements from land-based sites around the Arctic provide valuable context for atmospheric processes in the Arctic region in general, and the continuous measurements at the land-based sites is an asset compared to the sparsely sampled central Arctic Ocean. At the suggestion of the reviewer, we have removed Koike et al. (2019) and specified the time of year from Baccarini et al. (2020) and Karlsson et al. (2022). Please note that the statements in lines 45 – 48 are revised further in our response to the preceding comments.

Line 46: The Mauritzen et al. (2011) result only represents the inner Arctic summer; please add this information. Mauritsen et al. (2011) estimated a threshold of 10 to 16 CCN per cm3 as a minimum for cloud formation and sustenance. Past observation also found that the CCN concentrations around the ice sheet's edge were highest but dropped almost tenfold due to wet scavenging within 1-2 days of advection from the open sea into the pack ice (Bigg and Leck, 2001; Leck and Svensson, 2015).

We thank the reviewer for their suggestion to point out that the conclusions in Mauritsen et al. (2011) are based on the summertime Arctic and to include the discussion on wet scavenging here. We have rephrased line 46 as follows:

"Moreover, aerosol-cloud interactions have an important, yet uncertain, role in the surface energy budget in the central Arctic Ocean where summertime aerosol concentrations can be so low that cloud formation is CCN-limited (Mauritsen et al., 2011), due to wet scavenging of particles during advection from the marginal ice zone into the ice pack (Leck and Svensson, 2015)."

Line 47-48: The modeling study conducted by Bulatovic et al. in 2021 aimed to investigate whether Aitken mode particles can serve as CCN. The study simulated median supersaturations between 0.2% and 0.4%, with a range of up to 1%. The results showed that even small Aitken mode aerosols of ~30nm diameter can be activated as long as larger accumulation particles are low in number concentration, preventing the depletion of excess water vapor. The simulations used typical aerosol size distributions encountered in the central Arctic during the summer/early autumn of 1991, 1996, 2001, and 2008 (Heintzenberg and Leck, 2012). It was found that having a low concentration of accumulation mode particles and a high concentration of Aitken mode particles in the inner Arctic during summer and early autumn, which created a favorable environment for the activation of CCN in the small Aitken mode range, is a rare occurrence.

The statements that the high Arctic summer/early autumn studies by Baccarini et al., 2020, and Karlsson et al., 2021, show that very small particles can act as CCN has to be weakened. No direct evidence has been presented in the studies since the conclusions were based on inferred proofing. Please also possibly add the results of Bulatovic et al., 2021.

We thank the reviewer for highlighting the findings of the modelling study by Bulatovic et al. (2021). Our statement on lines 47 and 48 is in agreement with the discussion and conclusions presented in Bulatovic et al. (2021), as we specifically referred to the previous statement where "aerosol concentrations can be so low that cloud formation is CCN-limited".

The evidence presented from Baccarini et al. (2020) is based on field measurements of a growing particle mode, CCN concentrations, as well as cloud droplet residuals measured behind a counterflow virtual impactor, all of which point to the same conclusion that particles of ~30 nm, sourced from NPF, lead to an enhancement in CCN concentrations.

Likewise, the data from Karlsson et al., 2022 is also based on observations of cloud droplet residuals from the field. In their conclusions, they also specify that they "provide direct and extensive experimental evidence that cloud-forming particles over the central Arctic Ocean are strongly influenced by Aitken mode particles with an average mode diameter of around 30 nm" (Karlson et al., 2021).

The conclusions from Baccarini et al., (2020) and Karlsson et al., (2022) also agree with additional studies from Kecorius et al. (2019) and Chang et al. (2022), which are also based on field observations. Therefore, we conclude that our initial statement was appropriate.

According to the reviewer's suggestions, however, we have rephrased the statement to weaken it and also added further references, as follows:

"Recent observations suggest that very small particles may act as CCN under such conditions during summer and early autumn in the high Arctic (Leaitch et al., 2016; Kecorius 2019; Baccarini et al., 2020; Karlsson et al., 2021; Chang et al., 2022), which also agrees with modelled Aitken mode particle contribution to CCN at particle diameters < 50 nm (Bulatovic et al., 2021)"

Line 48: A citation is missing after "exceptionally high"; clarify when and where in the Arctic.

As described in our response to the last comment, to weaken this phrase according to the reviewer's suggestion, this clause was removed from line 48.

Line 49: To ensure the continuity of in situ data archives of the high Arctic north of 80°, unique measurements have been conducted during 5 research expeditions from 1991 to 2018. (Leck et al., 1996; Leck et al., 2001; Leck et al., 2004; Tjernström et al., 2014; Leck et al., 2019). Please remind the reader of their presence.

We thank the reviewer for their suggestion to highlight previous studies in the high Arctic, which are helpful to support the motivation for our study. The statement on line 49 concerns measurements of aerosol precursor vapor concentrations, which due to limitations in available instrumentation and short duration campaigns, has not been adequately measured to obtain insights on the complete seasonal cycle of these vapors in the central Arctic.

For example, Leck et al. (2001) states: "However, field measurements of methane sulfonic acid and $H_2SO_4$, and of hydroxyl radicals were difficult to measure with the 1991 technology available, given the low concentrations expected in clean marine air. They were therefore not included."

To point this out the reader we have added a sentence after line 49 as follows:

"Many of the past observations in the central Arctic Ocean are limited, either by short temporal coverage in summer/early autumn or insufficient instrumentation to quantify aerosol precursor vapor concentrations (Leck and Persson, 1996a; Leck et al., 2001, 2004; Tjernström et al., 2014; Leck et al., 2019)."

The reviewer has also helped us improve this point by ensuring that we specify that measurements from previous campaigns, as described in other specific comments, are typically limited to summer and early autumn.

Line 52-57: The cited references are not specific to observations/modeling during the biologically most active conditions over the pack ice and at the marginal ice zone (MIZ). I would encourage the authors to read and cite the following papers and useful references therein: Leck and Persson, 1996a,b; Kerminen and Leck, 2001; Lundén et al., 2007; 2010.

The reviewer is correct. The observations in the cited references in line 52 - 57 are not specific to the Arctic, but this is not the point of these citations here. Instead, these introductory sentences are intended to inform about the general sources of dimethyl sulfide (DMS) and its oxidation products. The references suggested by the reviewer are more appropriate in the specific discussion of DMS oxidation products in the Arctic, which starts on Line 67. This is addressed below in the comment on lines 67 – 69.

Line 58-63: An important precursor of aqueous phase-produced sulfate is the DMS oxidation product hydroperoxylmethyl orthoformate (HPMTF), which has been completely overlooked in this study. The global burden of HPMTF has been calculated to be 2.6–26 Gg S (see Cala et al., 2023). The general understanding is that HPMTF is taken up rapidly in cloud water; however, Cala et al. (2023) found that rapid cloud uptake of HPMTF worsens the model–observation comparison.

The reviewer is correct, and this is a helpful suggestion. We did not include HPMTF in our discussion of intermediate sulfur species that can also impact the yield of MSA, SA, and SA from DMS oxidation. Line 58 has been revised as follows:

"In addition, there are other intermediate sulfur species that result from DMS oxidation, including dimethyl sulfoxide (DMSO), methane sulfinic acid (MSIA), and hydroperoxyl methyl thioformate (HPMTF), that readily undergo aqueous-phase processing. Reactions involving these intermediate species can contribute directly to sulfur aerosol mass in the marine boundary layer and affect the yield of MSA, SO2, and SA from DMS oxidation (Hoffmann et al., 2016; Chen et al., 2018; Siegel et al., 2023; Cala et al., 2023)."

It would also be worthwhile to discuss the role of direct formation of sulfuric acid from the gas-phase oxidation of DMS with the OH radical, as demonstrated by Berndt et al. (2023). The direct production of sulfuric acid in DMS oxidation has been speculated about in the literature for decades (e.g., cite Lucas and Prinn, 2002 and Karl et al., 2007).

In fact, during summer, the gas phase data collected in the MOSAiC expedition after a convincing decontamination could potentially help understand the relevance of direct sulfuric acid formation in the central Arctic.

As per the reviewer's suggestion, we have added a statement about the direct production of SA from DMS on line 64:

"Recent observations also suggest that direct formation of SA is possible via oxidation of DMS with the OH radical (Berndt et al., 2023), a process which has been proposed for decades (Karl et al., 2007; Lucas and Prinn, 2002)."

Line 64: Please cite Leck and Bigg, 2011, Heintzenberg et al., 2017, and Karl et al., 2019. Yet another explanation for the occurrence of high numbers of nucleation mode particles in the high Arctic, involving granular nano gels in addition to sulfuric acid, was proposed by Karl et al. (2013). The appropriate discussion of these findings on nanoparticle events observed over the Arctic Ocean must be included in the introduction.

Please note that the sentence on line 64 is in reference to the known role that SA and MSA play in NPF and the subsequent growth of the newly formed particles. The studies that we cited in line 64 featured online measurements of aerosol precursor vapors, the chemical composition of growing clusters during NPF events, and the number size distribution of particles and ions at the smallest sizes (< 1 nm). It is essential to measure these parameters to adequately describe the mechanism of NPF from observations. Thus, these cited references are suitable here, as they demonstrate the contribution from these precursor vapors. In contrast, the sources recommended by the reviewer do not include such measurements and instead infer characteristics about NPF. Therefore, we choose not to include them here.

Discussion of alternative hypotheses of nucleation mode aerosol sources is not relevant to this statement and was already discussed in a previous comment, where we added a statement discussing the source of nanogels to Arctic aerosol on line 40.

Line 67-69: The citations report results from observations during spring and summer in the Arctic. This information is too unspecific. Please clarify for each of the citations used from where (baseline station, latitude and longitude, at the MIZ, or over the pack ice area) and more exactly during which period they represent.

The reviewer is correct, we could have been more specific for each of these cited studies. We have now updated lines 67 – 69 to include sampling location from these studies, as follows:

"The concentration of gaseous DMS and its oxidation products (SA and MSA) in both the gas and particle phases have been observed to increase at various locations across the Arctic, including during spring and summer at Ny Ålesund, Svalbard, Norway (Beck et al., 2021); during spring at Villum Research Station, Greenland, Denmark (Nielsen et al., 2019), during spring and summer in the Canadian Arctic (Abbatt et al., 2019; Ghahremaninezhad et al., 2017); and in summer over ice pack in the central Arctic ocean (Kerminen and Leck, 2001; Lundén et al., 2010); which corresponds to the appearance of sunlight and phytoplankton activity in the region."

Please note that while we mentioned the time of year and the sampling locations, we did not include the latitude and longitude of these measurements, which can be obtained by looking at each individual reference.

Line 69-70: The cited papers are irrelevant to the present study as they provide general information or cover only a small part and a short period in the central Arctic Basin. Therefore, it is recommended that all the citations be removed.

Ghahremaninezhad et al., 2019, do not report any observations on CCN. They implemented DMS(g) in Environment and Climate Change Canada's forecast model and compared the model simulations with DMS(g) measurements made in Baffin Bay and the Canadian Arctic Archipelago in July and August 2014.

We included Ghahremaninezhad et al. (2019) since their simulations suggest that "DMS oxidation products affect the natural aerosol populations", as we wrote on line 71. Please also note that in our introduction to these sources, we wrote that "other studies suggest" and did not state that they include direct observations. We have not misquoted their interpretation of their results here and argue that our citation of their conclusions is relevant for the introduction of our study. For example, here is a direct quote from Ghahremaninezhad et al. (2019) on DMS's contribution to Arctic aerosol and CCN according to their results:

> "Our modelling results indicate the formation/growth of biogenic sulfate aerosol in the size range from 10 to 200 nm. These results suggest that the non-sea-salt sulfate aerosol in the summertime Arctic is dominated by fine and ultrafine biogenic particles, which may act as a CCN and/or influence CDNCs and play a climatic role." Ghahremaninezhad et al. (2019)

Mayer et al., 2020, did not perform any direct measurements of CCN and only inferred them from a 6-day mesocosm experiment throughout a phytoplankton bloom.

The inclusion of Mayer at al. (2020) is very similar to the inclusion of Ghahremaninezhad et al. (2019). Please note, as stated above, that we stated "other studies suggest" in the introduction of these sources and did not imply that they made direct measurements in the marine boundary layer. The reviewer is correct; the study of Mayer et al. (2020) was performed using mesocosm experiments, where they measured the hygroscopicity of the droplets produced by secondary aerosol production and nascent sea spray. They observed that sulfur components dominated aerosol mass in their experiments, which they were able directly connect to DMS oxidation products and subsequent CCN activity. Therefore, we feel that our citation of this source here was appropriate, as their measurements do "suggest that DMS oxidation products affect the natural aerosol population and CCN concentrations in the marine atmosphere (Mayer et al., 2020)...", as we stated on line 71.

Charlson et al., 1987, presented a hypothesis about the feedback between ocean temperature and cloud radiative properties via DMS ocean and air, particulate Sulftat, and CCN/numbers of cloud droplets. Observations from the Arctic and other marine regions have questioned the key role attributed to DMS in the CLAW hypothesis for over a decade (Leck and Bigg, 2007; Quinn and Bates, 2011).

The reviewer is correct about the hypothesis presented by Charlson et al. (1987). Their discussion on the role of DMS is not focused on the Arctic specifically, but in remote marine regions, where they propose that DMS can contribute to secondary aerosol, and subsequently, CCN. While we understand that the CLAW hypothesis has since been questioned due to its overly simple treatment of various processes connecting DMS to CCN, namely in the articles mentioned by the reviewer, the contribution of DMS to gaseous sulfur compounds that can nucleate, in the presence of other gases, is supported by observations in the Arctic region near Svalbard (e.g., Lee et al., 2020; Beck et al.,

2021). For this reason, we included the reference to Charlson et al. (1987), but in effort to weaken our statement according to the reviewer's comments, we have removed this citation and instead cited the study of Lee et al., 2020.

Carslaw et al., 2010 is a review paper that discusses natural aerosols in the Earth System as a whole.

The reviewer is correct, Carslaw et al. (2010) does present a discussion of natural aerosols and their climate impacts on the earth system as a whole. In their discussion of DMS-derived aerosol, they present the need for further observations of DMS production in sea ice due to the potential impact that the subsequent DMS may have on CCN and climate, which is why we included the citation here. We have removed the citation here at the request of the reviewer.

Considering each of these comments, lines 69 – 70 now read as follows:

"Other studies suggest that DMS oxidation products affect the natural aerosol population, and potentially the CCN concentrations, in the marine atmosphere (Mayer et al., 2020), over the Arctic Ocean (Ghahremaninezhad et al., 2019), and in Svalbard (Lee et al., 2020)."

Line 72-73: I disagree; there is no direct evidence that decadal DMS emission trends are positive across the Arctic due to decreasing sea ice coverage. Both studies cited infer temporal variations in ocean dimethyl sulfide emissions using either a remote sensing algorithm based on either estimated sea-surface DMS concentration (nM) from remotely sensed chlorophyll concentration light penetration depths and photosynthetically available radiation or reconstruction of the annual and seasonal MSA flux with monthly resolution from a high-resolution ice core obtained from the SE-Dome, southeastern Greenland Ice Sheet (ca 250 masl) and satellite-derived chlorophyll-a datasets.

Please replace "direct evidence" with "inferred evidence".

It should be noted that the estimates made by Gali et al. for the central Arctic basin are subject to large uncertainties as they are based solely on inferred remote sensing. Additionally, it is important to consider the satellite orbit inclination and instrument swath, which create a data gap north of approximately 87 degrees.

We thank the reviewer for summarizing the results of these studies.

Please note that in line 72 – 73, we did not use the word "direct" in relation to the evidence of increasing DMS trends. We stated that "there is evidence that decadal DMS emission trends are positive…". There are no direct in situ observations of DMS that span several decades in the central Arctic, meaning that evidence inferred by the cited studies is insightful and worth mentioning here. To satisfy the reviewer's concern, we have added the word "inferred" on line 73.

Line 75-77: The paper by Schmale et al. (2022) is referenced to support the claim that DMS oxidation products in the aerosol phase over the Arctic do not show a persistent positive trend. This is not

correct. Schmale et al. (2022) suggest no uniform picture of a trend in the Arctic region. The trends in particulate matter of MSA vary depending on the location and the decade.

We thank the reviewer for their comments on these trends, which is helpful to clarify our statement. Our statement that there is no "persistent positive trend" is similar to the reviewer's statement that there is "no uniform picture of a trend" on the DMS products in the aerosol phase from Schmale et al. (2022) and Sharma et al. (2019) on line 75 – 77, but it has been updated to make this more clear, as follows:

"At the same time, observations of DMS oxidation products in the aerosol phase in the Arctic do not show a uniform trend (Schmale et al., 2022; Sharma et al., 2019)…"

Line 77: Which precursor vapors? Please specify.

According to the reviewer's suggestion, we have changed "precursor vapors" to "SA and MSA" on line 77.

Line 77-80: Sentence starting with "As a result….. Omit the sentence as its content falls outside the scope of this paper. As such, it is not pertinent and only useful for raising funds.

We thank the reviewer for their concern with statements that are within the scope of our paper, however, we disagree with the reviewer's comment. The sentence on line 77 – 80 is not intended for funding purposes, but rather, as motivation for our study. The statement points out that knowledge gaps still exist in our understanding of DMS oxidation chemistry, and mentions that observations of such products could be useful in models, which our study provides.

We have not removed this sentence, however, we have weakened it according to the reviewer's suggestion by removing the end of the sentence. The statement on lines 77 – 80 now reads as follows:

"As a result, DMS oxidation remains poorly represented in large scale climate models, despite its climate relevance, and thus, direct observations of its oxidation products can help support modeling activities (Chen et al., 2018; Hoffmann et al., 2016, 2021; Quinn and Bates, 2011; Veres et al., 2020)."

Line 104-105: "of these gas-phase species in the Arctic." Please specify which gases were observed in the Arctic and where and when they were collected. None of the cited papers seems to report on new particle formation (observations and modeling using in situ observations) over the pack ice or any of the precursor gases discussed above. Suggested for further reading are Baccarini et al., 2020; Bigg et al., 2001; Kerminen et al., 2001; Leck and Persson, 1996a; Karl et al., 2012; 2013.

The reviewer is correct. It is more clear to the reader to explicitly state that gases to which we are referring here. Therefore, according to the reviewer's suggestion, we have changed "gas phase species" on line 103 to "SA, MSA, and IA."

Further, we do not understand why the reviewer stated that none of the cited papers refer to "any of the precursor gases discussed above". Each of the cited sources (Beck et al., 2021; Croft et al., 2019; Willis et al., 2018) discuss implication for sulfur compounds from DMS (SA and MSA) and iodine compounds (IA) on aerosol formation in the Arctic region and express the need for further

measurements of such precursor vapors to improve our understanding of aerosol processes in the Arctic, which is why we cited these studies here. The reviewer is correct, however, that we may have made our intended purpose for citing these papers unclear by including "in-situ observations" in this sentence. We have rephrased the sentence accordingly for clarity.

In addition, Baccarini et al. (2020) is the only article suggested by the reviewer that includes measurements of the aerosol precursor vapors in our study. We have included this citation according to their suggestion. The other studies have little relevance here.

Considering these comments, the statements on lines 104 and 105 were updated as follows:

"Despite the knowledge from previous studies, observations of SA, MSA, and IA across the Arctic region in general are scarce. This lack of observations limits our understanding of NPF, particle growth, and aerosol size distributions to resolve climate-relevant aerosol processes in the Arctic (Beck et al., 2021; Croft et al., 2019; Willis et al., 2018; Baccarini et al., 2020)."

Line 113-114: The study's main weakness is the lack of analysis of MSAp/nss-SO4 molar ratios, which results in unclear conclusions on the high gas phase MSA periods observed in the present study when compared with previously reported inner Arctic low particulate MSA concentrations (e.g., Leck and Persson, 1996a).

Our analysis focused on the seasonal cycle of gas phase concentrations and sources of MSA, which is why we did not focus on particulate MSA. Detailed analyses of the aerosol chemistry, particularly involving MSA, are beyond the scope of this study. Aerosol chemical composition was measured during MOSAiC. There are currently dedicated manuscripts on the aerosol chemistry and the source apportionment of aerosol chemical components, as measured during MOSAiC, in preparation. To specify these upcoming studies in the manuscript, we updated the text as follows in our discussion of gas phase MSA:

Line 528: "Note that these aerosol processes involving MSA are beyond the scope of the seasonal analysis presented here, however, the role of MSA in event-level analyses of the mechanism of NPF, investigations of aerosol chemistry, and source apportionment of aerosol chemical components during MOSAiC will be presented in future studies."

Line 155-161: Diesel exhaust is a direct source of sulfuric acid (depending on engine load, fuel type, and fuel sulfur content), especially when run in connection with a diesel particle filter or an oxidative after-treatment system, which reduces hydrocarbon emissions but simultaneously increases SO2 to SO3 conversion, responsible for direct sulfuric acid formation. Nucleation in diesel exhaust from engines equipped with after-treatment or particle filters may result in enhanced nucleated particles, see Pirjola and Karl (2015).). Typically, reported conversion rates of SO2 to SO3 for ships using low sulfur fuels are in the range of 1–3%, leading to sulfuric acid concentrations in the range of 0.1–0.5 x 10^11 molecules/cm^3 close to ship stack (Karl et al., 2020). Give details on the ship engine of the Polarstern and possible use after treatment systems of the diesel exhaust. Did they run the ship engine all the time during the drift? Which fuel was used, and what was the sulfur content? Also, when helicopter flights and snowmobiles are considered?

For fuel, *Polarstern* uses low-sulfur diesel fuel, specifically distillate marine fuel, which has a maximum sulfur content of 0.1%. During the drift, the main engines were turned off, however, fuel was still consumed to provide electricity and heat. The fuel consumption during the drift was estimated to be 15 tonnes of diesel per day, whereas diesel consumption during transit consumes 54 tonnes per day. We were unable to obtain detailed information on possible exhaust gas treatment systems onboard.

We included this information in the lines 155 – 161 as follows:

"For fuel, *Polarstern* used low-sulfur diesel fuel, specifically distillate marine fuel, which has a maximum sulfur content of 0.1% m/m. While drifting in the pack ice, *Polarstern*'s main engines were turned off, however, fuel was still consumed to provide electricity and heat."

Other logistical activities, such as helicopter flights and snowmobiles, are discussed extensively in Beck et al. (2022), which we cited as the source of the pollution detection algorithm used in our study. Beck et al. (2022) used a multistep approach to identify and flag pollution based on gradients in particle number concentration data collected during the MOSAiC expedition. The study also demonstrated the strengths of this pollution algorithm against traditional methods, such as wind sector filtering, which would fail to detect transient pollution spikes associated with logistical activities that are not associated with the ship exhaust. The pollution detection algorithm was also compared between multiple instruments, to other measurement sites, and to visual detection of pollution with 94% agreement. We would like to point out that the pollution flag obtained with this pollution detection algorithm is a published dataset and is also published with the aerosol precursor vapor concentration dataset presented in this work. We did not include all this information in the text, as we cited Beck et al. (2022), but for clarity of the inclusion of logistical activities in the pollution detection, we added the following information to line 173 in section 2.3:

"Polluted periods, including the local pollution influence from the ship exhaust and logistical activities, were determined by applying the pollution detection algorithm developed by Beck et al. (2022) to particle concentrations from a condensation particle counter (TSI, model 3025) located in the Swiss container."

Line 175-176: I firmly believe that the current data shown in panel A for respective Figs S2, S3, and S4 is inadequate in demonstrating the differences between clean and polluted periods. To better showcase these differences, it would be beneficial to include daily examples before and after applying the pollution mask. An example of this can be found in Figure S1 of Boyer et al.'s 2023 publication."

We understand the reviewer's concerns with pollution in the data. Please note that the differences between the "clean" and "polluted" periods are indicated in Fig 1 (now Fig S2), whereas the purpose of Figs S2, S3, and S4 is not show the differences between the "clean" and "polluted" periods, but rather the similarities. From Figs S2, S3, and S4 (now Figs S4, S5, and S6), it is apparent that, despite the inclusion of the "polluted" data, there are no key differences between the "clean" and "polluted" timeseries, suggesting that our conclusion on the seasonal cycles of each species remain intact with the inclusion of the "polluted" data. Please note that we also responded to similar concerns about the differences between "clean" and "polluted" periods in our response to reviewers #2 & #3, where we stated the following:

"We have tried using statistical tests on the concentration data between the "clean" and "polluted" periods. Interestingly, the tests suggest that there is a significant difference in the medians between the two groups (Mann-Whitney U test, p<0.01 for SA, MSA, and IA). Given the large number of data points in the two groups (~25 000 "clean" and ~50 000 "polluted" after removing data below the limit of detection), the medians presented in Figure 1 (now Figure S2) are already quite robust, and therefore, statistically significant differences between the two groups are not surprising.

Despite these differences, we have concluded that the variations between the clean and polluted periods is coincidental by interpreting other aspects of the data, as explained in the text in section 2.3. First, the differences between the clean and polluted data occur for all three vapors species. While we understand how local pollution could affect SA, we are not aware of any mechanisms where local pollution could affect the concentrations of MSA and IA, suggesting that the higher concentrations of each species during polluted periods could be coincidental due to temporal changes in concentrations. Second, the shapes of the distributions, as shown in Figure 1 (now Figure S2), are quite similar for each vapor between clean and polluted periods, even at the lower concentration ranges. If there were a systematic effect of pollution on the concentrations, we would expect the shapes of the data distribution to show more variability. The most notable exception in distribution shape occurs in the SA concentrations > 1e7 molec·cm-3 in the "polluted" group, however, our analysis suggests that these high SA concentrations are associated with episodic transport of anthropogenic pollution during winter, as described in section 3.1.

Furthermore, we concluded the seasonal analysis in this work remains unaffected by the inclusion of the "polluted" data. Figures S4, S5, and S6 in the supplement show the concentration timeseries between the "clean" and "polluted" periods for SA, MSA, and IA, respectively. Panel B in these figures directly compares the monthly median data between "clean" and "polluted" periods, where the seasonal cycle remains consistent between the two groups. This result demonstrates that our conclusions on the seasonal cycles are not biased by the "polluted" data."

The reviewer is correct that we could also show daily examples for added support. We have added the following figure to supplement, showing a daily example comparing particle number and vapor concentrations during a day with "clean" and "polluted" periods:

[Figure]

**Figure S3. A single-day comparison of particle number and aerosol precursor vapor concentration timeseries during clean and polluted periods.** The particle number concentrations during clean periods are shown with solid green circles, whereas the periods identified as local pollution are indicated by hollow green circles. The red, blue, and black lines correspond to the concentrations of SA, MSA, and IA, respectively. The reported particle number concentrations were measured with a condensation particle counter (TSI 3025), which was also used to identify local pollution using the pollution detection algorithm described in Beck et al. (2022).

Line 188-189: An alternative and much more likely explanation for the differences between polluted and clean sampling periods would be that all data collected suffer from varying degrees of pollution. It's naive to assume that all air pollution or conversion of SA from SO2 never recycles over the sampling platform.

As we stated in the previous comment, we understand the reviewer's concern with local pollution contamination in our data, and we agree that there are likely periods in our data that include some influence from local pollution from the ship's exhaust. We tried to be transparent about this in our discussion in section 2.3 by showing the differences between the clean and polluted data in Fig 1 (now Fig S2) and discussing pollution, particularly with respect to SA from $SO_2$. While we agree that there is likely some periodic influence of pollution in the data, we do not agree that all of our data is influenced by pollution, especially since we are not aware of any mechanisms through which MSA or IA would be influenced by ship exhaust, yet these vapors show the same increased concentrations during periods of local pollution. Since our measurements covered a full year during which we expect natural variability in the concentrations of these vapors, the higher occurrence of datapoints associated with the "polluted" group confounds the differences between the two groups. We expressed this on lines 178 – 180. Please also note that our interpretation of minimal pollution influence on our data is in reference to our analysis of the seasonal cycles of theses vapors which, as described in the previous comment, remains similar with the inclusion of the "polluted" data as shown in Figs. S2, S3, and S4 (now Figs. S4, S5, and S6).

To clarify this, we have reworded our statement starting on line 177 as follows:

"Overall, the comparisons presented in Figs. S2, S3, S4, S5, and S6 suggest that local pollution has very minimal impact on our analysis of the seasonal cycles for all three gas phase species during the MOSAiC expedition."

We also updated our statement on line 186 for more clarity:

"Therefore, we concluded that our analyses on the seasonal cycles of SA, MSA, and IA were not affected by primary pollution in our measurements, and we did not remove any data during periods of local pollution."

Previous studies (e.g. Bigg et al., 2001) of over-the-pack ice have shown that not only necessary to be able to specify gas phase concentrations in the atmosphere and their possible sources, but we also must understand the thermodynamic structure of the lower atmosphere (typically a well-mixed shallow boundary layer at the surface, only a couple hundred meters deep, capped by a temperature inversion below the free troposphere), the dynamics of the boundary layer, and processes important in exchange between the air and ocean top layers to fully consider the short time variability on the constituents under study. Have any in situ observed meteorological analyses of your data been considered? Suggested for further reading by Tjernström et al., 2012; 2014.

Indeed, temporal changes in boundary layer stability, obtained from radiosondes, local meteorological observations, and in-situ remote sensing instruments, were considered, where a stable, shallow boundary layer was the prevailing condition during the campaign (Jozef et al., 2023). Please note that our study focuses on insights obtained from seasonal cycles of these vapors and their source regions, and not their short-term variability, and hence, we did not elaborate on short term meteorological phenomenon.

Line 191: Clarify if the ppb unit is by volume or mass.

We have added to line 191 that the $SO_2$ mixing ratios were measured "by volume".

What is the meaning of a detection limit of 1 ppb for SO2?

Doesn't it imply that any data below 1 ppb, if detected by the Thermo Fisher Scientific instrument, should be disregarded since it has significant uncertainty? This is especially important as SO2 levels in pack ice during summer and early autumn are typically in the lower 10th of a ppt(v) range (Kerminen et al., Leck and Persson, 1996a; Bigg et al., 2001).

The reviewer is correct, the $SO_2$ data below the detection limit should be disregarded, which is why we clearly stated the detection limit of the $SO_2$ instrument in the methods on line 191, during the analysis of the $SO_2$ data on line 309, and in Figure 3 (now Figure 2) showing the timeseries of the $SO_2$ mixing ratios. Please note that our analysis does not consider the $SO_2$ mixing ratios below the detection limit and instead focused on the observed spikes in $SO_2$, which we expressed on lines 310 – 312 and on line 405 while discussion our results. To add further clarity to our use of the $SO_2$ data, we have added the following statement to line 312.

"SO$_2$ mixing ratios below the detection limit were not considered in our analysis but were included in Fig. 2 to show the continuity of the SO$_2$ measurements during the campaign."

Line 193: How much of the SO2 data was removed due to contamination? I see no black line for the SO2 mixing ratios, only black-filled circles. From this display, it's impossible to resolve the removed data periods.

We did not initially detail the fraction of SO$_2$ data remaining after treating pollution, as handling of pollution in the SO$_2$ dataset was previously described in Angot et al. (2022) and is described in the cited SO$_2$ dataset (Angot et al., 2022). To clarify this information, according to the reviewer's comment, we have added the fraction of polluted data that was removed from the SO$_2$ dataset on line 193 as follows:

"Local pollution affected 62% of the original SO$_2$ timeseries during the campaign."

We also thank the reviewer for pointing out our error in the caption of Figure 3 (now Figure 2). We have corrected the description of the SO$_2$ mixing ratio as the "black line" to "black circles".

Line 315: The seasonality of observed sulfuric acid suggests that high concentrations during winter and spring are related to the Arctic haze phenomenon. Although the paper states that the sulfur chemistry and conversion from SO2to sulfuric acid is out of the scope of the study (Page 12, L 315-316), an explanation must be provided on how gaseous sulfuric acid can be produced in the absence of sunlight. Furthermore, it should be addressed over which distances sulfuric acid in the gas phase can be transported from lower latitudes, concerning its strong tendency to attach to particles and the frequent fog scavenging and wet deposition sinks during its transport from lower latitudes into the Central Arctic.

The reviewer raises a good point about the conversion of SO$_2$ to SA during the polar night period. Similar observations between anthropogenic SO$_2$ and SA have also been observed previously by Sipilä et al. (2021), which we referenced in our discussion connecting the two gaseous sulfur-containing species. Our inverse model simulations suggest that the SO$_2$ originates from smelters near Norilsk, Russia and was transported due to rapid advection northward to the ship's location, which is also confirmed by the coupled FLEXPART/emission inventory simulations. The FLEXPART simulations indicate that rapid transport occurred in as little as 1 day from these source regions. These short-lived SO$_2$ spikes correspond with the timing of warm and moist air mass intrusion events that are also associated with a record-breaking positive phase of the Arctic Oscillation index. Therefore, we hypothesize that oxidation of SO$_2$ could have taken place near the emitted regions, which were further south, such that there could have been some available solar radiation present to initiate partial oxidation of the SO$_2$ to form SA during this time of year, as noted by Sipilä et al. (2021). The high concentrations of SO$_2$ and SA observed at the ship during such rapid transport episodes imply strong emissions of SO$_2$, which is also in agreement with the observations in Sipilä et al. (2021), who reported SO$_2$ concentrations up to 27 ppb. Since these events were observed during winter we expect that wet deposition sinks, including precipitation and fog, are considerably less than during the summer, which is a common feature associated with Arctic Haze transport during both winter and spring.

Figure 1: The frequency range of sulfuric acid, methanesulfonic acid, and iodic acid data in Figure 1 is not annotated on the x-axis.

We thank the reviewer for this helpful suggestion. The reviewer is correct: the violins in Fig 1 (now Fig S2) are normalized by the number of observations. While this is helpful for a comparison of the shape of the data distributions between the "clean" and "polluted" periods, it is missing information about the true frequency of observations, which, as noted in the text, were higher during polluted periods.

We have updated Fig 1 (now Fig S2) and its caption to also include a comparison of the frequency of observations between the "clean" and "polluted" data groups as follows:

[Figure]

**Figure S2. The influence of local pollution on the aerosol precursor vapor concentrations.** The violins show the distribution of the gas phase concentration for the clean (blue) and polluted (red) periods to evaluate the influence of local pollution, from the ship and logistical activities, on A) SA, B) MSA, and C) IA during the campaign. The medians and interquartile ranges of each data subset are represented by the solid and dashed black lines, respectively. Since the distributions of the violins are

normalized by the number of observations, the corresponding frequencies of the observations are included for clarity.

Figure 3. Could you please specify the differences in the sampling locations with the same graphic as shown in Figure 2?

As discussed previously in our response to the general comments, we do not feel that it is necessary to add the different sampling locations in every figure to avoid redundancy. To clarify the change in the ship's location throughout the year, we added a reference to Fig 2c (now Fig 1c) and Fig S1 in the caption of Fig 3 (now Fig 2) as follows:

Line 307: "For the relative location of Polarstern that corresponds to these measurements, refer to Figs. 1c and S1."

Litteratur

Baccarini et al., 2020

Berndt, T., Hoffmann, E.H., Tilgner, A. Et al. Direct sulfuric acid formation from the gas-phase oxidation of reduced-sulfur compounds. Nat Commun 14, 4849 (2023). https://doi.org/10.1038/s41467-023-40586-2.

Bigg, E.K., and C. Leck, 2001, Cloud-active particles over the central Arctic Ocean, J. Geophys. Res., 106 (D23), 32,155-32,166.

Bigg, E.K., C. Leck, and E.D. Nilsson, 2001, Sudden Changes in Aerosol and Gas concentrations in the central Arctic Marine Boundary Layer – Causes and Consequences, J. Geophys. Res., 106 (D23), 32,167-32,185.

Bigg, E.K., and C. Leck, 2008, The composition of fragments of bubbles bursting at the ocean surface, J. Geophys. Res., 113 (D1) 1209, doi:10.1029/2007JD009078.

Cala, B. A., Archer-Nicholls, S., Weber, J., Abraham, N. L., Griffiths, P. T., Jacob, L., Shin, Y. M., Revell, L. E., Woodhouse, M., and Archibald, A. T.: Development, intercomparison, and evaluation of an improved mechanism for the oxidation of dimethyl sulfide in the UKCA model, Atmos. Chem. Phys., 23, 14735–14760, https://doi.org/10.5194/acp-23-14735-2023, 2023.

Duplessis, P., Karlsson, L., Baccarini, A., Wheeler, M., Leaitch, W. R., Svenningsson, B., Leck, C., Schmale, J., Zieger, P., & Chang, R. Y. W., 2024. Highly Hygroscopic Aerosols Facilitate Summer and

Early-Autumn Cloud Formation at Extremely Low Concentrations Over the Central Arctic Ocean. Journal of Geophysical Research: Atmospheres, 129(2), Article e2023JD039159. https://doi.org/10.1029/2023JD039159.

Hamacher-Barth, E, C. Leck, and K. Jansson, 2016, Size-resolved morphological properties of the high Arctic summer aerosol during ASCOS-2008, Atmos. Chem. Phys., 16, 6577–6593.

Heintzenberg, J., C. Leck, and Tunved, P., 2015, Potential source regions and processes of the aerosol in the summer Arctic, Atmos. Chem. Phys., 15, 6487-6502, doi:10.5194/acp-15-6487-2015.

Heintzenberg, J., Tunved, P., Gali, M., and Leck, C., 2017, New particle formation in the Svalbard region 2006–2015, Atmos. Chem. Phys., 17, 10, doi:10.5194/acp-2016-1073.

Igel, A.L., A.M.L. Ekman, C. Leck, M.Tjernström, J. Savre, and J. Sedlar, 2017, The free troposphere as a potential source of arctic boundary layer aerosol particles, Geophysical Research Letters,44, 13, 7053-7060.

Karl, M., A. Gross, C. Leck and L. Pirjola, 2007, Intercomparison of Dimethyl sulfide Oxidation Mechanisms for the Marine Boundary Layer: Gaseous and particulate sulfur constituents, J. Geophys. Res., 112 (D15), D15304.

Karl, M., C. Leck, A. Gross, and L. Pirjola, 2012, A Study of New Particle Formation in the Marine Boundary Layer Over the Central Arctic Ocean using a Flexible Multicomponent Aerosol Dynamic Model. Tellus 64B, 17158, doi: 17110.13402/tel- lusb.v17164i17150.17158.

Karl, M, Leck, C, Coz, E., and Heintzenberg, J., 2013, Marine nanogels as a source of atmospheric nanoparticles in the high Arctic, Geophysical Research. Letters, 40, 3738–3743, doi:10.1002/grl.50661.

Karl, M., C. Leck, F. Mashayekhy Rad, A. Bäcklund, S. Lopez-Aparicio, J. Heintzenberg, 2019, New insights in sources of the sub-micrometre aerosol at Mt. Zeppelin observatory (Spitsbergen) in the year 2015, Tellus B, 71 (1), 1-29.

Karl, M., Pirjola, L., Karppinen, A., Jalkanen, J., Ramacher, M. O., & Kukkonen, J., 2020, Modeling of the Concentrations of Ultrafine Particles in the Plumes of Ships in the Vicinity of Major Harbors.International Journal of Environmental Research and Public Health, 17 (3), 777.https://doi.org/10.3390/ijerph17030777

Kerminen, V. T. and Leck, C.: Sulfur chemistry over the central Arctic Ocean during the summer: Gas-to-particle transformation, J. Geophys. Res, Vol. 106, No. D23, pp. 32087-32099, 2001.

Lawler, M.J., Saltzman, E.S., Karlsson, L., Zieger, P., Salter, M., Baccarini, A., Schmale, J., Leck, C ., 2021, New Insights Into the Composition and Origins of Ultrafine Aerosol in the Summertime High Arctic, Geophysical Research Letters, 48, 1–11.

Leck and Persson, 1996a,b.

Leck, C., and Bigg, E. K.: Biogenic particles in the surface microlayer and overlaying atmosphere in the central Arctic Ocean during summer, Tellus, 57B, 305–316, 2005.

Leck C and Bigg E K (2007) Environment Chemistry 4: 400-403.

Leck, C., and E.K. Bigg, 2010, New particle formation of marine biological origin, Aerosol Science and Technology, 44:570–577.

Leck, C. and Svensson, E., 2015, Importance of aerosol composition and mixing state for cloud droplet activation over the Arctic pack ice in summer, Atmospheric Chemistry and Physics 15 (5), 2545-2568.

Leck, C., E.K. Bigg, D.S. Covert, J. Heintzenberg, W. Maenhaut, E.D. Nilsson, and A. Wiedensohler, 1996, Overview of the Atmospheric research program during the International Arctic Ocean Expedition 1991 (IAOE-91) and its scientific results, Tellus 48B, 136-155.

Leck, C., E.D. Nilsson, K. Bigg, and L. Bäcklin, 2001, The Atmospheric program on the Arctic Ocean Expedition in the summer of 1996 (AOE-96) - A Technical Overview- Outline of experimental approach, instruments, scientific objectives, J. Geophys. Res., 106 (D23), 32,051-32,067.

Leck, C., M. Tjernström, P. Matrai, E. Swietlicki and E.K. Bigg, 2004, Can Marine Micro-organisms Influence Melting of the Arctic Pack Ice?, Eos Vol. 85, No3, 25-36.

Leck, C., Matrai, P., Perttu, A.-M., and Gårdfeldt, K.: Expedition report: SWEDARTIC Arctic Ocean 2018, Swedish Polar Research Secretariat, Stockholm ISBN 978-91-519-3671-0, 2019.

Lucas, D.D, R., G., Prinn, 2002, Mechanistic studies of dimethylsulfide oxidation products using an observationally constrained model, https://doi.org/10.1029/2001JD000843.

Lundén J., G. Svensson and C. Leck, 2007, Influence of meteorological processes on the spatial and temporal variability of atmospheric dimethyl sulfide in the high Arctic summer, J. Geophys. Res.,. 112, D13308.

Lundén, J., Svensson, G., Wisthaler, A., Tjernström, M., Hansel, A. and Leck, C.: The vertical distribution of atmospheric DMS in the high Arctic summer, Tellus, Series B, 62, pp. 160-171, 2010.

Orellana M.V., P.A. Matrai, C. Leck, C. D. Rauschenberg, A. M. Lee, and E. Coz 2011, Marine microgels as a source of cloud condensation nuclei in the high Arctic, PNAS, 108 (33): 13612-13617.

Quinn and Bates, 2011

Pirjola, L., Karl, M., Rönkkö, T., and Arnold, F.: Model studies of volatile diesel exhaust particle formation: are organic vapours involved in nucleation and growth?, Atmos. Chem. Phys., 15, 10435–10452, https://doi.org/10.5194/acp-15-10435-2015, 2015.

Tjernström, M., C. E. Birch, I. M. Brooks, M. D. Shupe, P. O. G. Persson, J. Sedlar, T. Mauritsen, C. Leck, J. Paatero, M., Szczodrak, and C. R. Wheeler, 2012, Meteorological conditions in the central Arctic summer during the Arctic Summer Cloud Ocean Study (ASCOS), Atmos. Chem. Phys., 12, 1-27.

Tjernström, M., Leck, C., Birch, C. E., Bottenheim, J.W., Brooks, B. J., Brooks, I. M., Bäcklin, L., Chang, R. Y.-W., Granath, E., Graus, M., Hansel, A., Heintzenberg, J., Held, Hind, A., de la Rosa, S., Johnston, P., Knulst, J., de Leuuw, G., di Liberto, L., Martin, M., Matrai, P. A., Mauritsen, T., Muller, 
[revised manuscript text omitted]

Heintzenberg, J., Tunved, P., Gali, M., and Leck, C., 2017, New particle formation in the Svalbard region 2006–2015, Atmos. Chem. Phys., 17, 10, doi:10.5194/acp-2016-1073.

Jozef, G. C., Cassano, J. J., Dahlke, S., Dice, M., Cox, C. J., and de Boer, G.: Thermodynamic and kinematic drivers of atmospheric boundary layer stability in the central Arctic during the Multidisciplinary drifting Observatory for the Study of Arctic Climate (MOSAiC), Atmospheric Chem. Phys., 23, 13087–13106, https://doi.org/10.5194/acp-23-13087-2023, 2023.

Karl, M., Gross, A., Leck, C., and Pirjola, L.: Intercomparison of dimethylsulfide oxidation mechanisms for the marine boundary layer: Gaseous and particulate sulfur constituents, J. Geophys. Res., 112, D15304, https://doi.org/10.1029/2006JD007914, 2007.

Karl, M., Leck, C., Coz, E., and Heintzenberg, J.: Marine nanogels as a source of atmospheric nanoparticles in the high Arctic, Geophys. Res. Lett., 40, 3738–3743, https://doi.org/10.1002/grl.50661, 2013.

Karl, M., C. Leck, F. Mashayekhy Rad, A. Bäcklund, S. Lopez-Aparicio, J. Heintzenberg, 2019, New insights in sources of the sub-micrometre aerosol at Mt. Zeppelin observatory (Spitsbergen) in the year 2015, Tellus B, 71 (1), 1-29.

[revised manuscript text omitted]

---

## Author Response (AR2)

Authors' response to Reviewer #5

Reviewer comments in black, Author responses in Red

In this paper, the authors present annual cycles of some important gases that may act as precursors to aerosol formation. They attempt to interpret these either as caused by long range transport, using FLEXPART invers modeling, or from local sources, using rather hand-waiving arguments but no observations. The motivation is to understand the environment for aerosol formation, but the formation of aerosols itself is not supported by any direct observations. The primary results are sort of unique and should be published, the inverse modeling is what it is and but the results are in my mind a bit unclear and unbalanced since the long-range aspects is supported by rather advanced, but uncertain, methods while the discussion on local sources are rather speculative. Based on this, I believe the manuscript could benefit from a clearer focus on what is the core data provided, what are the central conclusions and what is speculation; hence I recommend major revision.

We thank the reviewer for their comments and criticisms of this manuscript. We have addressed each of the reviewer's comments below. We appreciate that the reviewer pointed out a major need for revision for several aspects that were not well explained. This helped us to streamline and clarify the provided information. We address these aspects point by point in the comments below. Other suggestions by the reviewer extend well beyond the scope of this manuscript, but they are good points that could be addressed in the future. We pointed out these cases in our responses to their comments. The reviewer did highlight several points related to our source region identification methods, using FLEXPART and inverse modelling, that required further description in the text, as we have addressed in the specific comments below. In addition, we took the reviewers suggestion to weaken the language in parts of discussion to highlight that there are aspects of this work that remain uncertain and could warrant future investigation by the scientific community.

Major comments:

I have no problem with the quality and handling of the observations. A ship is about the worst place one can do measurements but pollution and other effects on the observations are I my opinion handled with care. My concerns are instead in the modeling and the interpretation of the results.

We thank the reviewer for their general agreement with our handling and presentation of pollution in this data, as we did our best to maintain transparency with respect to pollution and its influence in the datasets that accompany this manuscript. Ship based campaigns are indeed challenging in this regard, however, there are currently very limited options to obtain measurement data in the central Arctic Ocean otherwise.

First, I'd like to underscore that while measuring the complete annual cycle is unique for the central Arctic, this is but one annual cycle and how representative this is for the true (average) annual cycle is unclear. I would therefore have liked some of the space devoted to rather speculative discussions on various local sources to have been spent on documenting how representative this year is compared to other years. And while on the topic of representativity; while I understand there may be source regions that does not impact where MOSAiC happened to be, I would like some discussion on

how regional or local is the source contributions from long-range transport are? In short, now we have one annual cycle at one (or series of) location; what does this mean for pan-central-Arctic conditions? Would the very localized Russian smelter emissions, that dominate the SO2 results, be the same if MOSAiC had been say in the Beaufort gyre? After all, if the motive is Arctic aerosol precursor gases" and how they may change in the future, we need to see the whole Arctic.

We understand the concerns of the reviewer. However, it is challenging to compare this annual cycle with previous years due to the general lack of observations of these precursor gases within the central Arctic Ocean. Understanding these observations over multiples years or from a pan-Arctic perspective are out of the scope of the current work. The other key factor limiting a comparison between our annual cycle and other years is the different locations at different times of year, as also pointed out by the reviewer. We feel that we adequately expressed that localized emissions, such as from the Russian smelter, were observed at *Polarstern's* location due to evidence and insights obtained from the FLEXPART simulations and inverse modeling. In addition, while our measurements are restricted to specific locations at specific times, our observations generally agree with the general knowledge on prevailing source regions. For example, emissions from Siberia are known as a dominant source region of atmospheric transport in the central Arctic during winter (Moschos et al., 2022; Stohl, 2006; Stohl et al., 2013). It is important to point out that the FLEXPART simulations used emission fields from a known emission inventory (ECLIPSE v6b) and the inverse model made use of the measured timeseries data from the MOSAiC expedition and the FLEXPART footprint emission sensitivy (FES) to assign emissions to grids and subsequently simulate concentrations from those emission grids. Interestingly, the source regions identified are comparable between these different methods. Although the source region analysis is qualitative in nature, which we expressed in the manuscript on Line 288 of the main text, it is less "speculative" that attempting to extrapolate the concentrations of these vapors over a larger area in the Arctic. For this reason, we focused our analysis on the seasonal cycles of these vapors and on various source regions that we could identify, qualitatively, to have an influence on their concentrations.

To clarify this in the text, we have toned down various phrases which imply that our observations could be extrapolated to the entire central Arctic Ocean region. These changes are as follows:

- Line 128: "The results of this study are valuable for improving our understanding of these key aerosol precursor vapors and their potential sources from direct observations within the central Arctic, a region which is currently undergoing rapid changes."
- Line 289: "These datasets offer unique insights on the seasonal variability of these vapors during the MOSAiC expedition in the high Arctic."
- Line 436: "Our observations highlight the combined influence of both natural and anthropogenic sources of atmospheric SA during the year, where the highest concentrations occurred in winter/spring."
- Line 462: "This is particularly clear during March and April when the MSA concentration in our measurements starts to increase despite the observation that chl-a concentrations are still low at the northernmost latitudes."
- Line 597: "Our observations suggest that IA concentrations are also strongly linked to seasonal changes in sea ice conditions."
- Line 650: "Our results show the influence of both natural and anthropogenic sources on SA concentrations."
- Line 654: "Localized anthropogenic emissions in Siberia/Northern Russia, especially from the region of Norilsk in Northern Russia, contribute substantially to our observed SA concentrations during winter."

- Line 658: "Our analyses additionally show that biological activity in the open ocean areas south of the marginal ice zone within the Arctic region contributes to enhanced MSA concentrations, an important component of aerosol formation and growth, during late spring through summer."
- Line 663: "Transport from regions south of the marginal ice zone appear to be the primary driver of MSA concentrations in our observations over the sea ice."
- Line 675: "Our observations provide circumstantial evidence that the current seasonal cycles of SA, MSA, IA, and $SO_2$ in the central Arctic Ocean are linked to sea ice conditions and solar radiation due to their role in biological activity and air mass transport from southern regions."

Other examples of where the language of our conclusions was weakened are given in our responses to the detailed comments below.

This could have been addressed using for example CAMS; come to think of it, a comparison to – or maybe even an evaluation of – CAMS would have been really interesting and I'm a bit surprised to see that CAMS is not even mentioned. The inverse modeling is what it is. There is a saying about models: "shit in, shit out". I'm not saying these results are shit, I'm just observing that 30-day back-trajectories are awfully long and uncertain, and that the results are very much dependent in the emissions inventories used. It also seems to be a systematic problem with limiting the domain at 60 °N.

The reviewer raises an important point here. Future analyses could implement and use our dataset to extrapolate over the larger central Arctic region using further modelling such as CAMS, but we conclude that such an analysis is beyond the scope of the work presented here. Our work intends to highlight the novel, year-long seasonal cycle of these condensable vapors and suggest the prevailing source regions that contributed to their concentrations in our observations. This is apparent from the title of the manuscript. To address the concerns of the reviewer about the scope of our work, and to clarify the extent to which we can extrapolate our observations, we have updated the title to indicate that our observations are reported "during the MOSAiC expedition." The title now reads:

"The annual cycle and sources of relevant aerosol precursor vapors in the central Arctic during the MOSAiC expedition."

We thank the reviewer for pointing out the discussion of the 30-day backward particle dispersion simulations, as this is a point that should be clarified further in the text. First, we would like to specify that FLEXPART is an air tracer dispersion model, which is different than the back trajectory analyses to which the reviewer refers to in their comment. Dispersion simulations are systematically better than backward trajectories in the sense that they account for the stochastic nature of atmospheric transport and that the source receptor results are in essence probabilistic (given the use of thousands of particles rather than few single-particle back trajectories) (Stohl et al., 2002).

Please note that we stated that the FLEXPART simulations were carried out for periods "up to 30 days backward in time" on Line 234. This statement was meant to generally describe the FLEXPART simulations that were conducted during the MOSAiC expedition, which are openly available for scientific community to use. These FLEXPART simulations are available for 1, 7, 10, and 30 days backward in time. We did not intend to imply that we used 30-day simulations for each of our

analyses in this work, but we understand the confusion, as we did not explicitly state the length of the FLEXPART simulations that we used in every instance. For example, refer to Fig. S12 which utilized 10-day backward simulations. For clarity we have added the description of the FLEXAPART air tracer ages used for various aspects of our analysis in the text. Refer to our response to the first detailed comment below for the locations where the text was updated.

We would also like to point out that the FLEXPART simulations were not performed specifically for our analysis; we simply used the results of the FLEXPART simulations as a tool to gain further insights into possible source regions in our analysis and used inverse modelling to fill in gaps for species not in the emission inventories. It is for this reason that the domain was limited to 60 °N, as the simulations were performed independently from the analysis presented in this manuscript.

It would have been useful to show the emissions that goes into these calculations in the paper. For example, the inverse modeling of SO2 does not fit all that well with the observations; most of the modeled annual cycle have values well above the instrument threshold, while there is almost no useful SO2 observations at all (above the instrument threshold). While the few spikes in SO2 coincide with the inverse modeling maximum, the modeling indicates a sharp increase in winter starting already in December, the measured SO2 have as low values as there is later, in summer, and still the modelled concentrations for summer is as high as early in the year. I guess I wonder how the inverse modeling works when there isn't any data? I have no problem believing the peak in the inferred concentrations is connected to the peaks in the observations; I just think the linkage is weak.

We do not understand the reviewer's concerns presented in this comment. The $SO_2$ emissions included in the emission inventory, as well as their geographic extent, are presented in Fig. S11. We also specified in the caption of Fig. S11 that the emission inventory uses anthropogenic $SO_2$ emissions and that FLEXPART converts this $SO_2$ to particulate sulfate, or $SO_4$-S. This was also specified on Line 366, where we stated that "FLEXPART treated anthropogenic $SO_2$ emissions as $SO_4$-S, which yields the $SO_4$-S weighted influence from anthropogenic sources." Please also note that we explicitly referred to Fig. S11 on Line 371: "Refer to Fig. S11 for a more detailed description of anthropogenic $SO_4$-S emissions from each source region in the emission inventory."

The concentrations simulated by the inverse model are also shown in their respective figures (Figs. 4, 5, & 6), both as the source region footprint maps (panel a) and then their simulated influence during the year (panel b). These footprint maps were determined using the inverse model itself, as described in detail in Section 2.6 of the methods. These figures clearly show the emissions and their geographic locations, determined from the inverse model, that were combined with the FLEXPART air tracer data to calculate the simulated concentrations presented in panel b of these figures.

We disagree with the reviewer's statements about the inverse model results of $SO_2$ in Fig. 4. There are two separate y-axes in Fig. 4: the left axis for the $SO_2$ mixing ratios simulated by the inverse model and the right axis for the measured $SO_2$ mixing ratios. None of the simulated $SO_2$ mixing ratios exceed the instrumental detection limit of 1 ppb. The detection limit associated with the measurements, indicated by the grey shaded area, may have caused confusion here. To clarify that the shaded region showing the detection limit is associated with the measurement data on the right axis, we added a statement to the figure caption to state this on Line 207:

> "The measured SO2 mixing ratio, presented as a rolling 10-minute median during the year, is included on the right axis for context. Note that the shaded area shows the regions where the SO2 measurement data are below the detection limit."

We do not understand why the reviewer suggested that the inverse model shows a sharp increase in $SO_2$ in December, as this is not shown in the data presented in Fig. 4b. To calculate the simulated concentrations, we used the complete $SO_2$ timeseries for the year, including the data below the detection limit. While the values below the $SO_2$ detection limit are not relevant for quantitative analysis, these low values could still be used to determine the footprint map in the inverse model (i.e., identifying regions with very low emissions). In general, the inverse model and source region identification work are qualitative, as we specified on Line 288 of the main text. The key point here, with which the reviewer agrees, is that the simulated $SO_2$ mixing ratios spiked while we observed measured spikes in $SO_2$ mixing ratios that correspond to air masses from the Russian smelter region, which is also consistent with geographic extent of known sulfur sources in the ECLIPSE v6b emission inventory data as shown in Fig. S11.

The SA source is very diverse compared to that of SO2 and almost all of it is over land surfaces (except for the Labrador Sea and a little in the Bering Sea). Prudhoe Bay is discussed and while it does fall within the green box, it seems like the largest sources in that box are further south, over land. In the discussion the ocean and DMS is discussed as a source, however, the modeling has no chemistry and land surfaces seems to dominate as source regions. There are no measurements of DMS; instead MSA is presented. But MSA lifetime is order of a few hours (line 519), so it is in fact incorrect to even talk about long-rang transport of MSA (such as e.g. line 507). It has to be DMS that is emitted, transported and then oxidized, to MSA and also SO2. To deal with this, chlorophyll-a is used as a proxy - for what, DMS or MSA emissions? It seems to me there are an awful number of hopeful guesses in this chain that I think is not very clearly described. That most of the DMS is probably long-range transported is no news; that has been published decades ago. Much of the source regions for MSA (or is it DMS?) are discussed to be tied to biological activity in the MIZ, but it appears that the authors have no clear picture of where the MIZ typically is located during the biologically most active part of the season. In the North-American sector, Labrador Sea qualify, but else most of the sources are – again – over land. In the North-Atlantic, the source regions are way south of where the MIZ would be; there's for example nothing in the Greenland Sea/Fram Strait, nothing north of Svalbard and nothing over the Siberian shelf area. Where does the DMS emitted here end up if not in the Arctic? This goes back to my previous question about locality; how local are the impact from different source regions?

It is unclear why the reviewer expresses such confusion between DMS and MSA in this manuscript. It is true that DMS is emitted, which is then subsequently oxidized during transport to form MSA. We stated this repeatedly throughout the text, for example in the following locations:

- The paragraph in the introduction, starting on Line 64.
- Line 82: "The concentration of gaseous DMS and its oxidation products (SA and MSA)…"
- Line 293: "…during which we expect DMS emissions that lead to the observed maximum MSA concentrations."
- Line 469: "…was used to evaluate the connection between DMS emissions (and subsequent formation of MSA) with biological activity…"
- Line 479: "…increase in chl-a and MSA concentrations is unsurprising, as gaseous DMS, and subsequently MSA, is a product of biological activity…"
- Line 490: "…potential air mass exposure to oceanic regions with biological influence, and hence potential DMS (MSA) emissions."
- Line 509: "…or important sources of DMS, the precursor of MSA, …"

- Line 525: "…we can infer that MSA production from DMS…"
- Line 527: "…we conclude that transport of DMS from the regions > 60°N in Fig. 6a, followed by subsequent chemical processing during transport, could explain our MSA measurements…"
- Line 608: "…organisms that produce DMS (MSA)…"

Based on this, and the other statements that the reviewer pointed out about lack of chemistry in the models used in our analysis, it should be clear that any reference to emissions during the discussion of MSA refer to emissions of DMS, followed by the subsequent oxidation and production of MSA. We made it a point to mention this in the discussion of each aspect of the MSA discussion, in addition to the points already mentioned (emissions: Line 496; transport: Line 517, Line 520-524; and the chl-a proxy, Line 470).

We also do not understand why the reviewer suggests that most of the sources of MSA were determined to be above land. From Fig. 6b, many of the inverse model polygons are located over the ocean or near coastal areas. The two exceptions to this are polygon "a" and "f", or the blue and brown polygons, respectively. We would like to point out that for these polygons, we wrote the following in the figure caption on Line 537 as follows:

> "Due to the limited domain of the FLEXPART simulations (> 60°N), source regions polygons "a" and "f" may represent the contribution of MSA transport from regions further south than the polygons depicted on the map, such as the oceanic regions on the western coast of North America and the Bering Sea, respectively."

This is also explicitly stated on Line 505:

> "Note that due to the limited domain of the FLEXPART simulations (> 60°N), source regions polygons a and f in Fig. 6a may represent the contribution of MSA transport from regions further south than the polygons depicted on the map, which could be associated with oceanic regions on the western coast of North America and Bering Sea, respectively."

Given the knowledge that DMS can be transported over distance, as pointed out by the reviewer, and subsequently oxidized during transport, this is a plausible hypothesis.

We also made it a point to discuss the influence of transport and air mass history for MSA. Emissions alone are not enough to describe measurements over the Arctic. There must be emissions combined with transport. We stated this on Line 487, where we introduced the chl-a proxy:

> "The source regions of the observed air masses in the central Arctic would need to correspond with the regions of enhanced biological activity to explain the MSA measured at the ship. Therefore, we coupled the FLEXPART air tracer simulations with the oceanic chl-a concentrations to calculate an index that quantifies potential air mass exposure to oceanic regions with biological influence, and hence potential DMS (MSA) emissions."

To again address the reviewer's comments about locality of emissions in the Arctic—we are reporting observations from a given point in time and space across a large region. It's reasonable to assume that emissions of DMS from other known source regions, as mentioned by the reviewer, could end up in the Arctic, but these regions were not identified to be influential in our measurements at *Polarstern*'s location at those times of year. We are not arguing against other sources—we are simply identifying sources associated with our dataset, which still adds value given the limited number of

observations in the central Arctic region—particularly for the precursor vapors highlighted in this study. Again, to make this clearer, we have added "during the MOSAiC expedition" to the title of the manuscript.

Finally, the results for IA are very interesting but the hand-waiving on the reasons for the two peaks, in spring and autumn, is not very impressing. Almost no references are given and no observations are offered to support the arguments of melting and refreezing blocking up the brine channels(?). This is nothing but a hypothesis; an interesting one but it feels the authors dwell on this for quite a while in the hope to convince the reader, while the fact is that we have clue. I stagger at the words "provides evidence" (Line 624); I can't find a shred of "evidence" here other than what could be called "circumstantial evidence"!

The mechanism of IA formation and its sources in the atmosphere are still not well understood and require further investigation, which is the key takeaway from Section 3.3. We do not intend to convince the reader of anything in this section, but rather, we present a discussion of what has been hypothesized about the seasonality of iodine compounds from previous studies and compare them with our study. We do not understand the reviewer's comment that there are no references given, as our discussion includes a number of citations to previous studies. In fact, we cited various sources that identify seasonal peaks in iodine compounds and others that propose mechanisms that could support why these peaks are observed, including the study that the proposed mechanism of iodine compound transport through brine channels before the melt season. We again want to point out that the brine channel transport is a hypothesis presented from previous decades by Saiz-Lopez et al. in 2007 and 2015, which was cited appropriately in the text; we are not proposing a new mechanism here. We have changed the wording on Line 624 from "provides evidence" to "suggests", according to the reviewer's concern with our statement about the studies that discusses iodine transport through brine channels in the sea ice.

Our analysis of the seasonal cycle in this work in is not sufficient to resolve these processes, but a more detailed analysis of IA is planned to address some of these ongoing hypotheses. We specified this in the main text on Line 648:

> "However, our analysis focuses on the seasonal cycle, which is not sufficient to resolve the relative contributions from these processes on IA concentrations. As such, atmospheric iodine processes, especially in the Arctic, require further investigation. A more detailed analysis of atmospheric IA formation mechanisms during the MOSAiC expedition will be given in a separate study."

We have also removed the statement about brine channels in the paragraph discussing IA in the conclusions on line 670:

> "In addition, the thinning sea ice could facilitate the exchange of iodine into the atmosphere and further reaction with O3 to form IA."

Finally, for a paper that confesses to not deal with aerosols in general or especially not with NPF, there's quite a bit of text on it.

This comment appears to be a follow up to one of our responses to reviewer #4, where we stated:

"Please note that this paper is not focused on the role of precursor gases in aerosol formation and growth, and any mention of these processes in the introduction is used to give context for our measurements and analysis. The scope of this work focuses on understanding the seasonal cycle of vapor that are known to be potential aerosol precursors and the observed source regions that contributed to their concentrations in our measurements."

To highlight why these aerosol precursor vapors are relevant, it is necessary to include discussions on their potential implications for aerosol process or new particle formation (NPF). Otherwise, there is no context for our measurements. The observations and the science presented in this manuscript do not focus on aerosol processes or NPF, but the paper would be incomplete without discussion connecting these condensable vapors to aerosol processes.

Detailed comments:

Line 234: Awfully long trajectories(!); the added uncertainly from this should be addressed. There is a literature on this; use it.

As mentioned above, the statement about FLEXPART simulations "up to 30 days backward in time" is general to the FLEXPART data produced during the MOSAiC campaign, but it is not specific to the length of the dispersion simulations used in each aspect of our analysis. For example, we specified for the chl-a influence index in the caption of Fig. 1 on Line 303, as well as in the caption of Fig. S12 in the supplemental information that we used 10-day backward simulations. Backward dispersion simulations up to 30 days were used in the inverse model, but this should not be a major concern as more weight was placed on the shorter simulations, and many of the simulated particles were outside of the domain after 30 days. We have updated the main text to specify the length of FLEXPART simulations used for each aspect of the analysis as follows:

- Line 226: "Specifically, we used the ECLIPSE v6b (Evaluating the Climate and Air Quality Impacts of Short-Lived Pollutants) emission inventory and 10-day backward air tracer simulations from FELXAPRT to estimate source regions of anthropogenic sulfur, from $SO_2$ emissions, as described in section 3.1."
- Line 283: "The calculation of these emission fields in the inverse model used the FLEXPART FES air tracer from 1, 7, 10, and 30 day simulations backward in time, where the shorter simulations were given more weight in the estimated emissions."
- Line 495: "The index, called the sea surface chl-a influence index, was obtained by multiplying the residence time (in seconds) of the FLEXPART air tracer (based on 10-day backward simulations) below 100 m altitude with the corresponding chl-a concentration maps (in $mg \cdot m^{-3}$)."

Without directly providing the citation the reviewer is referring to here, we can only assume that they are referring to one of the studies from ASCOS (Tjernström et al., 2012, 2014). We feel that our updated descriptions specifying the lengths of the backward particle dispersion simulations used in our analyses are sufficient, as well as our claims that the source region analyses are qualitative in nature.

Lines 335-341: Here the peak SA concentrations in early spring is discussed as if it and the so-called Arctic haze are two things that here happen to coincide. Aren't they two sides of the same coin?

We thank the reviewer for their comment. In this section we are stating that the high winter/spring concentrations of SA are resulting from transport of air masses further south associated with Arctic haze—so indeed, two sides of the same coins, as the reviewer stated. This is stated clearly on Line 339:

> "… which suggests that Arctic haze, or anthropogenic pollution, is a key source of the high SA concentrations in winter and the dominant source of SA during the annual cycle."

Line 368-369: I think this is an overstatement, or maybe wishful thinking. There is some agreement, but I'm not very much impressed.

We thank the reviewer for their comment. In this statement, we were highlighting the general consistency in the timing of temporal spikes for each sulfur species in Fig. 2, which occurs predominately during January and February. We have weakened the sentence on Line 368 – 369 according to the reviewer's concern, as follows:

> "Overall, the SO4-S simulations peak during the same time of year as the measured gas phase sulfur species, especially in January and February when we observed temporal spikes of each species."

Line 439-440: I wonder what the source is here; I think the authors are wrong. Most of the long-range meridional transport occurs in association with weather systems, and with those there is usually plenty of precipitation. In the Arctic, there is very little convective precipitation at all, except over land in summer.

It is not clear on the point with which the reviewer's disagrees here. The reviewer agrees that there is precipitation during poleward transport in summer and that there is convective precipitation over land. It is worth noting that the prevailing anthropogenic sources are located on land, and therefore, we feel that our statement about convective precipitation increasing in summer, thereby decreasing northward transport of anthropogenic pollutants, is not inaccurate. We would also like to point out that we did not specify that the convective precipitation was occurring in the Arctic itself but during transport from regions further south. The reviewer's main concern here seems to be with the use of "convective" to describe the precipitation. To address their concern, we have rephrased the sentence on Line 439 – 440 follows:

> "In contrast, there is more precipitation during summer, which limits northward transport of anthropogenic pollution…"

Lines 441-442: Maybe I misunderstand, but here the authors seem to suggest that Arctic haze, which occurs in spring, is the source of SA in summer. That would mean SA has a life-time of months. At the very least drop the "demonstrates"

The point here was to compare the general SA concentrations during summer to the higher concentrations observed during the Arctic haze period. We did not intend to suggest that Arctic haze remained a source of SA in summer. We rephrased this sentence to be clearer and avoid such confusion. Line 441 now reads as follows:

"Based on this, our results show that SA concentrations from DMS emissions in summer are smaller in magnitude than anthropogenic sulfur sources from Arctic haze in spring."

Lines 500-516: The link between DMS and MIZ biology is certainly not new, but the results here indicate regions that are either over land or way south of any MIZ, so this doesn't work.

As discussed in our response to the general comment above, the polygons associated with DMS (MSA) are not predominately over land but are south of the marginal ice zone or are near coastal areas. In the same discussion, we also specified that regions south of the marginal ice zone were influential in our data. We refer to lines 501 – 505 where we discussed these topics:

"The key insight obtained from the inverse model results, shown in Figs. 6a and 6b, is that regions south of the marginal ice zone appear to be the most influential on MSA concentrations over the central Arctic. More specifically, the inverse model identifies several oceanic regions as potential sources of MSA in our observations, where the Kara, Barents, Norwegian, and Labrador Seas are the most prevalent source regions during spring and summer (polygons b, c, and d in Fig. 6a)."

We stated this again on Lines 509 – 512:

"Previous research has shown that the regions identified in Fig. 6a are biologically active or important sources of DMS, the precursor of MSA, from May to August (Hulswar et al., 2022; Lana et al., 2011; Leck and Persson, 1996a; Terhaar et al., 2021), which is consistent with the chl-a satellite data and again highlights the importance of air mass transport from biologically active source regions further south on our MSA measurements."

In these statements, we intended to emphasize regions south of the marginal ice zone and did not mean to imply that the DMS was sourced directly at the marginal ice zone, according to our qualitative source region analyses.

The statement on Line 513 - 515 is a comparison of our results to previous studies that have also identified DMS or MSA in the aerosol phase south of the marginal ice zone, which is consistent with our results. To minimize confusion, we have changed "near" on Line 514 to "south of".

Line 569: During the ASCOS expedition, CCN-limited conditions happened during less than one day, out of an ice camp that lasted ~3 weeks. It has been seen also in other datasets, so I wouldn't call it rare, but it is still "unusual" rather than "common" or as here "often" occurring.

We thank the reviewer for highlighting this information. This is a good point. Our intent in this sentence was to highlight the generally low background aerosol concentrations in the summertime Arctic atmosphere, which can indeed lead to periods of CCN-limited conditions. We have rephrased the sentence on Line 569 to be more accurate, according to the reviewer's suggestion:

"Once in the aerosol phase, secondary particles containing MSA are sufficiently hygroscopic such that they may enhance CCN concentrations in the summertime Arctic atmosphere, which can experience periods of CCN-limited conditions (Mauritsen et al., 2011)."

Lines 580-583: Drop this; when you get to it, let us know but for now there is no information here.

We thank the reviewer for this suggestion. Instead of completely dropping this text, which does not provide much new information as the reviewer noted, we have rephrased this sentence to point out to the reader that future studies are necessary to resolve such processes. Line 580 now reads as follows:

> "Note that these aerosol processes involving MSA are beyond the scope of the seasonal analysis presented here, however, future work should aim to investigate the role of MSA in event-level analyses of the mechanism of NPF and aerosol chemistry in the central Arctic region."

Line 624: I stagger at the words "provides evidence"! I can't find a shred of evidence here and, moreover, the whole sentence sound like a contradiction: "… provides evidence … could be …".

We have changed the wording on Line 624 from "provides evidence" to "suggests" to account for the reviewer's concern.

Line 663-664: This is a really myopic perspective. There is certainly clouds around the whole year, that must have formed on some sort of particle, so if IA is not present all the year, it has to be something else – duh!

It is unclear what the reviewer intends to criticize with this comment. Please note that the statement on lines 663 – 664 is not discussing CCN or the composition of all aerosols in the Arctic during the year. Instead, this statement is related to previous observations of the chemical mechanism of NPF in the central Arctic, which due to a very limited number of measurements over the central Arctic Ocean with sufficient instrumentation to resolve this process, is not well characterized. We mentioned IA in the context of two studies that did characterize the mechanism of NPF in the Arctic: one in the central Arctic during autumn, and another from land-based sites in Villum and Svalbard. There has been a lot of emphasis placed on the role of IA as a primary driver for secondary aerosol formation in the Arctic, but it's worth presenting here that the mechanism of NPF during the whole year is likely more complicated. We have decided to keep this discussion in the text.

Lines 687-688: It actually doesn't; look where the MIZ is in summer!

We thank the reviewer for pointing this out. In a similar fashion to the comment on Lines 500 – 516, we have changed the word "near" to "south of" on Line 687, which is consistent with our results.

**References**

Moschos, V., Schmale, J., Aas, W., Becagli, S., Calzolai, G., Eleftheriadis, K., Moffett, C. E., Schnelle-Kreis, J., Severi, M., Sharma, S., Skov, H., Vestenius, M., Zhang, W., Hakola, H., Hellén, H., Huang, L., Jaffrezo, J.-L., Massling, A., Nøjgaard, J. K., Petäjä, T., Popovicheva, O., Sheesley, R. J., Traversi, R., Yttri, K. E., Prévôt, A. S. H., Baltensperger, U., and Haddad, I. E.: Elucidating the present-day chemical

composition, seasonality and source regions of climate-relevant aerosols across the Arctic land surface, Environ. Res. Lett., 17, 34032, https://doi.org/10.1088/1748-9326/ac444b, 2022.

Stohl, A.: Characteristics of atmospheric transport into the Arctic troposphere, J. Geophys. Res. Atmospheres, 111, https://doi.org/10.1029/2005JD006888, 2006.

Stohl, A., Eckhardt, S., Forster, C., James, P., Spichtinger, N., and Seibert, P.: A replacement for simple back trajectory calculations in the interpretation of atmospheric trace substance measurements, Atmos. Environ., 36, 4635–4648, https://doi.org/10.1016/S1352-2310(02)00416-8, 2002.

Stohl, A., Klimont, Z., Eckhardt, S., Kupiainen, K., Shevchenko, V. P., Kopeikin, V. M., and Novigatsky, A. N.: Black carbon in the Arctic: the underestimated role of gas flaring and residential combustion emissions, Atmospheric Chem. Phys., 13, 8833–8855, https://doi.org/10.5194/acp-13-8833-2013, 2013.

Tjernström, M., Birch, C. E., Brooks, I. M., Shupe, M. D., Persson, P. O. G., Sedlar, J., Mauritsen, T., Leck, C., Paatero, J., Szczodrak, M., and Wheeler, C. R.: Meteorological conditions in the central Arctic summer during the Arctic Summer Cloud Ocean Study (ASCOS), Atmospheric Chem. Phys., 12, 6863–6889, https://doi.org/10.5194/acp-12-6863-2012, 2012.

Tjernström, M., Leck, C., Birch, C. E., Bottenheim, J. W., Brooks, B. J., Brooks, I. M., Bäcklin, L., Chang, R. Y.-W., de Leeuw, G., Di Liberto, L., de la Rosa, S., Granath, E., Graus, M., Hansel, A., Heintzenberg, J., Held, A., Hind, A., Johnston, P., Knulst, J., Martin, M., Matrai, P. A., Mauritsen, T., Müller, M., Norris, S. J., Orellana, M. V., Orsini, D. A., Paatero, J., Persson, P. O. G., Gao, Q., Rauschenberg, C., Ristovski, Z., Sedlar, J., Shupe, M. D., Sierau, B., Sirevaag, A., Sjogren, S., Stetzer, O., Swietlicki, E., Szczodrak, M., Vaattovaara, P., Wahlberg, N., Westberg, M., and Wheeler, C. R.: The Arctic Summer Cloud Ocean Study (ASCOS): overview and experimental design, Atmospheric Chem. Phys., 14, 2823–2869, https://doi.org/10.5194/acp-14-2823-2014, 2014.